# Log-Sum-Exponential Estimator for Off-Policy Evaluation and Learning

**Armin Behnamnia** [* 1]   **Gholamali Aminian** [* 2]   **Alireza Aghaei** [3]   **Chengchun Shi** [4]   **Vincent Y. F. Tan** [5]
**Hamid R. Rabiee** [1]

## Abstract

Off-policy learning and evaluation leverage logged bandit feedback datasets, which contain context, action, propensity score, and feedback for each data point. These scenarios face significant challenges due to high variance and poor performance with low-quality propensity scores and heavy-tailed reward distributions. We address these issues by introducing a novel estimator based on the log-sum-exponential (LSE) operator, which outperforms traditional inverse propensity score estimators. Our LSE estimator demonstrates variance reduction and robustness under heavy-tailed conditions. For off-policy evaluation, we derive upper bounds on the estimator's bias and variance. In the off-policy learning scenario, we establish bounds on the regret—the performance gap between our LSE estimator and the optimal policy—assuming bounded $(1 + \epsilon)$-th moment of weighted reward. Notably, we achieve a convergence rate of $O(n^{-\epsilon/(1+\epsilon)})$ for the regret bounds, where $\epsilon \in [0, 1]$ and $n$ is the size of logged bandit feedback dataset. Theoretical analysis is complemented by comprehensive empirical evaluations in both off-policy learning and evaluation scenarios, confirming the practical advantages of our approach. The code for our estimator is available at the following link: `https://github.com/armin-behnamnia/lse-offpolicy-learning`.

## 1. Introduction

Off-policy learning and evaluation from logged data are important problems in reinforcement learning (RL). The logged bandit feedback (LBF) dataset represents interaction logs of a system with its environment, recording context, action, propensity score (i.e., the probability of action selection for a given context under the logging policy), and feedback (reward). It is used in many real applications, e.g., recommendation systems (Aggarwal, 2016; Li et al., 2011), personalized medical treatments (Kosorok & Laber, 2019; Bertsimas et al., 2017), and personalized advertising campaigns (Tang et al., 2013; Bottou et al., 2013). The literature has considered this setting from two perspectives, off-policy evaluation (OPE) and off-policy learning (OPL). In *off-policy evaluation*, we utilize the LBF dataset from a logging (behavioural) policy and an estimator constructed via e.g., inverse propensity score (IPS) weighting, to evaluate (or estimate) the performance of a different target policy. In *off-policy learning*, we leverage the estimator and LBF dataset to learn an improved policy with respect to logging policy.

In both scenarios, OPL and OPE, the IPS estimator is proposed (Thomas et al., 2015; Swaminathan & Joachims, 2015a). However, this estimator suffers from significant variance in many cases (Rosenbaum & Rubin, 1983). To address this, some improved IPS estimators have been proposed, such as the IPS estimator with the truncated ratio of policy and logging policy (Ionides, 2008b), IPS estimator with truncated propensity score (Strehl et al., 2010), self-normalizing estimator (Swaminathan & Joachims, 2015b), exponential smoothing (ES) estimator (Aouali et al., 2023), implicit exploration (IX) estimator (Gabbianelli et al., 2023) and power-mean (PM) estimator (Metelli et al., 2021).

In addition to the significant variance issue of IPS estimators, there are two more challenges in real problems: estimated propensity scores and heavy-tailed behaviour of weighted reward due to noise or outliers. Previous works such as Swaminathan & Joachims (2015a), Metelli et al. (2021), and Aouali et al. (2023) have made assumptions when dealing with LBF datasets. Specifically, these works assume that rewards are not subject to perturbation (noise) and that true propensity scores are available. However, these assumptions

---

*Equal contribution   [1]Department of Computer Engineering, Sharif University of Technology [2]The Alan Turing Institute, London, UK [3]Department of Computer Science, Stony Brook University [4]Department of Statistic, London School of Economics [5]Department of Electrical and Computer Engineering, National University of Singapore. Correspondence to: Gholamali Aminian <gaminian@turing.ac.uk>, Hamid R. Rabiee <rabiee@sharif.edu>.

*Proceedings of the 42$^{nd}$ International Conference on Machine Learning*, Vancouver, Canada. PMLR 267, 2025. Copyright 2025 by the author(s).

may not hold in real-world scenarios.

*Noisy or heavy-tailed reward:* three primary sources of noise in reward of LBF datasets can be identified as (Wang et al., 2020): (1) *inherent noise*, arising from physical conditions during feedback collection; (2) *application noise*, stemming from uncertainty in human feedback; and (3) *adversarial noise*, resulting from adversarial perturbations in the feedback process. Furthermore, In addition to noisy (perturbed) rewards, a heavy-tailed reward can be observed in many real-life applications, e.g., financial markets (Cont & Bouchaud, 2000) and web advertising (Park et al., 2013), the rewards do not behave bounded and follows heavy-tailed distributions where the variance is not well defined.

*Noisy (estimated) propensity scores:* The access to the exact values of the propensity scores may not be possible, for example, when human agents annotate the LBF dataset. In this situation, one may settle for training a model to estimate the propensity scores. Then, the propensity score stored in the LBF dataset can be considered a noisy version of the true propensity score.

Therefore, there is a need for an estimator that can effectively manage the heavy-tailed condition and noisy rewards or propensity scores in the LBF dataset.

### 1.1. Contributions

In this work, we propose a novel estimator for off-policy learning and evaluation from the LBF dataset that outperforms existing estimators when dealing with estimated propensity scores and heavy-tailed or noisy weighted rewards. The contribution of our work is three-fold.

**First,** we propose a novel non-linear estimator based on the log-sum-exponential (LSE) operator which can be applied to both OPE and OPL scenarios. This LSE estimator effectively reduces variance and is applicable to noisy propensity scores, heavy-tailed reward and noisy reward scenarios.

**Second,** we provide comprehensive theoretical guarantees for the LSE estimator's performance in OPE and OPL setup. In particular, we first provide bounds on the regret, i.e. the difference between the LSE estimator performance and the true average reward, under mild assumptions. Then, we studied bias and variance of the LSE estimator and its robustness under noisy and heavy-tailed reward scenarios.

**Third,** we conducted a set of experiments on different datasets to show the performance of the LSE in scenarios with true, estimated propensity scores and noisy reward in comparison with other estimators. We observed an improvement in the performance of the learning policy using LSE in comparison with other state-of-the-art algorithms under different scenarios.

## 2. Log-Sum-Exponential Estimator

**Notation:** We adopt the following convention for random variables and their distributions in the sequel. A random variable is denoted by an upper-case letter (e.g., $Z$), an arbitrary value of this variable is denoted with the lower-case letter (e.g., $z$), and its space of all possible values with the corresponding calligraphic letter (e.g., $\mathcal{Z}$). This way, we can describe generic events like $\{Z = z\}$ for any $z \in \mathcal{Z}$, or events like $\{g(Z) \leq 5\}$ for functions $g : \mathcal{Z} \to \mathbb{R}$. $P_Z$ denotes the probability distribution of the random variable $Z$. The joint distribution of a pair of random variables $(Z_1, Z_2)$ is denoted by $P_{Z_1, Z_2}$. The cardinality of set $\mathcal{Z}$ is denoted by $|\mathcal{Z}|$. We denote the set of integer numbers from 1 to $n$ by $[n] \triangleq \{1, \cdots, n\}$. In this work, we consider the natural logarithm, i.e., $\log(x) := \log_e(x)$. For two probability measures $P$ and $Q$ defined on the space $\mathcal{Z}$ and a probability measure $\tilde{P}$ define on the space $\mathcal{Y}$, the *total variation distance* between two densities $P$ and $Q$, is defined as $\mathbb{TV}(P, Q) := \int_{\mathcal{Z}} |P - Q|(\mathrm{d}z)$. We also define the conditional total variation distance as $\mathbb{TV}_c(P_{Z|Y}, Q_{Z|Y}) := \int_{\mathcal{Y}} \tilde{P}_{Y=y} \int_{\mathcal{Z}} |P_{Z=z|Y=y} - Q_{Z=z|Y=y}|(\mathrm{d}z)\mathrm{d}y$.

**Main Idea:** Inspired by the log-sum-exponential operator with applications in multinomial linear regression, naive Bayes classifiers and tilted empirical risk (Calafiore et al., 2019; Murphy, 2012; Williams & Barber, 1998; Li et al., 2023), we define the LSE estimator with parameter $\lambda < 0$,

$$\mathrm{LSE}_\lambda(\mathbf{Z}) = \frac{1}{\lambda} \log \left( \frac{1}{n} \sum_{i=1}^{n} e^{\lambda z_i} \right), \tag{1}$$

where $\mathbf{Z} = \{z_i\}_{i=1}^{n}$ are samples from the positive random variable $Z$. The key property of the LSE operator is its robustness to noisy samples in a limited number of data samples. Here a noisy sample, by intuition, is a point with abnormally large positive $z_i$. Such points vanish in the exponential sum as $\lim_{z_i \to +\infty} e^{\lambda z_i} = 0$ for $\lambda < 0$. Therefore the LSE operator ignores terms with large values for negative $\lambda$. The robustness of LSE has also been explored in the context of supervised learning by Li et al. (2023) from a practical perspective. Furthermore, in Appendix (App) C, we discuss the connection between the LSE and entropy regularization.

**Motivating example:** We provide a toy example to investigate the behaviour of LSE as a general estimator and its difference from the Monte-Carlo estimator (a.k.a. simple average) for *mean estimation*. Suppose that $Z$ is distributed as a Pareto distribution with scale $x_m$ and shape $\zeta$. Note that for $Z \sim \mathrm{Pareto}(x_m, \zeta)$ as a heavy-tailed distribution, we have $f_Z(z) = \frac{\zeta x_m^\zeta}{z^{\zeta+1}}$. Let $\zeta = 1.5$ and $x_m = \frac{1}{3}$, then we have $\mathbb{E}[Z] = \frac{\zeta x_m}{\zeta - 1} = 1$. The objective is to estimate $\mathbb{E}[Z]$ with $n$ independent samples drawn from the Pareto distribution. We set $n \in \{10, 50, 100, 1000, 10000\}$ and

compute the Monte-Carlo (a.k.a. simple average) and LSE estimation of the expectation of $Z$. Table 1 shows that LSE (with $\lambda = -0.1$) effectively keeps the variance and mean-square error (MSE), low without significant side-effects on bias. We also observe that the LSE estimator works well under heavy tail distributions.

Table 1: Bias, variance, and MSE of LSE (with $\lambda = -0.1$) and Monte-Carlo estimators. We run the experiment 10000 times and report the variance, bias, and MSE of the estimations.

|  | Estimator | $n = 10$ | $n = 50$ | $n = 100$ | $n = 1000$ | $n = 10000$ |
|---|---|---|---|---|---|---|
| Bias | Monte-Carlo | 0.0154 | 0.0155 | 0.0083 | 0.0061 | 0.0044 |
|  | LSE | 0.1576 | 0.1606 | 0.1616 | 0.1624 | 0.1629 |
| Variance | Monte-Carlo | 1.5406 | 1.5289 | 1.3229 | 1.0203 | 0.8384 |
|  | LSE | 0.1038 | 0.0616 | 0.0443 | 0.0335 | 0.0268 |
| MSE | Monte-Carlo | 1.5409 | 1.5292 | 1.3229 | 1.0203 | 0.8384 |
|  | LSE | 0.1287 | 0.0874 | 0.0704 | 0.0598 | 0.0534 |

## 3. Related Works

We categorize the estimators based on their approach to reward estimation. Estimators that incorporate reward estimation techniques are classified as model-based estimators. In contrast, those that work without reward estimation are termed model-free estimators. Below, we review model-based and model-free estimators. Furthermore, we study the estimators which are designed for unbounded reward (heavy-tailed) scenarios in RL.

**Model-free Estimators:** In model-free estimators, e.g., IPS estimators, we have many challenges, including, high variance and heavy-tailed scenarios. Recently, many model-free estimators have been proposed for high variance problems in model-free estimators (Strehl et al., 2010; Ionides, 2008b; Swaminathan & Joachims, 2015b; Aouali et al., 2023; Metelli et al., 2021; Neu, 2015; Aouali et al., 2023; Metelli et al., 2021; Sakhi et al., 2024). However, under heavy-tailed or unbounded reward scenario, the performance of these estimators degrade. In this work, our proposed LSE estimator demonstrates robust performance even under heavy-tailed assumptions, backed by theoretical guarantees.

**Model-based Estimators:** The direct method for off-policy learning from the LBF datasets is based on the estimation of the reward function, followed by the application of a supervised learning algorithm to the problem. However, this approach does not generalize well, as shown by Beygelzimer & Langford (2009). A different approach where the direct method and the IPS estimator are combined, i.e., doubly-robust, is introduced by Dudík et al. (2014). A different approach based on policy optimization and boosted base learner is proposed to improve the performance in direct methods (London et al., 2023).Furthermore, the optimistic

shrinkage (Su et al., 2020) and Dr-Switch (Wang et al., 2017) as other model-based estimators. Our approach differs from this area, as we do not estimate the reward function in the LSE estimator. A combination of the LSE estimator with the direct method is presented in App. G.3. In this work, we focus on *model-free approach*.

**Unbounded Reward:** Unbounded rewards (or returns) have been observed in various domains, including finance (Lu & Rong, 2018) and robotics (Bohez et al., 2019). In the context of multi-arm bandit problems, unbounded rewards can emerge as a result of adversarial attacks on reward distributions (Guan et al., 2020). Within the broader field of RL, researchers have investigated poisoning attacks on rewards and the manipulation of observed rewards (Rakhsha et al., 2020; 2021; Rangi et al., 2022). These studies highlight the importance of considering unbounded reward scenarios in RL and bandits algorithms. In particular, in our work, we focus on off-policy learning and evaluation under heavy-tailed (unbounded reward) assumption, employing a bounded $(1 + \epsilon)$-th moment of weighted-reward assumption for $\epsilon \in [0, 1]$.

## 4. Problem Formulation

Let $\mathcal{X}$ be the set of contexts and $\mathcal{A}$ the set of actions. We consider policies as conditional distributions over actions, given contexts. For each pair of context and action $(x, a) \in \mathcal{X} \times \mathcal{A}$ and policy $\pi_\theta \in \Pi_\theta$, where $\Pi_\Theta$ is defined as the set of all policies (policy set) which are parameterized by $\theta \in \Theta$, where $\Theta$ is the set of parameters, e.g., the parameters of a neural network. Furthermore, the $\pi_\theta(a|x)$ is defined as the conditional probability of choosing an action given context $x$ under the policy $\pi_\theta$.[1]

A reward function [2] $r : \mathcal{X} \times \mathcal{A} \rightarrow \mathbb{R}^+$, which is unknown, defines the expected reward (feedback) of each observed pair of context and action. In particular, $r(x, a) = \mathbb{E}_{P_{R|X=x, A=a}}[R]$ where $R \in \mathbb{R}^+$ is random reward and $P_{R|X=x, A=a}$ is the conditional distribution of reward $R$ given the pair of context and action, $(x, a)$. Note that, in the LBF setting, we only observe the reward (feedback) for the chosen action $a$ in a given context $x$, under the known logging policy $\pi_0(a|x)$. We have access to the LBF dataset $S = (x_i, a_i, p_i, r_i)_{i=1}^n$ with $n$ i.i.d. data points where each

---

[1]In more details, consider an action space $\mathcal{A}$ with a $\sigma$-algebra and a $\sigma$-finite measure $\mu$. For any policy $\pi$ and context $x$, let $\pi(.|x)$ be a probability measure on $\mathcal{A}$ that is absolutely continuous with respect to $\mu$, with density $\pi(.|x) = \frac{d\pi_c(a|x)}{d\mu}$ where $\pi_c(a|x)$ is absolute continuous with respect to $\mu$.

[2]The reward can be viewed as the opposite (negative) of the cost. Consequently, a low cost (equivalent to maximum reward) signifies user (context) satisfaction with the given action, and conversely. For the cost function, we have $c(x, a) = -r(x, a)$ as discussed in (Swaminathan & Joachims, 2015a).

'data point' $(x_i, a_i, p_i, r_i)$ contains the context $x_i$ which is sampled from unknown distribution $P_X$, the action $a_i$ which is sampled from the known logging policy $\pi_0(\cdot|x_i)$, the propensity score $p_i \triangleq \pi_0(a_i|x_i)$, and the observed feedback (reward) $r_i$ as a sample from distribution $P_{R|X=x_i, A=a_i}$ under logging policy $\pi_0(a_i|x_i)$.

We define the expected reward of a learning policy, $\pi_\theta \in \Pi_\theta$, which is called the *value function* evaluated at the learning policy, as

$$V(\pi_\theta) = \mathbb{E}_{P_X}[\mathbb{E}_{\pi_\theta(A|X)}[\mathbb{E}_{P_{R|X,A}}[R]]]$$
$$= \mathbb{E}_{P_X}[\mathbb{E}_{\pi_\theta(A|X)}[r(A, X)|X]]. \quad (2)$$

We denote the importance weighted reward as $w_\theta(A, X)R$, where $w_\theta(A, X)$ is the weight,

$$w_\theta(A, X) = \frac{\pi_\theta(A|X)}{\pi_0(A|X)}.$$

As discussed by Swaminathan & Joachims (2015b), the IPS estimator is applied over the LBF dataset $S$ (Rosenbaum & Rubin, 1983) to get an unbiased estimator of the value function by considering the weighted reward as,

$$\widehat{V}(\pi_\theta, S) = \frac{1}{n}\sum_{i=1}^{n} r_i w_\theta(a_i, x_i), \quad (3)$$

where $w_\theta(a_i, x_i) = \frac{\pi_\theta(a_i|x_i)}{\pi_0(a_i|x_i)}$.

The IPS estimator as an unbiased estimator has bounded variance if the $\pi_\theta(A|X)$ is absolutely continuous with respect to $\pi_0(A|X)$ (Strehl et al., 2010; Langford et al., 2008). Otherwise, it suffers from a large variance.

**LSE in OPE and OPL scenarios:** The LSE estimator is defined as

$$\widehat{V}_{\text{LSE}}^\lambda(S, \pi_\theta) := \text{LSE}_\lambda(S) = \frac{1}{\lambda}\log\left(\frac{1}{n}\sum_{i=1}^{n} e^{\lambda r_i w_\theta(a_i, x_i)}\right),$$

where $\lambda < 0$ is a tunable parameter which helps us to recover the IPS estimator for $\lambda \to 0$. Furthermore, the LSE estimator is an increasing function with respect to $\lambda$.

*OPE scenario:* One of the evaluation metrics for an estimator in OPE scenarios is the MSE which is decomposed into squared bias and the variance of the estimator. In particular, for the LSE estimator, we consider the following MSE decomposition in terms of bias and variance,

$$\text{MSE}(\widehat{V}_{\text{LSE}}^\lambda(S, \pi_\theta)) = \mathbb{B}(\widehat{V}_{\text{LSE}}^\lambda(S, \pi_\theta))^2 + \mathbb{V}(\widehat{V}_{\text{LSE}}^\lambda(S, \pi_\theta)),$$

$$\mathbb{B}(\widehat{V}_{\text{LSE}}^\lambda(S, \pi_\theta)) = \mathbb{E}[w_\theta(A, X)R] - \mathbb{E}[\widehat{V}_{\text{LSE}}^\lambda(S, \pi_\theta)],$$

$$\mathbb{V}(\widehat{V}_{\text{LSE}}^\lambda(S, \pi_\theta)) = \mathbb{E}[(\widehat{V}_{\text{LSE}}^\lambda(S, \pi_\theta) - \mathbb{E}[\widehat{V}_{\text{LSE}}^\lambda(S, \pi_\theta)])^2],$$

where $\mathbb{B}(\widehat{V}_{\text{LSE}}^\lambda(S, \pi_\theta))$ and $\mathbb{V}(\widehat{V}_{\text{LSE}}^\lambda(S, \pi_\theta))$ are bias and variance of the LSE estimator, respectively.

*OPL scenario:* Our objective in OPL scenario is to find an optimal $\pi_{\theta^\star}$, one which maximize $V(\pi_\theta)$, i.e.,

$$\pi_{\theta^\star} = \arg\max_{\pi_\theta \in \Pi_\Theta} V(\pi_\theta). \quad (4)$$

We define the *estimation error*[3], as the difference between the value function and the LSE estimator for a given learning policy $\pi_\theta \in \Pi_\theta$, i.e.,

$$\text{Est}_\lambda(\pi_\theta) := V(\pi_\theta) - \widehat{V}_{\text{LSE}}^\lambda(S, \pi_\theta). \quad (5)$$

For the OPL scenario, we also define $\pi_{\widehat{\theta}}$ policy as the maximizer of the LSE estimator for a given dataset $S$,

$$\pi_{\widehat{\theta}}(S) = \arg\max_{\pi_\theta \in \Pi_\Theta} \widehat{V}_{\text{LSE}}^\lambda(S, \pi_\theta). \quad (6)$$

Finally, we define *regret*, as the difference between the value function evaluated at $\pi_{\theta^*}$ and $\pi_{\widehat{\theta}}$,

$$\mathfrak{R}_\lambda(\pi_{\widehat{\theta}}, S) := V(\pi_{\theta^*}) - V(\pi_{\widehat{\theta}}(S)). \quad (7)$$

More discussion regarding the LSE properties is provided in App. C.

## 5. Theoretical Foundations of the LSE Estimator

In this section, we study the regret, bias-variance and robustness of the LSE estimator. We compare our LSE estimator with other model-free estimators in Table 2. All the proof details are deferred to App.D.

**Non-linearity of LSE:** The LSE estimator is a non-linear model-free estimator with respect to the weighted reward or reward, which is different from linear model-free estimators. In particular, most estimators can be represented as the weighted average of reward (feedback),

$$\widehat{V}(\pi_\theta, S) = \frac{1}{n}\sum_{i=1}^{n} g(r_i, w_\theta(a_i, x_i)), \quad (8)$$

where $g : \mathbb{R}^+ \times \mathbb{R}^+ \to \mathbb{R}$ is a transformation of $r_i w_\theta(a_i, x_i)$ and is defined for each model-free estimator. For example, we have $g(r, y) = ry$ in the IPS estimator, $g(r, y) = r\min(y, M)$ in the truncated IPS estimator (Ionides, 2008b), $g(r, y) = r((1 - \widehat{\lambda})y^s + \widehat{\lambda})^{1/s}$ in the PM estimator (Metelli et al., 2021), $g(r, y) = ry^\alpha$ for $\alpha \in (0, 1)$ in the ES estimator (Aouali et al., 2023) and $g(r, y) = r\frac{\tau y}{y^2 + \tau}$ in the optimistic shrinkage (OS) (Su et al., 2020). For the IX-estimator with parameter $\eta$ (Gabbianelli et al., 2023), we

---

[3]In statistical learning theory, the difference between the expected value of a random variable and its empirical estimate is referred to as the estimation error. When applied to learning algorithms, this gap—between expected and empirical performance—is known as the generalization gap.

have $g(r, y) = r\frac{y}{1+\eta/\pi_0}$. Furthermore, recently a logarithmic smoothing (LS) estimator with parameter $\tilde{\lambda}$ is proposed by Sakhi et al. (2024) where $g(r, y) = \frac{1}{\tilde{\lambda}}\log(1 + \tilde{\lambda}ry)$. However, the LSE estimator is a non-linear function with respect to a whole set of weighted reward samples. Therefore, the previous techniques for regret and bias-variance analysis under linear estimators are not applicable.

**Theoretical comparison with other estimators:** The comparison of our LSE estimator with other estimators, including, IPS, self-normalized IPS (Swaminathan & Joachims, 2015b), truncated IPS with weight truncation parameter $M$, ES-estimator with parameter $\alpha$ (Aouali et al., 2023), IX-estimator with parameter $\eta$, PM-estimator with parameter $\lambda$ (Metelli et al., 2021), OS-estimator with parameter $\tau$ (Su et al., 2020) and LS-estimator with parameter $\tilde{\lambda}$ (Sakhi et al., 2024) is provided in Table 2.

Note that the truncated IPS (IPS-TR) (Ionides, 2008a) employs truncation, resulting in a non-differentiable estimator. This non-differentiability complicates the optimization phase, often necessitating additional care and sometimes leading to computationally intensive discretizations (Papini et al., 2019). Furthermore, tuning the threshold $M$ in IPS-TR is sensitive (Aouali et al., 2023).

In the following sections, we provide more details regarding heavy-tail assumption and theoretical results for the LSE estimator.

### 5.1. Heavy-tail Assumption

In this section, the following heavy-tail assumption is made in our theoretical results.

**Assumption 5.1** (Heavy-tail weighted reward). The reward distribution $P_{R|X,A}$ and $P_X \otimes \pi_0(A|X)$ are such that for all learning policy $\pi_\theta(A|X) \in \Pi_\theta$ and some $\epsilon \in [0, 1]$, the $(1 + \epsilon)$-th moment of the weighted reward is bounded[4],

$$\mathbb{E}_{P_X \otimes \pi_0(A|X) \otimes P_{R|X,A}}\left[\left(w_\theta(A, X)R\right)^{1+\epsilon}\right] \leq \nu. \quad (9)$$

We make a few remarks. First, in comparison with the bounded reward function assumption in literature, (Metelli et al., 2021; Aouali et al., 2023), in Assumption 5.1, the reward function can be unbounded. Moreover, our assumptions are weaker with respect to the uniform overlap assumption [5]. In heavy-tailed bandit learning (Bubeck et al., 2013; Shao et al., 2018; Lu et al., 2019), a similar assumption to Assumption 5.1 on $(1 + \epsilon)$-th moment of reward for some $\epsilon \in [0, 1]$ is assumed. In contrast, in Assumption 5.1, we consider the weighted reward. Note that, under uniform

coverage (overlap) assumption, Assumption 5.1 can be interpreted as a heavy-tailed assumption on reward. Furthermore under a bounded reward, Assumption 5.1 would be equivalent with the heavy-tailed assumption on the $(1 + \epsilon)$-th moment of weight function, $w_\theta(a, x)$. A more detailed theoretical comparison is provided in App. D.1. We also provide a comparison with other estimators under bounded reward assumption in App. D.1.7 where Assumption 5.1 reduces to heavy-tailed assumption on weights.

### 5.2. Regret Bounds

In this section, we provide an upper bound on the regret of the LSE estimator as discussed in the OPL scenario. The following novel is a helpful lemma to prove some results.

**Lemma 5.2.** *Consider the random variable $Z > 0$. For $\epsilon \in [0, 1]$, then $\mathbb{V}\left(e^{\lambda Z}\right) \leq |\lambda|^{1+\epsilon}\mathbb{E}[Z^{1+\epsilon}]$ holds for $\lambda < 0$.*

In the following Theorem, we provide an upper bound on the regret of learning policy under the LSE estimator.

**Theorem 5.3** (Regret bounds). *For any $\gamma \in (0, 1)$, given Assumption 5.1, assuming finite policy set $|\Pi_\theta| < \infty$ and $n \geq \frac{\left(2|\lambda|^{1+\epsilon}\nu + \frac{4}{3}\gamma\right)\log\frac{|\Pi_\theta|}{\delta}}{\gamma^2 \exp(2\lambda\nu^{1/(1+\epsilon)})}$ for $\lambda < 0$, with probability at least $(1 - \delta)$, the following upper bound holds on the regret of the LSE estimator,*

$$0 \leq \mathfrak{R}_\lambda(\pi_{\widehat{\theta}}, S) \leq \frac{|\lambda|^\epsilon}{1+\epsilon}\nu - \frac{4(2-\gamma)}{3(1-\gamma)}\frac{\log\frac{4|\Pi_\theta|}{\delta}}{n\lambda\exp(\lambda\nu^{1/(1+\epsilon)})}$$
$$- \frac{(2-\gamma)}{(1-\gamma)\lambda}\sqrt{\frac{4|\lambda|^{1+\epsilon}\nu\log\frac{4|\Pi_\theta|}{\delta}}{n\exp(2\lambda\nu^{1/(1+\epsilon)})}},$$

*where $\pi_{\widehat{\theta}}$ is defined in equation 6.*

***Sketch of Proof:*** Using Bernstein's inequality, Boucheron et al., 2013 and Lemma 5.2, we provide lower and upper bounds on estimation error for a fixed learning policy $\pi_\theta$. Then, we consider the following decomposition of regret,

$$V(\pi_{\theta^*}) - V(\pi_{\widehat{\theta}}) = \text{Est}_\lambda(\pi_{\theta^*}) \quad (10)$$
$$+ \widehat{V}_{\text{LSE}}^\lambda(S, \pi_{\theta^*}) - \widehat{V}_{\text{LSE}}^\lambda(S, \pi_{\widehat{\theta}}) - \text{Est}_\lambda(\pi_{\widehat{\theta}}).$$

Note that, the second is negative. We can provide upper and lower bounds on estimation error (Theorem D.2 and Theorem D.3 in App. D.2), respectively. $\square$

As the regret bound in Theorem 5.3 depends on $\lambda$, we need to select an appropriate $\lambda$ to study the convergence rate of regret bound with respect to $n$.

**Proposition 5.4** (Convergence rate). *Given Assumption 5.1, for any $0 < \gamma < 1$, assuming $n \geq \frac{\left(2\nu + \frac{4}{3}\gamma\right)\log\frac{|\Pi_\theta|}{\delta}}{\gamma^2 \exp(2\nu^{1/(1+\epsilon)})}$ and setting $\lambda = -n^{-\frac{1}{1+\epsilon}}$, then the overall convergence rate of the regret upper bound is $O(n^{-\epsilon/(1+\epsilon)})$.*

---

[4]We assume unbounded $(1+\kappa)$-th moment of weighted reward for $\kappa > \epsilon$.

[5]In the uniform coverage (overlap) assumption, it is assumed that $\sup_{(a,x)\in\mathcal{A}\times\mathcal{X}} w_\theta(a, x) = U_c < \infty$.

Table 2: Comparison of estimators. We consider the bounded reward function, i.e., $R_{\max} := \sup_{(a,x) \in \mathcal{A} \times \mathcal{X}} r(a, x)$ for all estimators except LSE. $\mathbb{B}^{\text{SN}}$ and $\mathbb{V}^{\text{SN}}$ are the Bias and the Efron-Stein estimate of the variance of self-normalized IPS. For the ES-estimator, we have $T^{ES} = \mathbb{B}^{ES} + (1/n)\big(D_{\text{KL}}(\pi_\theta \| \pi_0) + \log(4/\delta)\big)$. where $D_{\text{KL}}(\pi_\theta \| \pi_0) = \int_{\mathcal{A}} \pi_\theta(a|x) \log(\pi_\theta(a|x)/\pi_0(a|x)) \mathrm{d}a$. We also define power divergence as $P_\alpha(\pi_\theta \| \pi_0) := \int_{\mathcal{A}} \pi_\theta(a|x)^\alpha \pi_0(a|x)^{(1-\alpha)} \mathrm{d}a$ is the power divergence with order $\alpha$. For the IX-estimator, $C_\eta(\pi)$ is the smoothed policy coverage ratio. We compare the convergence rate of the concentration (or regret bound) for estimators. $B$ and $C$ are constants. For LS estimator, $\mathcal{S}_{\tilde{\lambda}}(\pi_\theta)$ is the discrepancy between $\pi$ and $\pi_0$.

| Estimator | Concentration | Convergence Rate | Heavy-tailed | Regret Bound | Noisy Reward | Differentiability | Subgaussian Like Tail |
|---|---|---|---|---|---|---|---|
| IPS | $R_{\max}^2 \sqrt{\frac{P_2(\pi_\theta \| \pi_0)}{\delta n}}$ | $O(n^{-1/2})$ | $\times$ | $\checkmark$ | $\times$ | $\checkmark$ | $\times$ |
| SN-IPS (Swaminathan & Joachims, 2015b) | $R_{\max}(B^{\text{SN}} + \sqrt{V^{\text{ES}} \log \frac{1}{\delta}})$ | - | $\times$ | $\times$ | $\times$ | $\checkmark$ | $\times$ |
| IPS-TR ($M > 0$) (Ionides, 2008a) | $R_{\max} \sqrt{\frac{P_2(\pi_\theta \| \pi_0) \log \frac{1}{\delta}}{n}}$ | $O(n^{-1/2})$ | $\times$ | $\checkmark$ | $\times$ | $\times$ | $\checkmark$ |
| IX ($\eta > 0$) (Gabbianelli et al., 2023) | $R_{\max}(2\eta C_\eta(\pi_\theta) + \frac{\log(2/\delta)}{\eta n})$ | $O(n^{-1/2})$ | $\times$ | $\checkmark$ | $\times$ | $\checkmark$ | $\checkmark$ |
| PM ($\hat{\lambda} \in [0,1]$) (Metelli et al., 2021) | $R_{\max} \sqrt{\frac{P_2(\pi_\theta \| \pi_0) \log \frac{1}{\delta}}{n}}$ | $O(n^{-1/2})$ | $\times$ | $\times$ | $\times$ | $\checkmark$ | $\checkmark$ |
| ES ($\alpha \in [0,1]$) (Aouali et al., 2023) | $R_{\max} \sqrt{\frac{D_{\text{KL}}(\pi_\theta \| \pi_0) + \log(4\sqrt{n}/\delta)}{n}} + T^{ES}$ | $O\big((\log(n)/n)^{1/2}\big)$ | $\times$ | $\checkmark$ | $\times$ | $\checkmark$ | $\times$ |
| OS ($\tau > 0$) (Su et al., 2020) | $\max_{\beta \in \{2,3\}} \sqrt[\beta]{\frac{P_\beta(\pi_\theta \| \pi_0)(\log \frac{1}{\delta})^{\beta-1}}{n^{\beta-1}}}$ | $O\big(n^{(1-\beta)/\beta}\big)$ | $\times$ | $\times$ | $\times$ | $\checkmark$ | $\times$ |
| LS ($\bar{\lambda} \geq 0$) (Sakhi et al., 2024) | $\bar{\lambda} \mathcal{S}_{\tilde{\lambda}}(\pi_\theta) + \frac{\log(2/\delta)}{\bar{\lambda} n}$ | $O(n^{-1/2})$ | $\times$ | $\checkmark$ | $\times$ | $\checkmark$ | $\checkmark$ |
| **LSE** ($0 > \lambda > -\infty$ and $\epsilon \in [0,1]$) **(ours)** | $C\big(\frac{2\log(2|\Pi_\theta|/\delta)}{n}\big)^{\epsilon/(1+\epsilon)}$ | $O(n^{-\epsilon/(1+\epsilon)})$ | $\checkmark$ | $\checkmark$ | $\checkmark$ | $\checkmark$ | $\checkmark$ |

**Discussion:** Note that, if Assumption 5.1 holds for $\epsilon = 1$ where the second moment of weighted reward is bounded, then we have the convergence rate of $O(n^{-1/2})$. Moreover, if higher moments of the weighted reward are bounded, the second moment is also bounded, allowing our results for a bounded second moment to apply. Note that, our theoretical results on regret can be applied to unbounded weighted reward under Assumption 5.1 and other estimators can not guarantee the convergence rate of $O(n^{-\epsilon/(1+\epsilon)})$ under bounded $(1 + \epsilon)$-th moment of weighted reward.

**Bounded reward:** Our results in Theorem 5.3 also holds under bounded reward and heavy-tailed weights assumption. More discussion is provided in App. D.1.7 .

**Finite policy set:** The results in this section assumed that the policy set, $\Pi_\theta$, is finite; this is for example the case in off-policy learning problems with a finite number of policies. If this assumption is violated, we can apply the growth function technique which is bounded by VC-dimension (Vapnik, 2013) or Natarajan dimension (Holden & Niranjan, 1995) as discussed in (Jin et al., 2021). Furthermore, we can apply PAC-Bayesian analysis (Gabbianelli et al., 2023) for the LSE estimator. More discussion regarding the PAC-Bayesian approach is provided in App. D.6.

**Subgaussian Concentration:** We also study achieving subgaussian concentration for the LSE estimator where the dependency of regret on $\delta$ is subgaussian $O\left(\sqrt{\frac{\log(1/\delta)}{n}}\right)$, in App. D.7.

### 5.3. Bias and Variance

In this section, we provide an analysis of bias and variance for the LSE estimator.

**Proposition 5.5** (Bias bound). *Given Assumption 5.1, the following lower and upper bounds hold on the bias of the LSE estimator with $\lambda < 0$,*

$$\frac{(n-1)}{2n|\lambda|} \mathbb{V}(e^{\lambda w_\theta(A,X)R}) \leq \mathbb{B}(\widehat{\mathbb{V}}_{\text{LSE}}^\lambda(S, \pi_\theta)) \qquad (11)$$

$$\leq \frac{1}{1+\epsilon} |\lambda|^\epsilon \nu + \frac{1}{2n\lambda} \mathbb{V}(e^{\lambda w_\theta(A,X)R}).$$

*Remark* 5.6 (Asymptotically Unbiased). By selecting $\lambda$ as a function of $n$, which tends to zero as $n \to \infty$, e.g. $\lambda(n) = -n^{-\varsigma}$ for some $\varsigma > 0$, the bounds in Proposition 5.5 becomes asymptotically zero. The overall convergence rate for upper bound is $O(n^{-\epsilon/(1+\epsilon)})$ by choosing $\varsigma = \frac{1}{1+\epsilon}$. For example, if Assumption 5.1 holds for $\epsilon = 1$, then by choosing $\varsigma = 1/2$, we have the convergence rate of $O(n^{-1/2})$ for the bias of the LSE estimator. Consequently, the LSE estimator is asymptotically unbiased.

For the variance of the LSE estimator, we provide the following upper bound.

**Proposition 5.7** (Variance Bound). *Assume that $\mathbb{E}[(w_\theta(A, X)R)^2] \leq \nu_2$ [6] holds. Then the variance*

---

[6]Assumption 5.1 for $\epsilon = 1$.

*of the LSE estimator with $\lambda < 0$, satisfies,*

$$\mathbb{V}(\widehat{V}_{\text{LSE}}^{\lambda}(S, \pi_\theta)) \leq \frac{1}{n}\mathbb{V}(w_\theta(A, X)R) \leq \frac{1}{n}\nu_2. \quad (12)$$

**Variance Reduction:** We can observe that the variance of the LSE is less than the variance of the IPS estimator for all $\lambda < 0$.

Combining the results in Proposition 5.5 and Proposition 5.7, we can derive an upper bound on MSE of the LSE estimator using MSE. The bias and variance trade-off for the LSE estimator and the comparison of different estimators in terms of bias and variance are provided in App. D.1.1.

### 5.4. Robustness of the LSE Estimator: Noisy Reward

In this section section, we study the robustness of the LSE estimator under noisy reward. We also investigate the performance of the LSE estimator under noisy (estimated) propensity scores in the App. E. To analyze the robustness of the LSE estimator, we extend the approach of tilted empirical risk introduced by Aminian et al. (2025), which provides generalization error bounds[7] under distributional shifts in supervised learning scenario under tilted empirical risk. Our analysis leverages these tools to quantify the robustness of the LSE to noisy rewards.

Suppose that due to an outlier or noise in receiving the feedback (reward), the underlying distribution of the reward given a pair of actions and contexts, $P_{R|X,A}$ is shifted via the distribution of noise or outlier, denoted as $\widetilde{P}_{R|X,A}$. We model the distributional shift of reward via distribution $\widetilde{P}_{R|X,A}$ due to inspiration by the notion of influence function (Marceau & Rioux, 2001; Christmann & Steinwart, 2004). Furthermore, we define the noisy reward LBF dataset as $\widetilde{S}$ with $n$ data samples. For our result in this section, the following assumption is made.

**Assumption 5.8** (Heavy-tailed Weighted Noisy Reward). The $P_X \otimes \pi_0(A|X)$ and noisy reward distribution $\widetilde{P}_{R|X,A}$ are such that for all learning policy $\pi_\theta(A|X) \in \Pi_\theta$ and some $\epsilon \in [0,1]$, the $(1+\epsilon)$-th moment of the weighted reward is bounded,

$$\mathbb{E}_{P_X \otimes \pi_0(A|X) \otimes \widetilde{P}_{R|X,A}}\left[\left(w_\theta(A,X)R\right)^{1+\epsilon}\right] \leq \widetilde{\nu}. \quad (13)$$

Under the noisy reward LBF dataset, we derive the following learning policy,

$$\pi_{\widehat{\theta}}(\widetilde{S}) = \arg\max_{\pi_\theta \Pi_\theta} \widehat{V}_{\text{LSE}}^{\lambda}(\pi_\theta, \widetilde{S}). \quad (14)$$

In the following theorem, we provide an upper bound on the regret of $\pi_{\widehat{\theta}}(\widetilde{S})$ as the learning policy under the noisy reward LBF dataset.

[7]Generalization error is defined as difference between population and empirical risks in supervised learning scenario.

**Theorem 5.9.** *For any $\gamma \in (0,1)$, given Assumption 5.1, Assumption 5.8 and assuming $n \geq \frac{\left(2|\lambda|^{1+\epsilon}\nu + \frac{4}{3}\gamma\right)\log\frac{|\Pi_\theta|}{\delta}}{\gamma^2 \exp(2\lambda\nu^{1/(1+\epsilon)})}$ for $\lambda < 0$, with probability at least $(1-\delta)$, the following upper bound holds on the regret of the LSE estimator under noisy reward logged data,*

$$0 \leq \mathfrak{R}_\lambda(\pi_{\widehat{\theta}}(\widetilde{S}), \widetilde{S}) \leq \frac{|\lambda|^\epsilon}{1+\epsilon}\nu + 2A(\gamma)\sqrt{|\lambda|^\epsilon\widetilde{\nu}C_1(\lambda, n)}$$

$$+ 2A(\gamma)C_1(\lambda, n) + \frac{2\mathbb{TV}(P_{R|X,A}, \widetilde{P}_{R|X,A})}{\lambda^2}D(\tilde{\nu}, \nu),$$

*where $A(\gamma) = \frac{(2-\gamma)}{(1-\gamma)}$, $C_1(\lambda, n) = \frac{\log\frac{4|\Pi_\theta|}{\delta}}{n|\lambda|\exp(\lambda\widetilde{\nu}^{1/(1+\epsilon)})}$, $D(\tilde{\nu}, \nu) = \frac{\exp(|\lambda|\tilde{\nu}^{1/(1+\epsilon)}) - \exp(|\lambda|\nu^{1/(1+\epsilon)})}{\tilde{\nu}^{1/(1+\epsilon)} - \nu^{1/(1+\epsilon)}}$, $\pi_{\widehat{\theta}}(\widetilde{S})$ is defined in equation 14 and $\mathbb{TV}_c(P_{R|X,A}, \widetilde{P}_{R|X,A}) = \mathbb{E}_{P_X \otimes \pi_0(A|X)}[\mathbb{TV}(P_{R|X,A}, \widetilde{P}_{R|X,A})]$*

**Data-driven $\lambda$:** For large number of samples, $n \to \infty$, the second and third terms in Theorem 5.9 become negligible. Thus, under a noisy reward setting, $\lambda$ can be chosen using the following objective function.

$$\lambda_{\text{ND}} := \arg\min_{\lambda \in (-\infty, 0)} \frac{|\lambda|^\epsilon}{1+\epsilon}\nu + \frac{2\mathbb{TV}_c(P_{R|X,A}, \widetilde{P}_{R|X,A})}{\lambda^2}D(\tilde{\nu}, \nu),$$

where $D(\tilde{\nu}, \nu)$ is defined in Theorem 5.9 .

**Robustness:** The term $\frac{\mathbb{TV}_c(P_{R|X,A}, \widetilde{P}_{R|X,A})}{\lambda^2}$ in the upper bound on the LSE regret under noisy reward scenario (Theorem 5.9) can be interpreted as the cost of noise associated with noisy reward. This cost can be reduced by increasing $|\lambda|$. However, increasing $|\lambda|$ also amplifies the term $\frac{|\lambda|^\epsilon}{1+\epsilon}\nu$ in the upper bound on regret. Therefore, there is a trade-off between robustness and regret, particularly for $\lambda < 0$ in the LSE estimator. More discussion regarding the robustness of the LSE is provided in App. D.9 .

## 6. Experiments

We present our experiments for OPE and OPL. Our aim is to demonstrate that our proposed estimators not only possess desirable theoretical properties but also compete with baseline estimators in practical scenarios. More details can be found in App.F.

### 6.1. Off-policy Evaluation

**Baselines:** For our experiments in OPE setting, we consider truncated IPS estimator (Swaminathan & Joachims, 2015a), PM estimator (Metelli et al., 2021), ES estimator (Aouali et al., 2023), IX estimator (Gabbianelli et al., 2023), SNIPS (Swaminathan & Joachims, 2015b), LS-LIN and LS estimators (Sakhi et al., 2024), and OS (shrinkage) (Su et al., 2020) estimator as baselines.

Table 3: Comparison of different estimators LSE, PM, ES, IX, BanditNet, LS-LIN and OS accuracy for EMNIST with different qualities of logging policy ($\tau \in \{1, 10\}$) and true/noisy (estimated) propensity scores with $b \in \{5, 0.01\}$ and noisy reward with $P_f \in \{0.1, 0.5\}$. The best-performing result is highlighted in **bold** text, while the second-best result is colored in red for each scenario.

| Dataset | $\tau$ | $b$ | $P_f$ | LSE | PM | ES | IX | BanditNet | LS-LIN | OS | Logging Policy |
|---|---|---|---|---|---|---|---|---|---|---|---|
| EMNIST | 1 | — | — | $88.49_{\pm0.04}$ | $\mathbf{89.19_{\pm0.03}}$ | $88.61_{\pm0.06}$ | $88.33_{\pm0.13}$ | $66.58_{\pm6.39}$ | $88.70_{\pm0.02}$ | $88.71_{\pm0.26}$ | $88.08$ |
| | | 5 | — | $\mathbf{89.16_{\pm0.03}}$ | $88.94_{\pm0.05}$ | $88.48_{\pm0.03}$ | $88.51_{\pm0.23}$ | $65.10_{\pm0.69}$ | $88.38_{\pm0.18}$ | $88.70_{\pm0.15}$ | $88.08$ |
| | | 0.0 | — | $\mathbf{86.07_{\pm0.01}}$ | $85.62_{\pm0.10}$ | $85.71_{\pm0.04}$ | $81.39_{\pm4.02}$ | $66.55_{\pm3.11}$ | $84.64_{\pm0.17}$ | $84.59_{\pm0.09}$ | $88.08$ |
| | | — | 0.1 | $\mathbf{89.29_{\pm0.04}}$ | $89.08_{\pm0.05}$ | $88.45_{\pm0.09}$ | $88.14_{\pm0.14}$ | $59.90_{\pm3.78}$ | $88.30_{\pm0.12}$ | $88.74_{\pm0.09}$ | $88.08$ |
| | | — | 0.5 | $88.72_{\pm0.08}$ | $\mathbf{88.78_{\pm0.03}}$ | $87.27_{\pm0.10}$ | $87.08_{\pm0.14}$ | $56.95_{\pm3.06}$ | $87.20_{\pm0.32}$ | $88.06_{\pm0.09}$ | $88.08$ |
| | 10 | — | — | $88.59_{\pm0.03}$ | $\mathbf{88.61_{\pm0.04}}$ | $88.38_{\pm0.08}$ | $87.43_{\pm0.19}$ | $85.48_{\pm3.13}$ | $88.58_{\pm0.08}$ | $86.88_{\pm0.34}$ | $79.43$ |
| | | 5 | — | $88.42_{\pm0.07}$ | $\mathbf{88.43_{\pm0.07}}$ | $88.39_{\pm0.10}$ | $88.39_{\pm0.06}$ | $84.90_{\pm3.10}$ | $88.23_{\pm0.27}$ | $86.00_{\pm0.37}$ | $79.43$ |
| | | 0.01 | — | $\mathbf{82.15_{\pm0.21}}$ | $80.85_{\pm0.29}$ | $81.07_{\pm0.07}$ | $77.49_{\pm2.77}$ | $27.02_{\pm1.92}$ | $78.43_{\pm3.13}$ | $21.70_{\pm4.11}$ | $79.43$ |
| | | — | 0.1 | $\mathbf{88.29_{\pm0.06}}$ | $88.22_{\pm0.02}$ | $88.19_{\pm0.08}$ | $87.93_{\pm0.35}$ | $84.89_{\pm3.21}$ | $87.50_{\pm0.17}$ | $87.68_{\pm0.16}$ | $79.43$ |
| | | — | 0.5 | $88.71_{\pm0.16}$ | $88.52_{\pm0.07}$ | $84.42_{\pm0.34}$ | $83.25_{\pm3.45}$ | $63.35_{\pm13.39}$ | $85.75_{\pm0.04}$ | $\mathbf{89.09_{\pm0.05}}$ | $79.43$ |

**Datasets:** We conduct synthetic experiments to evaluate our proposed LSE estimator performance in the OPE setting. For this purpose, we consider an LBF dataset which has only a single context (state), denoted as $x_0$. We consider the learning and logging policies as Gaussian distributions, $\pi_\theta(\cdot|x_0) \sim \mathcal{N}(\mu_1, \sigma^2)$ and $\pi_0(\cdot|x_0) \sim \mathcal{N}(\mu_2, \sigma^2)$. The reward function is a positive exponential function $e^{\alpha x^2}$ which is unbounded. We also set our parameters to observe different tail distributions. We fix $\mu_1 = 0.5, \mu_2 = 1, \sigma^2 = 0.25$ and change the value of $\alpha$ which controls the tail of the weighted reward variable, $\alpha \in \{1.4, 1.6\}$. We also examine different values of $\alpha$ and the effect of a number of samples for a fixed $\alpha$ in App. G.1. Moreover, we conduct a similar experiment when logging and learning policies are Lomax distributions[8] in App. G.1.

**Metrics:** We calculate the bias, variance, and MSE of estimators by running the experiments for $10K$ times each one over 1000 samples.

**Discussion:** The results presented in Table 4 demonstrate that the LSE estimator has better performance in terms of both MSE and variance when compared to other baselines. We also conducted experiments for OPE under some UCI datasets in App G.12 . More experiments are provided in App. G.11 for comparison of LSE estimator with LS estimator .

### 6.2. Off-policy Learning

**Baselines:** For our experiments in OPL, we compare our LSE estimator against several non-regularized baseline estimators, including, truncated IPS (Swaminathan & Joachims, 2015a), PM (Metelli et al., 2021), ES (Aouali et al., 2023), IX (Gabbianelli et al., 2023), BanditNet (Joachims et al., 2018), LS-LIN (Sakhi et al., 2024) and OS estimator (Su

Table 4: Bias, variance, and MSE of LSE, ES, PM, IX, and IPS-TR estimators. The experiment is run 10000 times with 1000 samples. The variance, bias, and MSE of the estimations are reported. The best-performing result is highlighted in **bold** text, while the second-best result is colored in red for each scenario.

| Estimator | $\alpha = 1.1$ | | | $\alpha = 1.4$ | | |
|---|---|---|---|---|---|---|
| | Bias | Variance | MSE | Bias | Variance | MSE |
| PM | $0.004$ | $0.063$ | $0.063$ | $-0.301$ | $164.951$ | $165.041$ |
| ES | $-0.001$ | $0.054$ | $0.054$ | $1.959$ | $0.396$ | $4.232$ |
| LSE | $0.052$ | $0.006$ | $\mathbf{0.009}$ | $0.615$ | $0.292$ | $\mathbf{0.670}$ |
| IPS-TR | $0.020$ | $0.052$ | $0.052$ | $0.053$ | $133.688$ | $133.691$ |
| IX | $0.237$ | $0.002$ | $0.058$ | $1.340$ | $0.048$ | $1.842$ |
| SNIPS | $-0.005$ | $0.059$ | $0.059$ | $-0.029$ | $133.520$ | $133.521$ |
| LS-LIN | $0.284$ | $0.001$ | $0.082$ | $2.164$ | $0.005$ | $4.687$ |
| LS | $0.082$ | $0.007$ | $0.013$ | $0.564$ | $0.458$ | $0.776$ |
| OS | $0.521$ | $0.020$ | $0.292$ | $0.623$ | $23.589$ | $23.977$ |

et al., 2020).

**Datasets:** In off-policy learning scenario, we apply the standard supervised to bandit transformation (Beygelzimer & Langford, 2009) on a classification dataset: Extended-MNIST (EMNIST) (Xiao et al., 2017) to generate the LBF dataset. We also run on FMNIST in App.G.2 . This transformation assumes that each of the classes in the datasets corresponds to an action. Then, a logging policy stochastically selects an action for every sample in the dataset. For each data sample $x$, action $a$ is sampled by logging policy. For the selected action, propensity score $p$ is determined by the softmax value of that action. If the selected action matches the actual label assigned to the sample, then we have $r = 1$, and $r = 0$ otherwise. So, the 4-tuple $(x, a, p, r)$ makes up the LBF dataset.

**Noisy (Estimated) propensity score:** For noisy propensity score, motivated by Halliwell (2018) and the discussion in

---

[8]The Lomax distribution is a Pareto Type II distribution which is a heavy-tailed distribution.

App.E.1, we assume a multiplicative inverse Gamma noise[9] on $\pi_0$ for $b \in \mathbb{R}^+$, $\widehat{\pi}_0 = \frac{1}{U}\pi_0$, where $\widehat{\pi}(a|x)$ is the estimated propensity scores and $U \sim \text{Gamma}(b,b)$.

**Noisy reward:** Inspired by Metelli et al. (2021), we also consider noise in reward samples. In particular, we model noisy reward by a reward-switching probability $P_f \in [0,1]$ to simulate noise in the reward samples. For example, a reward sample of $r = 1$ may switch to $r = 0$ with probability $P_f$.

**Logging policy:** To have logging policies with different performances, given inverse temperature[10] $\tau \in \{1, 10\}$, first, we train a linear softmax logging policy on the fully-labeled dataset. Then, when we apply standard supervised-to-bandit transformation on the dataset, the results obtained from the linear logging policy which are weights of each action according to the input, will be multiplied by the inverse temperature $\tau$ and then passed to a softmax layer. Thus, as the inverse temperature $\tau$ Increases, we will have more uniform and less accurate logging policies.

**Metric:** We evaluate the performance of the different estimators based on the accuracy of the trained model. Inspired by London & Sandler (2019), we calculate the accuracy for a deterministic policy where the accuracy of the model based on the argmax of the softmax layer output for a given context is computed.

For each value of $\tau$, we apply the LSE estimator and observe the accuracy over three runs on EMNIST. The deterministic accuracies of LSE, PM, ES, IX, BanditNet, OS and LS-LIN for $\tau \in \{1, 10\}$ are presented in Table 3.

**Discussion:** The results presented in Table 3 demonstrate that the LSE estimator achieves maximum accuracy (with less variance) in most scenarios compared to all baselines. Furthermore, an experiment on a real-world dataset, KUAIREC (Gao et al., 2022), is provided in App. G.4 . More discussion and experiments are provided in App. G .

## 7. Conclusion

In this work, inspired by the log-sum-exponential operator, we proposed a novel estimator for off-policy learning and evaluation applications. Subsequently, we conduct a comprehensive theoretical analysis of the LSE estimator, including a study of bias and variance, along with an upper bound on regret under heavy-tailed assumption. Furthermore, we explore the performance of our estimator in sce-

narios involving estimated propensity scores or heavy-tailed weighted rewards. Results from our experimental evaluation demonstrate that our estimator, guided by our theoretical framework, performs competitively compared to most baseline estimators in off-policy learning and evaluation.

## 8. Future Works

In this section, we outline several potential directions for future work based on our LSE estimator.

**LSE and Regularization:** Using regularization for off-policy learning can improve the performance of estimators (Aminian et al., 2024; Metelli et al., 2021). As future works, we plan to study the effect of regularization on the LSE estimator from both theoretical and practical perspectives.

**LSE with Positive $\lambda$:** Inspired by the application of LSE operator in supervised learning for positive ($\lambda > 0$ (Li et al., 2023), exploring the LSE estimator for positive $\lambda$ in scenarios where the logged dataset is imbalance in terms of rewards or actions, can be an interesting direction. Note that our current theoretical analysis can be applied to negative $\lambda$ and is not applicable for positive $\lambda$.

**LSE and RL:** We envision extending the application of our estimator to more challenging reinforcement learning settings, such as those considered by Chen & Jiang (2022); Zanette et al. (2021); Xie et al. (2019a), where the i.i.d. assumption does not necessarily hold. In these scenarios, theoretical guarantees must be adapted to account for dependencies in the data—for example, by extending the analysis to martingale difference sequences.

**LSE and Missing reward:** Note that, in our problem formulation, we assumed that we have access to reward for all logged samples. However, in some applications as discussed in (Aminian et al., 2024), for some data samples the reward (feedback) is missed. In future work, we extend the application of LSE function to these scenarios where the reward (feedback) is partially missed.

## Acknowledgment

The authors thank the anonymous referees and the area chair for their insightful and constructive comments. For the purpose of Open Access, the authors note that a CC BY public copyright license applies to any Author Accepted Manuscript version arising from this submission.

## Impact Statement

This paper presents work whose goal is to advance the field of Machine Learning. There are many potential societal consequences of our work, none of which we feel must be specifically highlighted here.

---

[9]If $Z \sim \text{Gamma}(\alpha, \beta)$, then we have $f_Z(z) = \frac{\beta^\alpha}{\Gamma(\alpha)} z^{\alpha-1} e^{-\beta z}$.

[10]The inverse temperature $\tau$ is defined as $\pi_0(a_i|x) = \frac{\exp(h(x,a_i)/\tau)}{\sum_{j=1}^k \exp(h(x,a_j)/\tau)}$ where $h(x, a_i)$ is the $i$-th input to the softmax layer for context $x \in \mathcal{X}$ and action $a_i \in \mathcal{A}$.

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

# Appendix

## Table of Contents

# A. Other Related Works

In this section, we provide other related works.

**Other Methods for OPL:** A balance-based weighting approach, which outperforms traditional estimators, was proposed by Kallus (2018). Other extensions of batch learning as a scenario for off-policy learning have been studied, Papini et al. (2019) consider samples from different policies and Sugiyama et al. (2007) propose Direct Importance Estimation, which estimates weights directly by sampling from contexts and actions. Chen et al. (2019) introduced a convex surrogate for the regularized value function based on the entropy of the target policy.

**Pessimism Method and Off-policy RL:** The pessimism concept originally, introduced in offline RL(Buckman et al., 2020; Jin et al., 2021), aims to derive an optimal policy within Markov decision processes (MDPs) by utilizing pre-existing datasets (Rashidinejad et al., 2022; 2021; Yin & Wang, 2021; Yan et al., 2023). This concept has also been adapted to contextual bandits, viewed as a specific MDP instance. Recently, a 'design-based' version of the pessimism principle was proposed by Jin et al. (2022) who propose a data-dependent and policy-dependent regularization inspired by a lower confidence bound (LCB) on the estimation uncertainty of the augmented-inverse-propensity-weighted (AIPW)-type estimators which also includes IPS estimators. Our work differs from that of Jin et al. (2022) as our estimator is non-linear estimator. Note that for our theoretical analysis, we consider heavy-tailed assumption for $(1 + \epsilon)$-th moment for some $\epsilon \in [0, 1]$. However, (Jin et al., 2022) also considers 3rd and 4th moments of weights bounded.

**Action Embedding and Clustering:** Due to the extreme bias and variance of IPS and doubly-robust (DR) estimators in large action spaces, Saito & Joachims (2022) proposed using action embeddings to leverage marginalized importance weights and address these issues. Subsequent studies, including (Saito et al., 2023; Peng et al., 2023; Sachdeva et al., 2023), have introduced alternative methods to tackle the challenge of large action spaces. Our work can be integrated with these approaches to further mitigate the effects associated with large action spaces. We consider this combination as future work.

**Individualized Treatment Effects:** The individual treatment effect aims to estimate the expected values of the squared difference between outcomes (reward or feedback) for control and treated contexts (Shalit et al., 2017). In the individual treatment effect scenario, the actions are limited to two actions (treated/not treated) and the propensity scores are unknown (Shalit et al., 2017; Johansson et al., 2016; Alaa & van der Schaar, 2017; Athey et al., 2019; Shi et al., 2019; Kennedy, 2020; Nie & Wager, 2021). Our work differs from this line of works by considering multiple action scenario and assuming the access to propensity scores in the LBF dataset.

**Noisy/Corrupted Rewards:** Agnihotri et al. (2024) utilized offline data with noisy preference feedback as a warm-up step for online bandit learning. In linear bandits, Kveton et al. (2019) estimated a set of pseudo-rewards for each perturbed reward in the history and used it for reward parameter estimation. Lee & Lim (2022) assumes a heavy-tailed noise variable on the observed rewards and proposes two exploration strategies that provide minimax regret guarantees for the multi-arm bandit problem under the heavy-tailed reward noise. In the linear bandits, Kang et al. (2024) Huang et al. (2024) tackles the issue of heavy-tailed noise on cost function by modifying the reward parameter estimation objective. The former one uses Huber loss for reward function parameter estimation and the latter one truncates the rewards. Zhong et al. (2021) and Xue et al. (2024) propose the median of means and truncation to handle the heavy-tailed noise in the observed rewards. In this work, we study the performance of our proposed estimator, the LSE estimator, under noisy and heavy-tailed reward.

**Estimation of Propensity Scores:** We can estimate the propensity score using different methods, e.g., logistic regression (D'Agostino Jr, 1998; Weitzen et al., 2004), generalized boosted models (McCaffrey et al., 2004), neural networks (Setoguchi et al., 2008), parametric modeling (Xie et al., 2019b) or classification and regression trees (Lee et al., 2010; 2011). Note that, as discussed in (Tsiatis, 2006; Shi et al., 2016), under the estimated propensity scores (noisy propensity score), the variance of the IPS estimator is reduced. In this work, we consider both true and estimated propensity scores, where the estimated propensity scores are modeled via Gamma noise. Our work differs from the line of works on the estimation methods of propensity scores.

**Bandit and Reinforcement Learning under Heavy-tailed Distributions:** Some works discussed the heavy-tailed reward in bandit learning (Medina & Yang, 2016; Bubeck et al., 2013; Shao et al., 2018; Lu et al., 2019; Zhong et al., 2021). Furthermore, some works also discussed the heavy-tailed rewards in RL (Zhuang & Sui, 2021; Zhu et al., 2024). However, off-policy learning with the LBF dataset under a heavy-tailed distribution of weighted reward is overlooked.

**Mean-estimation under Heavy-tailed Distributions:** In (Lugosi & Mendelson, 2019; LUGOSI & MENDELSON, 2021; Hopkins, 2018), the performance of median-of-means and trimmed mean estimators have been studied and the

sub-Gaussian behavior of these estimators are studied. However, median-of-means estimator presents practical challenges in implementation: it requires additional computational resources for data partitioning and mean calculations, while also introducing discontinuities that can prevent gradient-based optimization methods.

**Generalization Error Bound under Heavy-tailed Assumption:** There are also some works that studied the generalization bound – the difference between population risk and empirical risk– of supervised learning under unbounded loss functions, in particular, under heavy-tailed assumption via the PAC-Bayesian approach. Losses with heavier tails are studied by Alquier & Guedj (2018) where probability bounds (non-high probability) are developed. Using a different risk than empirical risk, PAC-Bayes bounds for losses with bounded second and third moments are developed by Holland (2019). Notably, their bounds include a term that can increase with the number of samples $n$. Kuzborskij & Szepesvári (2019) and Haddouche & Guedj (2022) also provide bounds for losses with a bounded second moment. The bounds in (Haddouche & Guedj, 2022) rely on a parameter that must be selected before the training data is drawn. Information-theoretic bounds based on the second moment of loss function $\sup_{h \in \mathcal{H}} |\ell(h, Z) - \mathbb{E}[\ell(h, \tilde{Z})]|$ are also derived in (Lugosi & Neu, 2022). Furthermore, in (Lugosi & Mendelson, 2019, Section 4), the uniform bound via Rademacher complexity analysis over the $L_2$ bounded function space is studied for the median-of-means estimator. Furthermore, the generalization error of tilted empirical risk under heavy-tailed assumption is studied by Aminian et al. (2025). In contrast, we focus on estimation bound and regret analysis of the LSE estimator as a non-linear estimator in OPL and OPE scenarios. For more detailed comparison between batch learning and supervised learning see (Swaminathan & Joachims, 2015a, Table 1).

**Heavy-tailed Rewards in Bandits:** Bandit learning with heavy-tailed reward distributions has been extensively studied. Bubeck et al. (2013) proposed Robust UCB, and Vakili et al. (2013) introduced DSEE as bandit algorithms with theoretical regret guarantees. Yu et al. (2018) proposed a bandit algorithm based on pure exploration with heavy-tailed reward distributions. Heavy-tailed reward distributions are also studied in the context of linear bandits (Shao et al., 2018; Medina & Yang, 2016). Dubey et al. (2020) proposed a decentralized algorithm for cooperative multi-agent bandits when the reward distribution is heavy-tailed. Our work differs from this line of works by considering the heavy-tailed assumption on the weighted reward.

**Heavy-tailed Rewards in RL:** The challenge of heavy-tailed distributions in decision-making has been studied for more than two decades (Georgiou et al., 1999; Hamza & Krim, 2001; Huang & Zhang, 2017; Ruotsalainen et al., 2018). There is a significant amount of study in RL dealing with heavy-tailed reward distribution (Zhu et al., 2023; Zhuang & Sui, 2021; Huang et al., 2024). Moreover, big sparse rewards are a prominent issue in reinforcement learning (Park et al., 2022; Agarwal et al., 2021; Dawood et al., 2023). In such scenarios, there is a far-reaching goal, possibly accompanied by sparse failure states in which the agent attains big positive and negative rewards respectively. For example in safe autonomous driving, accidents are so costly and, hence are assigned large negative rewards. They are also delayed and sparse, which means that they are observed after many steps with a lot of exploration in the environment (Kiran et al., 2021; Amini et al., 2020). This hinders the training and leads to an infeasible slow learning curve. A common approach to tackle this issue is reward shaping which inserts new engineered reward functions alongside the agent's trajectory to provide guidelines for the agent (Kiran et al., 2021). This strategy may fail because it biases the model into the strategy hinted by the new rewards, which may not be the optimal solution for the original problem. Moreover, the method of reward shaping will not necessarily avoid the low-probability high-value rewards, because the imputed rewards are mostly small and high-value rewards still happen with low probability. Therefore, handling low-probability large reward is one of the challenges in this field, which can be modeled by heavy-tailed distributions as discussed with more details in App. G.13.

# B. Preliminaries

## B.1. Notations and Diagram

All notations are summarized in Table 5. An overview of our main theoretical results is provided in Fig. 1.

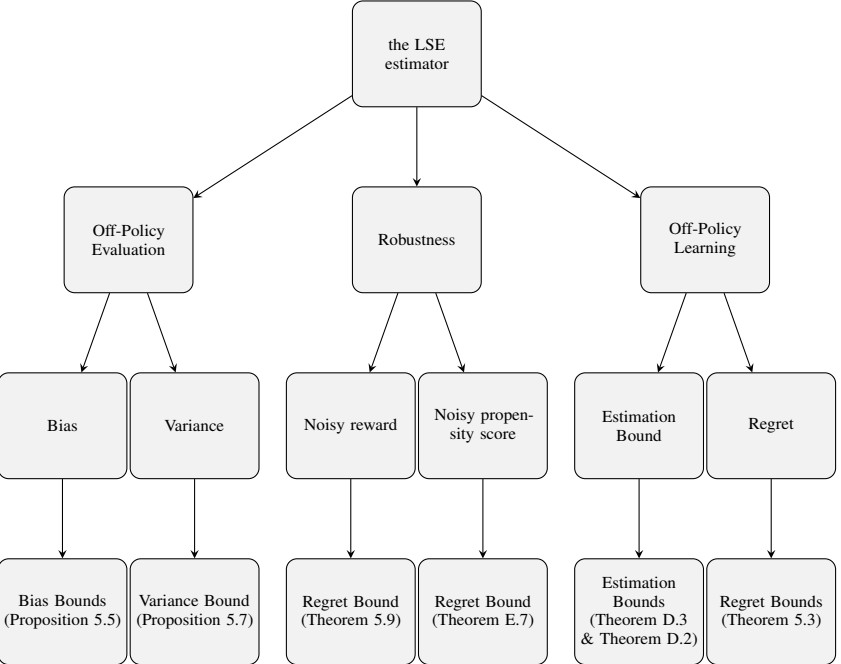

Figure 1: Overview of the main results

Table 5: Summary of notations in the paper

| Notation | Definition | Notation | Definition |
|---|---|---|---|
| $X$ | Context | $A$ | Action |
| $r(X, A)$ | Reward function | $R$ | Reward |
| $n$ | The number of logged data samples | $P_X$ | Distribution over context set |
| $S$ | LBF dataset | $p_i$ | Propensity score $(\pi_0(a_i|x_i))$ |
| $\pi_\theta$ | Learning policy | $w_\theta(A|X)$ | weight $(\pi_\theta(A|X)/\pi_0(A|X))$ |
| $\widehat{V}^\lambda_{\mathrm{LSE}}(S, \pi_\theta)$ | the LSE estimator | $V(\pi_\theta)$ | Value of learning policy $\pi_\theta$ |
| $\nu$ | Upper bound on $(1+\epsilon)$-th moment of weighted reward (Assumption 5.1) | $\pi_0(a|X)$ | Logging policy |
| $\mathrm{Est}_\lambda(\pi_\theta)$ | Estimation error of the LSE estimator | $\mathfrak{R}_\lambda(\pi_{\widehat{\theta}}, S)$ | Regret of the LSE estimator |
| $\mathbb{B}(\widehat{V}^\lambda_{\mathrm{LSE}}(S, \pi_\theta))$ | Bias of the LSE estimator | $\mathbb{V}(\widehat{V}^\lambda_{\mathrm{LSE}}(S, \pi_\theta))$ | Variance of the LSE estimator |

## B.2. Definitions

We define the softmax function

$$\text{softmax}(x_1, x_2, \cdots, x_n) = (s_1, s_2, \cdots, s_n),$$

$$s_i = \frac{e^{x_i}}{\sum_{j=1}^{n} x^{x_j}}, \quad 1 \leq i \leq n.$$

The diag function, $\text{diag}(a_1, a_2, \cdots, a_n) \in \mathbb{R}^{n \times n}$, defines a diagonal matrix with $a_1, a_2, \cdots, a_n$ as elements on its diagonal.

**Definition B.1.** (Cardaliaguet et al., 2019) A functional $U : \mathcal{P}(\mathbb{R}^n) \to \mathbb{R}$ admits a *functional linear derivative* if there is a map $\frac{\delta U}{\delta m} : \mathcal{P}(\mathbb{R}^n) \times \mathbb{R}^n \to \mathbb{R}$ which is continuous on $\mathcal{P}(\mathbb{R}^n)$, such that for all $m, m' \in \mathcal{P}(\mathbb{R}^n)$, it holds that

$$U(m') - U(m) = \int_0^1 \int_{\mathbb{R}^n} \frac{\delta U}{\delta m}(m_\lambda, a)\,(m' - m)(da)\,\mathrm{d}\lambda,$$

where $m_\lambda = m + \lambda(m' - m)$.

## B.3. Theoretical Tools

In this section, we provide the main lemmas which are used in our theoretical proofs.

**Lemma B.2** (Kantorovich-Rubenstein duality of total variation distance, see (Polyanskiy & Wu, 2022)). *The Kantorovich-Rubenstein duality (variational representation) of the total variation distance is as follows:*

$$\mathbb{TV}(m_1, m_2) = \frac{1}{2L} \sup_{g \in \mathcal{G}_L} \left\{ \mathbb{E}_{Z \sim m_1}[g(Z)] - \mathbb{E}_{Z \sim m_2}[g(Z)] \right\}, \tag{15}$$

*where $\mathcal{G}_L = \{g : \mathcal{Z} \to \mathbb{R}, ||g||_\infty \leq L\}$.*

**Lemma B.3** (Hoeffding Inequality, Boucheron et al., 2013). *Suppose that $Z_i$ are sub-Gaussian independent random variables, with means $\mu_i$ and sub-Gaussian parameter $\sigma_i^2$, then we have:*

$$\mathbb{P}\left( \sum_{i=1}^{n}(Z_i - \mu_i) \geq t \right) \leq \exp\left( \frac{-t^2}{2\sum_{i=1}^{n}\sigma_i^2} \right) \tag{16}$$

**Lemma B.4** (Bernstein's Inequality, Boucheron et al., 2013). *Suppose that $S = \{Z_i\}_{i=1}^{n}$ are i.i.d. random variable such that $|Z_i - \mathbb{E}[Z]| \leq R$ almost surely for all $i$, and $\mathbb{V}(Z) = \sigma^2$. Then the following inequality holds with probability at least $(1 - \delta)$ under $P_S$,*

$$\left| \mathbb{E}[Z] - \frac{1}{n}\sum_{i=1}^{n} Z_i \right| \leq \sqrt{\frac{4\sigma^2 \log(2/\delta)}{n}} + \frac{4R\log(2/\delta)}{3n}. \tag{17}$$

The rest of the lemmas are provided with proofs.

---

**Lemma B.5** (Change of variables). *Assume that the following equation holds,*

$$\epsilon = \exp\left\{ -\frac{A\delta^2}{B + C\delta} \right\},$$

*for some positive parameters $A, B, C, \epsilon \geq 0$ and $0 \leq \delta \leq 1$. Then, we have,*

$$\delta \leq \frac{C \log \frac{1}{\epsilon}}{A} + \sqrt{\frac{B \log \frac{1}{\epsilon}}{A}}.$$

*Also, for some $D > 0$, if $A \geq \frac{B \log \frac{1}{\epsilon} + 2DC \log \frac{1}{\epsilon}}{D^2}$, then we have $\delta \leq D$.*

---

*Proof.* We have,

$$\epsilon = \exp\left\{-\frac{A\delta^2}{B + C\delta}\right\} \leftrightarrow A\delta^2 - C\log\frac{1}{\epsilon}\delta - B\log\frac{1}{\epsilon} = 0$$

Given $\delta > 0$ and solving the quadratic equation, we have,

$$\delta = \frac{1}{2A}\left(C\log\frac{1}{\epsilon} + \sqrt{C^2\log^2\frac{1}{\epsilon} + 4AB\log\frac{1}{\epsilon}}\right) = \frac{C}{2}\sqrt{\frac{\log\frac{1}{\epsilon}}{A}}\left(\sqrt{\frac{\log\frac{1}{\epsilon}}{A}} + \sqrt{\frac{\log\frac{1}{\epsilon}}{A} + 4\frac{B}{C^2}}\right)$$

$$\leq C\sqrt{\frac{\log\frac{1}{\epsilon}}{A}}\left(\sqrt{\frac{\log\frac{1}{\epsilon}}{A}} + \sqrt{\frac{B}{C^2}}\right)$$

$$= \frac{C\log\frac{1}{\epsilon}}{A} + \sqrt{\frac{B\log\frac{1}{\epsilon}}{A}},$$

where the inequality is derived from $\sqrt{a+b} \leq \sqrt{a} + \sqrt{b}$.
For the second part, similar argument works for $a = \sqrt{A}$ as the variable ,

$$\frac{C\log\frac{1}{\epsilon}}{A} + \sqrt{\frac{B\log\frac{1}{\epsilon}}{A}} \leq D \leftrightarrow Da^2 - \sqrt{B\log\frac{1}{\epsilon}}a - C\log\frac{1}{\epsilon} \geq 0$$

which is satisfied if $a$ is greater than the bigger root,

$$a \geq \frac{\sqrt{B\log\frac{1}{\epsilon}} + \sqrt{B\log\frac{1}{\epsilon} + 4DC\log\frac{1}{\epsilon}}}{2D}$$

So,

$$A \geq \frac{B\log\frac{1}{\epsilon} + 2DC\log\frac{1}{\epsilon}}{D^2} \geq \left(\frac{\sqrt{B\log\frac{1}{\epsilon}} + \sqrt{B\log\frac{1}{\epsilon} + 4DC\log\frac{1}{\epsilon}}}{2D}\right)^2$$

where the last inequality comes from $\frac{a^2+b^2}{2} \geq \left(\frac{a+b}{2}\right)^2$. Hence if $A \geq \frac{B\log\frac{1}{\epsilon}+2DC\log\frac{1}{\epsilon}}{D^2}$, a is bigger than the largest root and the proposed inequality holds. □

---

**Lemma B.6.** *Assume $A, B, C \in \mathbb{R}^+$. For any $x \in \mathbb{R}^+$ such that,*

$$x \leq \frac{C^2}{2AC + B},$$

*we have,*

$$Ax + \sqrt{Bx} \leq C \tag{18}$$

---

*Proof.* Given $Ax \leq C$, equation equation 18 is equivalent to the following quadratic form.

$$A^2x^2 - (B + 2AC)x + C^2 \geq 0$$

Let $0 < r_1 < r_2$ be the roots of the abovementioned quadratic form. If $X < r_1$, $Ax \leq C$ holds and the quadratic form is positive. So we have the following condition on $x$ to satisfy Equation 18,

$$x \leq \frac{B + 2AC - \sqrt{(B + 2AC)^2 - 4A^2C^2}}{2A^2} = \frac{2C^2}{B + 2AC + \sqrt{(B + 2AC)^2 - 4A^2C^2}}.$$

Since,

$$\frac{C^2}{2AC + B} \leq \frac{2C^2}{B + 2AC + \sqrt{(B + 2AC)^2 - 4A^2C^2}},$$

the condition in the lemma is sufficient for equation 18 to hold. □

**Lemma B.7.** *Let us consider the functions $h_b(y) = \log(y) + \frac{1}{2b^2}y^2$ and $h_a(y) = \log(y) + \frac{1}{2a^2}y^2$ for $a < y < b$. Then $h_b(y)$ and $h_a(y)$ are concave and convex, respectively.*

*Proof.* Taking the second derivative gives us the result, $\frac{d^2}{dy^2}\left(\log(y) + \beta y^2\right) = -\frac{1}{y^2} + 2\beta$.

$\square$

**Lemma B.8.** *We have the following inequality for $y < 0$ and $\epsilon \in [0, 1]$,*

$$e^y \leq 1 + y + \frac{|y|^{1+\epsilon}}{1+\epsilon}. \tag{19}$$

*Proof.* For $y = 0$, equality holds. If suffices to prove that the derivative of LHS of equation 19 is more than the derivative of RHS $\forall y < 0$, i.e.,

$$e^y - 1 + |y|^\epsilon \geq 0.$$

Note that for $y \leq -1$, $|y|^\epsilon \geq 1$ and the inequality trivially holds. For $y > -1$, $|y|^\epsilon$ is minimized at $\epsilon = 1$, so it is sufficient to prove the inequality only for $\epsilon = 1$, which is,

$$e^y - 1 - y \geq 0 \leftrightarrow e^y \geq y + 1$$

and holds $\forall y \leq 0$.

$\square$

*Remark* B.9. In Lemma 28 in (Lugosi & Neu, 2023), the following upper bound for $y < 0$ and $\epsilon \in [0, 1]$ is proposed,

$$e^y \leq 1 + y + |y|^{1+\epsilon}. \tag{20}$$

In contrast, Lemma B.8 is tighter via using $\frac{|y|^{1+\epsilon}}{1+\epsilon}$ instead of $|y|^{1+\epsilon}$.

**Lemma B.10.** *For a positive random variable, $Z > 0$, suppose $\mathbb{E}[Z^{1+\epsilon}] < \nu_z$ for some $\epsilon \in [0, 1]$. Then, the following inequality holds,*
$$\mathbb{E}[Z] \leq \nu_z^{1/(1+\epsilon)}$$

*Proof.* Due to Jensen's inequality, we have,

$$\mathbb{E}[Z] = \mathbb{E}\left[(Z^{1+\epsilon})^{1/(1+\epsilon)}\right] \leq \mathbb{E}\left[Z^{1+\epsilon}\right]^{1/(1+\epsilon)} \leq \nu_z^{1/(1+\epsilon)}.$$

$\square$

**Lemma B.11.** *For a positive random variable, $Z > 0$, suppose $\mathbb{E}[Z^{1+\epsilon}] < \infty$ for some $\epsilon \in [0, 1]$. Then, following inequality for $\lambda < 0$ holds,*

$$\mathbb{E}[Z] \geq \frac{1}{\lambda} \log \mathbb{E}[e^{\lambda Z}] \geq \mathbb{E}[Z] - \frac{1}{1+\epsilon}|\lambda|^\epsilon \mathbb{E}[Z^{1+\epsilon}].$$

*Proof.* The left side inequality follows from Jensen's inequality on $f(z) = \log(z)$. For the right side, we have for $z < 0$,

$$1 + z \leq e^z \leq 1 + z + \frac{1}{1+\epsilon}|z|^{1+\epsilon}.$$

Therefore, we have,

$$
\begin{aligned}
\frac{1}{\lambda} \log \mathbb{E}[e^{\lambda Z}] &\geq \frac{1}{\lambda} \log \mathbb{E}[1 + \lambda Z + \frac{1}{1+\epsilon}|\lambda|^{1+\epsilon} Z^{1+\epsilon}] \\
&= \frac{1}{\lambda} \log \left( 1 + \lambda \mathbb{E}[Z] + \frac{1}{1+\epsilon}|\lambda|^{1+\epsilon} \mathbb{E}\left[Z^{1+\epsilon}\right] \right) \\
&\geq \frac{1}{\lambda} \left( \lambda \mathbb{E}[Z] + \frac{1}{1+\epsilon}|\lambda|^{1+\epsilon} \mathbb{E}[Z^{1+\epsilon}] \right) \\
&= \mathbb{E}[Z] - \frac{1}{1+\epsilon}|\lambda|^{\epsilon} \mathbb{E}[Z^{1+\epsilon}].
\end{aligned}
$$

$\square$

## C. Other Properties of the LSE estimator

**Proposition C.1** (LSE Asymptotic Properties). *The following asymptotic properties of LSE with respect to $\lambda$ holds,*

$$
\lim_{\lambda \to 0} \widehat{V}^{\lambda}_{\text{LSE}}(S) = \frac{1}{n} \left( \sum_{i=1}^{n} r_i w_\theta(a_i, x_i) \right),
$$

$$
\lim_{\lambda \to -\infty} \widehat{V}^{\lambda}_{\text{LSE}}(S) = \min_{i} r_i w_\theta(a_i, x_i),
$$

$$
\lim_{\lambda \to \infty} \widehat{V}^{\lambda}_{\text{LSE}}(S) = \max_{i} r_i w_\theta(a_i, x_i).
$$

*Proof.* For the first limit, we use L'Hopital's rule:

$$
\begin{aligned}
\lim_{\lambda \to 0} \widehat{V}^{\lambda}_{\text{LSE}}(S) &= \lim_{\lambda \to 0} \frac{\log \left( \frac{\sum_{i=1}^{n} e^{\lambda r_i w_\theta(a_i, x_i)}}{n} \right)}{\lambda} \\
&= \lim_{\lambda \to 0} \frac{\left( \frac{\sum_{i=1}^{n} r_i w_\theta(a_i, x_i) e^{\lambda r_i w_\theta(a_i, x_i)}}{\sum_{i=1}^{n} e^{\lambda r_i w_\theta(a_i, x_i)}} \right)}{1} \\
&= \frac{\sum_{i=1}^{n} r_i w_\theta(a_i, x_i)}{n}.
\end{aligned}
$$

For the second limit for $\lambda \to -\infty$ we have:

$$
\begin{aligned}
\min_{i} r_i w_\theta(a_i, x_i) = \frac{1}{\lambda} \log \left( \frac{\sum_{i=1}^{n} e^{\lambda \min_i r_i w_\theta(a_i, x_i)}}{n} \right) &\leq \frac{1}{\lambda} \log \left( \frac{\sum_{i=1}^{n} e^{\lambda r_i w_\theta(a_i, x_i)}}{n} \right) \\
&\leq \frac{1}{\lambda} \log \left( \frac{e^{\lambda \min_i r_i w_\theta(a_i, x_i)}}{n} \right) \\
&= \min_{i} r_i w_\theta(a_i, x_i) - \frac{1}{\lambda} \log n.
\end{aligned}
$$

As both lower and upper tends to $\min_i r_i w_\theta(a_i, x_i)$ we conclude that:

$$
\lim_{\lambda \to -\infty} \frac{1}{\lambda} \log \left( \frac{\sum_{i=1}^{n} e^{\lambda r_i w_\theta(a_i, x_i)}}{n} \right) = \min_{i} r_i w_\theta(a_i, x_i).
$$

A similar argument proves the third limit ($\lambda \to \infty$). $\square$

*Remark* C.2. As shown in (Zhang, 2006, Proposition 1.1), the LSE function is an increasing function with respect to $\lambda$.

**Derivative of the LSE estimator:** The derivative of the LSE estimator can be represented as,

$$\nabla_\theta \widehat{V}_{\text{LSE}}^\lambda(S, \pi_\theta) = \frac{1}{n} \sum_{i=1}^n r_i e^{\lambda(r_i w_\theta(a_i, x_i) - \widehat{V}_{\text{LSE}}^\lambda(S, \pi_\theta))} \nabla_\theta w_\theta(a_i, x_i). \tag{21}$$

Note that, in equation 21, we have a weighted average of the gradient of the weighted reward samples. In contrast to the linear estimators for which the gradient is a uniform mean of reward samples, in the LSE estimator, the gradient for large values of $r_i w_\theta(a_i, x_i)$, $\forall i \in [n]$ (small absolute value), contributes more to the final gradient. It can be interpreted as the robustness of the LSE estimator with respect to the very large absolute values of $r_i w_\theta(a_i, x_i)$ (i.e. high $w_\theta(a, x)$), $\forall i \in [n]$.

It is interesting to study the sensitivity of the LSE estimator with respect to its values.

---

**Lemma C.3.** *The gradient and hessian of the LSE estimator with respect to its values are as follows,*

$$\nabla \widehat{V}_{\text{LSE}}^\lambda(S, \pi_\theta) = \text{softmax}(\lambda r_1 w_\theta(a_1, x_1), \cdots, \lambda r_n w_\theta(a_n, x_n)), \tag{22}$$
$$\nabla^2 \widehat{V}_{\text{LSE}}^\lambda(S) = \lambda \text{diag}(S_n) - \lambda S_n S_n^T, \tag{23}$$

*where $S_n = \text{softmax}(\lambda r_1 w_\theta(a_1, x_1), \cdots, \lambda r_n w_\theta(a_n, x_n))$. Also, LSE is convex when $\lambda > 0$ and concave otherwise.*

---

*Proof.* The two equations can be derived with simple calculations. About the convexity and concavity of $\widehat{V}_{\text{LSE}}^\lambda$, we prove that for $\lambda \geq 0$ the Hessian matrix is positive semi-definite. The proof for concavity for $\lambda < 0$ is similar.

$$\mathbf{z}^T \nabla^2 \widehat{V}_{\text{LSE}}^\lambda \mathbf{z} = \lambda \left( \mathbf{z}^T \text{diag}(S_n) \mathbf{z} - \mathbf{z}^T S_n S_n^T \mathbf{z} \right) = \lambda \left( \sum_{i=1}^n S_n(i) z_i^2 - \left( \sum_{i=1}^n S_n(i) z_i \right)^2 \right)$$

$$= \lambda \left( \left( \sum_{i=1}^n S_n(i) z_i^2 \right) \left( \sum_{i=1}^n S_n(i) \right) - \left( \sum_{i=1}^n S_n(i) z_i \right)^2 \right) \geq 0.$$

Where the last inequality is derived from the Cauchy–Schwarz inequality. $\qquad \square$

Using Lemma C.3, we can show that $\widehat{V}_{\text{LSE}}^\lambda$ is convex for $\lambda \geq 0$ and concave for $\lambda < 0$. Applying Lemma C.3, we can prove that the derivative of the LSE estimator is positive and less than one, i.e.,

$$0 \leq \nabla \widehat{V}_{\text{LSE}}^\lambda(S, \pi_\theta) \leq 1. \tag{24}$$

Furthermore, we prove equation 21 by applying Lemma C.3.

### C.1. LSE estimator and KL Regularization

In this section, we will discuss the connection between the LSE estimator,

$$\text{LSE}_\lambda(\mathbf{Z}) = \frac{1}{\lambda} \log \left( \frac{1}{n} \sum_{i=1}^n e^{\lambda z_i} \right), \tag{25}$$

and the KL regularization problem.

Consider the following KL-regularized expected minimization for $\lambda < 0$,

$$\min_{\mathbf{P} \in \Delta^{n-1}} \sum_{i=1}^n p_i z_i - \frac{1}{\lambda} D_{\text{KL}}(\mathbf{P} \| \text{Uni}(n)), \tag{26}$$

where $\Delta^{n-1}$ denotes the probability simplex and $\text{Uni}(n)$ in the discrete uniform distribution over $n$ mass points. Note that $\lambda < 0$, and the KL divergence is strictly convex with respect to $\mathbf{P}$. Therefore, the objective function in equation 26 is convex.

Then, the solution of the regularized problem in equation 26, is the Gibbs distribution as follows,

$$p_i^\star = \frac{\exp(\lambda z_i)}{\sum_{i=1}^n \exp(\lambda z_i)}, \quad \forall i \in [n], \tag{27}$$

Using equation 27 in equation 26, we have,

$$\sum_{i=1}^n \frac{\exp(\lambda z_i) z_i}{\sum_{j=1}^n \exp(\lambda z_j)} - \frac{1}{\lambda} \sum_{i=1}^n \frac{\exp(\lambda z_i)}{\sum_{j=1}^n \exp(\lambda z_j)} \left( \lambda z_i - \log \left( \frac{1}{n} \sum_{i=1}^n \exp(\lambda z_i) \right) \right)$$
$$= \frac{1}{\lambda} \log \left( \frac{1}{n} \sum_{i=1}^n \exp(\lambda z_i) \right). \tag{28}$$

Therefore, the final value of the KL-regularized minimization problem is the LSE estimator with $\lambda < 0$. Therefore, the LSE estimator with a negative parameter can be interpreted as a KL-regularized expected minimization problem.

# D. Proofs and Details of Section 5

## D.1. Details of Theoretical Comparison

In this section, we compare our estimator with PM, ES, IX, LS and OS from a theoretical perspective in more detail.

### D.1.1. BIAS AND VARIANCE COMPARISON

In this section, we present the bias and variance comparison of different estimators in Table 6. We define power divergence as $P_\alpha(\pi_\theta \| \pi_0) := \int_a \pi_\theta(a|x)^\alpha \pi_0(a|x)^{(1-\alpha)} \mathrm{d}a$ is the power divergence with order $\alpha$. For a fair comparison, we consider the bounded reward function, i.e., $R_{\max} := \sup_{(a,x) \in \mathcal{A} \times \mathcal{X}} r(a,x)$. Therefore, we have $\nu \leq R_{\max}^{1+\epsilon} P_{1+\epsilon}(\pi_\theta \| \pi_0)$ and $\nu_2 \leq R_{\max}^2 P_2(\pi_\theta \| \pi_0)$. We can observe that LSE has the same behavior in comparison with other estimators.

Table 6: Comparison of bias and variance of estimators. $\mathbb{B}^{\mathrm{SN}}$ and $\mathbb{V}^{\mathrm{SN}}$ are the Bias and the Efron-Stein estimate of the variance of self-normalized IPS. For the ES-estimator, we have $T^{ES} = \mathbb{B}^{ES} + (1/n)\big(D_{\mathrm{KL}}(\pi_\theta \| \pi_0) + \log(4/\delta)\big)$. where $D_{\mathrm{KL}}(\pi_\theta \| \pi_0) = \int_a \pi_\theta(a|x) \log(\pi_\theta(a|x)/\pi_0(a|x)) \mathrm{d}a$. For the IX-estimator, $C_\eta(\pi)$ is the smoothed policy coverage ratio. We compare the convergence rate of the estimation bounds for estimators. $B$ and $C$ are constants. For LS estimator, $\mathcal{S}_{\tilde\lambda}(\pi_\theta)$ is the discrepancy between $\pi$ and $\pi_0$.

| Estimator | Variance | Bias |
|---|---|---|
| IPS | $\frac{R_{\max}^2 P_2(\pi_\theta \| \pi_0)}{n}$ | 0 |
| SN-IPS (Swaminathan & Joachims, 2015b) | $R_{\max}^2 V^{\mathrm{SN}}$ | $R_{\max} B^{\mathrm{SN}}$ |
| IPS-TR ($M > 0$) (Ionides, 2008a) | $R_{\max}^2 \frac{P_2(\pi_\theta \| \pi_0)}{n}$ | $R_{\max} \frac{P_2(\pi_\theta \| \pi_0)}{M}$ |
| IX ($\eta > 0$) (Gabbianelli et al., 2023) | $R_{\max} C_\eta(\pi_\theta)/n$ | $R_{\max} \eta C_\eta(\pi_\theta)$ |
| PM ($\lambda \in [0,1]$) (Metelli et al., 2021) | $\frac{R_{\max}^2 P_2(\pi_\theta \| \pi_0)}{n}$ | $R_{\max} \lambda P_2(\pi_\theta \| \pi_0)$ |
| ES ($\alpha \in [0,1]$) (Aouali et al., 2023) | $R_{\max}^2 \frac{\mathbb{E}_{\pi_\theta}[\pi_\theta \cdot \pi_0^{1-2\alpha}]}{n}$ | $R_{\max}(1 - \mathbb{E}_{\pi_\theta}[\pi_0^{1-\alpha}])$ |
| OS ($\tau > 0$) (Su et al., 2020) | $\frac{R_{\max}^2 P_2(\pi_\theta \| \pi_0)}{n}$ | $R_{\max} \frac{P_3(\pi_\theta \| \pi_0)}{\tau}$ |
| LS ($\tilde\lambda \geq 0$) (Sakhi et al., 2024) | $\frac{\mathcal{S}_{\tilde\lambda}(\pi_\theta)}{n}$ | $\tilde\lambda \mathcal{S}_{\tilde\lambda}(\pi_\theta)$ |
| **LSE** ($0 > \lambda > -\infty$ and $\epsilon \in [0,1]$) **(ours)** | $\frac{R_{\max}^2 P_2(\pi_\theta \| \pi_0)}{n}$ | $\frac{1}{1+\epsilon}|\lambda|^\epsilon R_{\max}^{1+\epsilon} P_{1+\epsilon}(\pi_\theta \| \pi_0) - \frac{B}{2n|\lambda|}$ |

**LSE Variance:** Note that in variance comparison between IPS and LSE, the LSE variance is less than IPS, as shown in Proposition 5.7. However in Table 6, we use a looser upper bound to compare bounds in terms of the same parameter $R_{\max}$.

**Bias and Variance Trade-off:** Observe that for the bias and variance of the LSE estimator, there is a trade-off with respect to $\lambda < 0$. Specifically, reducing $\lambda$ increases the bias of the LSE estimator,

$$\mathbb{B}(\widehat{V}_{\mathrm{LSE}}^\lambda(S, \pi_\theta)) = \mathbb{E}[w_\theta(A, X)R] - \mathbb{E}[\widehat{V}_{\mathrm{LSE}}^\lambda(S, \pi_\theta)]. \tag{29}$$

This is a consequence of the increasing property of the LSE with respect to $\lambda$ (see Remark C.2).

Additionally, for the variance, we have the following bound,

$$\text{Var}(\widehat{V}^{\lambda}_{\text{LSE}}(S, \pi_\theta)) \leq \mathbb{E}[(\widehat{V}^{\lambda}_{\text{LSE}}(S, \pi_\theta))^2]. \tag{30}$$

It is important to note that decreasing $\lambda$ reduces the upper bound on the variance of the LSE estimator.

Therefore, by decreasing $\lambda < 0$, the bias increases and the variance decreases.

### D.1.2. COMPARISON WITH PM ESTIMATOR

In (Metelli et al., 2018), the authors proposed the following PM estimator for two hyper-parameter $(\lambda_p, s)$,

$$\widehat{V}_{\text{PM}}(S, \pi_\theta) = \frac{1}{n} \sum_{i=1}^{n} ((1 - \lambda_p) w_\theta(a_i, x_i)^s + \lambda_p)^{\frac{1}{s}} r_i.$$

An upper bound on estimation error of PM estimator for $(\lambda_p, s = -1)$, is provided in (Metelli et al., 2018, Theorem 5.1),

$$\text{Est}_{\text{PM}}(S, \pi_\theta) \leq \|R\|_\infty (2 + \sqrt{3}) \left( \frac{2 P_\alpha(\pi_\theta \| \pi_0)^{\frac{1}{\alpha - 1}} \log \frac{1}{\delta}}{3(\alpha - 1)^2 n} \right)^{1 - \frac{1}{\alpha}}, \tag{31}$$

where $\text{Est}_{\text{PM}}(S, \pi_\theta) = V(\pi_\theta) - \widehat{V}_{\text{PM}}(S, \pi_\theta)$ and $\alpha \in (1, 2]$. In contrast to the bound presented in equation 31, which necessitates a bounded reward, exhibits a dependence on $\log(1/\delta)^{\frac{\epsilon}{1+\epsilon}}$ and two hyper-parameter $(s, \lambda_p)$, our work offers several advancements. We derive both upper and lower bounds on estimation error, as detailed in Theorem D.3 and Theorem D.2, respectively. These bounds help us for our subsequent derivation of an upper bound on regret. Notably, our bounds demonstrate a more favorable dependence of $\log(1/\delta)^{1/2}$. This improvement not only eliminates the requirement for bounded rewards but also provides a tighter concentration. Furthermore, we provide theoretical analysis for robustness with respect to both noisy reward and noisy propensity scores, and we just have one hyperparameter. Note that the assumption on $P_\alpha(\pi_\theta \| \pi_0)$ for $\alpha = 1 + \epsilon$ in (Metelli et al., 2018) is similar to bounded $(1 + \epsilon)$-th moment of weight function, $w_\theta(a, x)$ for a bounded reward function.

### D.1.3. COMPARISON WITH ES ESTIMATOR

The ES estimator (Aouali et al., 2023) is represented as,

$$\widehat{V}^{\alpha}_{\text{ES}}(\pi_\theta) = \frac{1}{n} \sum_{i=1}^{n} r_i \frac{\pi_\theta(a_i | x_i)}{\pi_0(a_i | x_i)^\alpha}, \quad \alpha \in [0, 1]. \tag{32}$$

In (Aouali et al., 2023, Theorem 4.1), an upper bound on concentration is derived via PAC-Bayesian approach for $\alpha \in [0, 1]$,

$$|V(\pi_\mathbb{Q}) - \widehat{V}^{\alpha}_{\text{ES}}(\pi_\mathbb{Q})| \leq \sqrt{\frac{\text{KL}_1(\pi_\mathbb{Q})}{2n}} + B^{\alpha}_n(\pi_\mathbb{Q}) + \frac{\text{KL}_2(\pi_\mathbb{Q})}{n\lambda}$$
$$+ \frac{\lambda}{2} \bar{V}^{\alpha}_n(\pi_\mathbb{Q}).$$

where $\text{KL}_1(\pi_\mathbb{Q}) = D_{\text{KL}}(\mathbb{Q} \| \mathbb{P}) + \ln \frac{4\sqrt{n}}{\delta}$, and

$$\text{KL}_2(\pi_\mathbb{Q}) = D_{\text{KL}}(\mathbb{Q} \| \mathbb{P}) + \ln \frac{4}{\delta}, \tag{33}$$

$$B^{\alpha}_n(\pi_\mathbb{Q}) = 1 - \frac{1}{n} \sum_{i=1}^{n} \mathbb{E}_{a \sim \pi_\mathbb{Q}(\cdot | x_i)} \left[ \pi_0^{1 - \alpha}(a | x_i) \right],$$

$$\bar{V}^{\alpha}_n(\pi_\mathbb{Q}) = \frac{1}{n} \sum_{i=1}^{n} \mathbb{E}_{a \sim \pi_0(\cdot | x_i)} \left[ \frac{\pi_\mathbb{Q}(a | x_i)}{\pi_0(a | x_i)^{2\alpha}} \right] + \frac{\pi_\mathbb{Q}(a_i | x_i) \|R\|_\infty^2}{\pi_0(a_i | x_i)^{2\alpha}},$$

where $\mathbb{Q}$ and $\mathbb{P}$ are posterior and prior distributions over the set of hypothesis, $\widehat{R}_n^\alpha(\pi_\mathbb{Q})$ is ES estimator and $R(\pi_\mathbb{Q})$ is true risk. The ES estimator's bound exhibits several limitations. Primarily, it requires a bounded reward. Moreover, the upper bound on the concentration (estimation error) of the ES estimator converges at a rate of $O(\log(n)n^{-1/2})$, which is suboptimal. A notable drawback is the presence of the term $B_n^\alpha(\pi_\mathbb{Q})$, which remains constant for $\alpha > 1$ and does not decrease with increasing sample size $n$. In contrast, we derive an upper bound on the Regret with a convergence rate of $O(n^{-1/2})$ under the condition of bounded second moment ($\epsilon = 1$) and can be extended for heavy-tailed scenarios under bounded reward. This improved rate not only eliminates the logarithmic factor but also demonstrates a tighter concentration. Furthermore, we have a theoretical analysis for robustness with respect to both noisy reward and noisy propensity scores. Finally, the noisy reward scenario is not studied under the ES estimator.

### D.1.4. COMPARISON WITH IX ESTIMATOR

The IX estimator (Gabbianelli et al., 2023) is defined as for $\eta > 0$,

$$\widehat{V}_{\text{ES}}^\eta(S, \pi_\theta) := \frac{1}{n} \sum_{i=1}^n \frac{\pi_\theta(a_i|x_i)}{\pi_\theta(a_i|x_i) + \eta} r_i.$$

The following upper bound on regret of IX estimator is derived in (Gabbianelli et al., 2023, Theorem 1),

$$\mathfrak{R}(\pi_{\theta^*}) \leq \sqrt{\frac{\log(2|\Pi_\theta|/\delta)}{n}}(2\eta C_\eta(\pi_{\theta^*}) + 1), \tag{34}$$

where

$$C_\eta(\pi_\theta) = \mathbb{E}\left[\sum_a \frac{\pi_\theta(a|X)}{\pi_0(a|X) + \eta} \cdot r(X, a)\right]. \tag{35}$$

In equation 34, it is assumed that reward is bounded. The term $C_\eta(\pi_\theta)$ can be large if $\eta$ is small. While a small $\eta$ is desirable for reducing bias, it can simultaneously increase $C_\eta(\pi_\theta)$, potentially compromising the tightness of the bound. The bounded reward in $[0, 1]$ is needed for the proof of regret bound as $R^2 \leq R$ for $R \in [0, 1]$. Moreover, the process of tuning $\eta$ in the IX estimator is particularly sensitive.

### D.1.5. COMPARISON WITH LOGARITHMIC SMOOTHING

We provide a theoretical comparison with the Logarithmic Smoothing (LS) estimator (Sakhi et al., 2024).

The LS estimator is,

$$\hat{V}_n^{\tilde{\lambda}}(\pi) = \frac{1}{n} \sum_{i=1}^n \frac{1}{\tilde{\lambda}} \log(1 + \tilde{\lambda} w_\theta(x_i, a_i) r_i),$$

for $\tilde{\lambda} > 0$. As mentioned in (Sakhi et al., 2024), a Taylor expansion of LS estimator around $\tilde{\lambda} = 0$ yields,

$$\hat{V}_n^{\tilde{\lambda}}(\pi) = \hat{V}_n(\pi) + \sum_{\ell=2}^\infty \frac{(-1)^\ell \tilde{\lambda}^{\ell-1}}{\ell} \left(\frac{1}{n} \sum_{i=1}^n (w_\theta(x_i, a_i) r_i)^\ell\right).$$

Furthermore, the authors introduced,

$$\mathcal{S}_{\tilde{\lambda}}(\pi) = \mathbb{E}\left[\frac{(w_\theta(X, A)r)^2}{(1 + \tilde{\lambda} w_\pi(X, A)r)}\right],$$

where in (Sakhi et al., 2024, Proposition 7), a bounded second moment is needed to derive the estimation error bound. Furthermore, for PAC-Bayesian analysis, the author proposed a linearized version,

$$\hat{V}_n^{\tilde{\lambda}\text{-LIN}}(\pi) = \frac{1}{n} \sum_{i=1}^n \frac{\pi(a_i|x_i)}{\tilde{\lambda}} \log\left(1 + \frac{\tilde{\lambda} r_i}{\pi_0(a_i|x_i)}\right),$$

Note that, the linearized version of the LS estimator is bounded by IPS estimator due to $\log(1 + x) \leq x$ inequality. Then, for the LS-LIN estimator the PAC-Bayesian upper bound on the Regret of the LS-LIN estimator is derived in (Sakhi et al., 2024, Proposition 11) as follows,

$$0 \leq V(\hat{\pi}_n) - V(\pi_Q^*) \leq \tilde{\lambda} S_{\tilde{\lambda}}^{\text{LIN}}(\pi_Q^*) + \frac{2(\text{KL}(Q\|P) + \ln(2/\delta))}{\tilde{\lambda} n},$$

where $S_{\tilde{\lambda}}^{\text{LIN}}(\pi) = \mathbb{E}\left[\frac{\pi(a|x)r^2}{\pi_0(a|x) + \tilde{\lambda}\pi_0(a|x)r}\right]$.

**Theoretical Comparison:** The key distinction between the LS estimator and our LSE estimator is that we explicitly assume the heavy-tailed weighted reward and can drive a better convergence rate.

In (Sakhi et al., 2024, Proposition 7), the authors demonstrate that under the assumption of *a bounded second moment of the weighted reward*, the convergence rate is $O(1/\sqrt{n})$.

However, if the second moment is not bounded, from (Sakhi et al., 2024) we only know that:

$$S_{\tilde{\lambda}}(\pi) = \mathbb{E}\left[\frac{(w(X,A)r)^2}{1 + \tilde{\lambda}w(X,A)r}\right] \leq \min\left(\frac{1}{\tilde{\lambda}}\mathbb{E}\left[w(X,A)r\right], \mathbb{E}\left[(w(X,A)r)^2\right]\right).$$

If we replace $S_{\tilde{\lambda}}(\pi)$ with $\frac{1}{\tilde{\lambda}}\mathbb{E}\left[w(X,A)r\right]$ in (Sakhi et al., 2024, Proposition 7), we get $O(1)$ as convergence rate. In contrast, our analysis yields a convergence rate of

$$O(n^{-\epsilon/(1+\epsilon)}),$$

for bounded $(1 + \epsilon)$-th moment.

This result demonstrates that our assumption is both precise and necessary to achieve the optimal convergence rate for regret under the heavy-tailed assumption.

### D.1.6. COMPARISON WITH OPTIMISTIC SHRINKAGE

The OS estimator (Su et al., 2020) is represented as for $\tau \geq 0$.

$$\widehat{V}_{\text{OS}}(\pi_\theta) = \frac{1}{n}\sum_{i=1}^{n}\frac{\tau w_\theta(a_i, x_i)}{w_\theta^2(a_i, x_i) + \tau}r_i. \tag{36}$$

In (Metelli et al., 2021, Theorem E.1), an upper bound for the right tail of the concentration inequality for the OS estimator is established, which depends on $P_3(\pi_\theta\|\pi_0)$. Consequently, this estimator fails to ensure reliable performance under heavy-tailed assumptions, even when the reward is bounded. Furthermore, due to applying the Bernstein inequality in the proof, theoretical results can not be extended to unbounded reward.

### D.1.7. COMPARISON UNDER BOUNDED REWARD ASSUMPTION

In this section, we compare different estimators by assuming bounded reward. Note that, under bounded reward assumption, $R \in [0, R_{\max}]$, our Assumption 5.1, would be simplified as follows,

**Assumption D.1.** The $P_X \otimes \pi_0(A|X)$ are such that for all learning policy $\pi_\theta(A|X) \in \Pi_\theta$ and some $\epsilon \in [0, 1]$, the $(1+\epsilon)$-th moment of the weight function is bounded,

$$\mathbb{E}_{P_X \otimes \pi_0(A|X)}\left[\left(w_\theta(A, X)\right)^{1+\epsilon}\right] \leq \nu_w. \tag{37}$$

Note that, under Assumption D.1, our theoretical results hold by replacing $\nu$ with $\nu_w R_{\max}^{1+\epsilon}$. In the following, we compare main estimators, PM, ES, IX, LS and OS with LSE under Assumption D.1,

- The PM estimator provides an upper bound on concentration inequality under Assumption D.1. However, a lower bound on estimation error (concentration inequality) is not provided. Furthermore, for $\epsilon = 0$, we can have a bounded upper bound on estimation error. However, (Metelli et al., 2021, Theorem 5.1) is infinite for $\epsilon = 0$.[11]

---

[11]Note that in (Metelli et al., 2021), the authors consider $\alpha \in (1, 2]$ where $\alpha = \epsilon + 1$ and $\epsilon \in (0, 1]$.

- The ES estimator, does not support Assumption D.1 and an assumption on bounded $\frac{\pi_\theta}{\pi_0^{2\alpha}}$ for $\alpha \in (0,1)$ is needed. Furthermore, the convergence rate of the estimation bound on the ES estimator is worse than ours in $\epsilon = 1$.

- For the OS estimator, the bounded assumption on the third moment of the weight function is needed. Therefore, it does not support Assumption D.1.

- The theoretical results for LS estimator do not need bounded $(1 + \epsilon)$-th moment of weight function, Assumption D.1. However, under Assumption D.1, we can not derive the optimal rate of regret, $O(n^{\frac{-\epsilon}{1+\epsilon}})$ for $\epsilon \in [0,1]$ under LS estimator.

- For IX estimator, using the upper bound on regret in (Gabbianelli et al., 2023, Theorem 7), requires bounded $C_0(\pi_{\theta*})$, which can impose a stronger condition than Assumption D.1.

### D.1.8. COMPARISON WITH THE ASSUMPTION 1 IN SWITCH ESTIMATOR

The switch estimator introduced in (Wang et al., 2017) adaptively chooses between model-free estimation and an estimated reward function based on importance weights. While (Wang et al., 2017) requires the existence of finite $(2 + \tilde{\epsilon})$-th moments (for $\epsilon > 0$) in their Assumption 1, our work operates under a weaker condition. We only require bounded $(1 + \epsilon)$-th moments for some $\epsilon \in [0,1]$. This distinction is significant—our assumption (Assumption 5.1) encompasses cases where the second moment and $(2 + \tilde{\epsilon})$-th moment for $\tilde{\epsilon} > 0$ do not exist. In contrast, (Wang et al., 2017, Assumption 1), which requires the finiteness of the $(2 + \tilde{\epsilon})$-th moments, imposes a strictly stronger condition on the underlying distribution. Therefore, we can not apply the approach in (Wang et al., 2017) in our case.

### D.2. Proofs and Details of Regret Bounds

---

**Lemma 5.2** (**Restated**). *Consider the random variable $Z > 0$. For $\epsilon \in [0,1]$, the following upper bound holds on the variance of $e^{\lambda Z}$ for $\lambda < 0$,*

$$\mathbb{V}\left(e^{\lambda Z}\right) \leq |\lambda|^{1+\epsilon}\mathbb{E}[Z^{1+\epsilon}]. \tag{38}$$

---

*Proof.* We have,

$$|e^{\lambda Z} - e^{\lambda C_1}| = \left|\int_{\lambda C_1}^{\lambda z} e^y dy\right| \leq |\lambda(z - C_1)|e^{\max(\lambda z, \lambda C_1)} \leq |\lambda||z - C_1|.$$

Then it holds that

$$\mathbb{V}(e^{\lambda Z}) = \min_{C_1 \in \mathbb{R}^+} \mathbb{E}\left[(e^{\lambda Z} - e^{\lambda C_1})^2\right] = \min_{C_1 \in \mathbb{R}^+} \mathbb{E}\left[|e^{\lambda Z} - e^{\lambda C_1}|^{1-\epsilon}|e^{\lambda Z} - e^{\lambda C_1}|^{1+\epsilon}\right]$$

$$= \min_{C_1 \in \mathbb{R}^+} \mathbb{E}\left[|e^{\lambda Z} - e^{\lambda C_1}|^{1-\epsilon}|\lambda|^{1+\epsilon}|Z - C_1|^{1+\epsilon}\right]$$

$$\leq \min_{C_1 \in \mathbb{R}^+} \mathbb{E}\left[|\lambda|^{1+\epsilon}|Z - C_1|^{1+\epsilon}\right] \leq |\lambda|^{1+\epsilon}\mathbb{E}[Z^{1+\epsilon}],$$

where the last inequality holds due to the fact that $|e^{\lambda Z} - e^{\lambda C_1}|^{1-\epsilon} \leq 1$. $\qquad\square$

Furthermore, we are interested in providing high probability upper and lower bounds on $\mathrm{Est}_\lambda(\pi_\theta)$,

$$P(\mathrm{Est}_\lambda(\pi_\theta) > g_u(\delta, n, \lambda)) \leq \delta, \quad \text{and}, \quad P(\mathrm{Est}_\lambda(\pi_\theta) < g_l(\delta, n, \lambda)) \leq \delta.$$

where $0 < \delta < 1$ and $n$ is the number of samples in LBF dataset. We first provide an upper bound on estimation error.

---

**Theorem D.2.** *Given Assumption 5.1, with probability at least $1 - \delta$, then the following upper bound holds on the estimation error of the LSE for a learning policy $\pi_\theta \in \Pi_\theta$*

$$\mathrm{Est}_\lambda(\pi_\theta) \leq \frac{1}{1+\epsilon}|\lambda|^\epsilon \nu - \frac{1}{\lambda}\sqrt{\frac{4|\lambda|^{1+\epsilon}\nu \log(2/\delta)}{n \exp(2\lambda\nu^{1/(1+\epsilon)})}} - \frac{4\log(2/\delta)}{3\lambda \exp(\lambda\nu^{1/(1+\epsilon)})n}.$$

---

*Proof.* To ease the notation, we consider $Y_\theta(A, X) = w_\theta(A, X)R$. Using Bernstein's inequality (Lemma B.4), with probability $(1 - \delta)$, we have,

$$\mathbb{E}[\exp(\lambda Y_\theta(A, X))] - \frac{1}{n} \sum_{i=1}^n \exp(\lambda Y_\theta(a_i, x_i)) \geq -\sqrt{\frac{4\mathbb{V}(\exp(\lambda Y_\theta(A, X)))\log(2/\delta)}{n}} - \frac{4\log(2/\delta)}{3n}.$$

Using Lemma 5.2, $\mathbb{V}(\exp(\lambda Y_\theta(A, X))) \leq |\lambda|^{1+\epsilon}\nu$, we have,

$$\mathbb{E}[\exp(\lambda Y_\theta(A, X))] - \frac{1}{n} \sum_{i=1}^n \exp(\lambda Y_\theta(a_i, x_i)) \geq -\sqrt{\frac{4|\lambda|^{1+\epsilon}\nu\log(2/\delta)}{n}} - \frac{4\log(2/\delta)}{3n}.$$

As the log function is an increasing function, the following holds with probability at least $(1 - \delta)$,

$$\widehat{V}_{\text{LSE}}^\lambda(S, \pi_\theta) \geq \frac{1}{\lambda} \log\left(\mathbb{E}[e^{\lambda Y_\theta(A,X)}] + \sqrt{\frac{4|\lambda|^{1+\epsilon}\nu\log(2/\delta)}{n}} + \frac{4\log(2/\delta)}{3n}\right).$$

where recall that $\widehat{V}_{\text{LSE}}^\lambda(S, \pi_\theta) = \frac{1}{\lambda} \log\left(\frac{1}{n} \sum_{i=1}^n \exp(\lambda y_\theta(a_i, x_i))\right)$. With probability at least $(1 - \delta)$, using the inequality $\log(x + y) \leq \log(x) + y/x$ for $x > 0$,

$$\widehat{V}_{\text{LSE}}^\lambda(S, \pi_\theta) \geq \frac{1}{\lambda} \log\left(\mathbb{E}[e^{\lambda Y_\theta(A,X)}] + \sqrt{\frac{4|\lambda|^{1+\epsilon}\nu\log(2/\delta)}{n}} + \frac{4\log(2/\delta)}{3n}\right)$$

$$\geq \frac{1}{\lambda} \log\left(\mathbb{E}[e^{\lambda Y_\theta(A,X)}])\right) + \frac{1}{\lambda\mathbb{E}[e^{\lambda Y_\theta(A,X)}]} \sqrt{\frac{4|\lambda|^{1+\epsilon}\nu\log(2/\delta)}{n}} + \frac{4\log(2/\delta)}{3\lambda\mathbb{E}[e^{\lambda Y_\theta(A,X)}]n}.$$

Using Lemma B.11, we have with probability at least $(1 - \delta)$,

$$\widehat{V}_{\text{LSE}}^\lambda(S, \pi_\theta) \geq \mathbb{E}[Y_\theta(A, X)] - \frac{1}{1 + \epsilon}|\lambda|^\epsilon\mathbb{E}[Y_\theta(A, X)^{1+\epsilon}]$$

$$+ \frac{1}{\lambda\mathbb{E}[e^{\lambda Y_\theta(A,X)}]} \sqrt{\frac{4|\lambda|^{1+\epsilon}\nu\log(2/\delta)}{n}} + \frac{4\log(2/\delta)}{3\lambda\mathbb{E}[e^{\lambda Y_\theta(A,X)}]n}$$

$$\geq \mathbb{E}[Y_\theta(A, X)] - \frac{1}{1 + \epsilon}|\lambda|^\epsilon\mathbb{E}[Y_\theta(A, X)^{1+\epsilon}]$$

$$+ \frac{1}{\lambda\mathbb{E}[e^{\lambda Y_\theta(A,X)}]} \sqrt{\frac{4|\lambda|^{1+\epsilon}\nu\log(2/\delta)}{n}} + \frac{4\log(2/\delta)}{3\lambda\mathbb{E}[e^{\lambda Y_\theta(A,X)}]n}$$

$$\geq \mathbb{E}[Y_\theta(A, X)] - \frac{1}{1 + \epsilon}|\lambda|^\epsilon\nu + \frac{1}{\lambda} \sqrt{\frac{4|\lambda|^{1+\epsilon}\nu\log(2/\delta)}{n\exp(2\lambda\nu^{1/(1+\epsilon)})}} + \frac{4\log(2/\delta)}{3\lambda\exp(\lambda\nu^{1/(1+\epsilon)})n}.$$

The final result holds by by applying Lemma B.10 to $\mathbb{E}[e^{\lambda Y_\theta(A,X)}] \geq \exp(\lambda\nu^{1/(1+\epsilon)})$. $\qquad\square$

Next, we provide a lower bound on estimation error.

**Theorem D.3.** *Given Assumption 5.1, and assuming $n \geq \frac{\left(2|\lambda|^{1+\epsilon}\nu + \frac{4}{3}\gamma\right)\log\frac{1}{\delta}}{\gamma^2 \exp(2\lambda\nu^{1/(1+\epsilon)})}$, then there exists $\gamma \in (0, 1)$ such that with probability at least $1 - \delta$, the following lower bound on estimation error of the LSE for a learning policy $\pi_\theta \in \Pi_\theta$ holds*

$$\text{Est}_\lambda(\pi_\theta) \geq \frac{1}{\lambda(1 - \gamma)} \sqrt{\frac{4|\lambda|^{1+\epsilon}\nu\log(2/\delta)}{n\exp(2\lambda\nu^{1/(1+\epsilon)})}} + \frac{4\log(2/\delta)}{3(1 - \gamma)\lambda\exp(\lambda\nu^{1/(1+\epsilon)})n}$$

*Proof.* To ease the notation, we consider $Y_\theta(A, X) = Rw_\theta(A, X)$. Using Bernstein's inequality (Lemma B.4), with probability $(1 - \delta)$, we have,

$$\mathbb{E}[\exp(\lambda Y_\theta(A, X))] - \frac{1}{n} \sum_{i=1}^{n} \exp(\lambda Y_\theta(a_i, x_i)) \leq \sqrt{\frac{4\mathbb{V}(\exp(\lambda Y_\theta(A, X))) \log(2/\delta)}{n}} + \frac{4\log(2/\delta)}{3n}.$$

Using Lemma 5.2, $\mathbb{V}(\exp(\lambda Y_\theta(A, X))) \leq |\lambda|^{1+\epsilon}\nu$, we have,

$$\mathbb{E}[\exp(\lambda Y_\theta(A, X))] - \frac{1}{n} \sum_{i=1}^{n} \exp(\lambda Y_\theta(a_i, x_i)) \leq \sqrt{\frac{4|\lambda|^{1+\epsilon}\nu \log(2/\delta)}{n}} + \frac{4\log(2/\delta)}{3n}.$$

As the log function is an increasing function, the following holds with probability at least $(1 - \delta)$,

$$\widehat{V}_{\mathrm{LSE}}^{\lambda}(S, \pi_\theta) \leq \frac{1}{\lambda} \log\left(\mathbb{E}[e^{\lambda Y_\theta(A,X)}] - \sqrt{\frac{4|\lambda|^{1+\epsilon}\nu \log(2/\delta)}{n}} - \frac{4\log(2/\delta)}{3n}\right).$$

where recall that $\widehat{V}_{\mathrm{LSE}}^{\lambda}(S, \pi_\theta) = \frac{1}{\lambda} \log\left(\frac{1}{n} \sum_{i=1}^{n} \exp(\lambda y_\theta(a_i, x_i))\right)$. Without loss of generality, we can assume that,

$$\sqrt{\frac{4|\lambda|^{1+\epsilon}\nu \log(2/\delta)}{n}} + \frac{4\log(2/\delta)}{3n} \leq \gamma \mathbb{E}[e^{\lambda Y_\theta(A,X)}] \tag{39}$$

for some $\gamma \in (0, 1)$. Using the inequality $\log(z - y) \geq \log(z) - \frac{y}{z-y}$ for $z > y > 0$, and assuming $z = \mathbb{E}[e^{\lambda Y_\theta(A,X)}]$ and $y = \sqrt{\frac{4|\lambda|^{1+\epsilon}\nu \log(2/\delta)}{n}} + \frac{4\log(2/\delta)}{3n}$ and combining with equation 39, then with probability $(1 - \delta)$, we have,

$$\begin{aligned}
\widehat{V}_{\mathrm{LSE}}^{\lambda}(S, \pi_\theta) &\leq \frac{1}{\lambda} \log\left(\mathbb{E}[e^{\lambda Y_\theta(A,X)}] - \sqrt{\frac{4|\lambda|^{1+\epsilon}\nu \log(2/\delta)}{n}} - \frac{4\log(2/\delta)}{3n}\right) \\
&\leq \frac{1}{\lambda} \log\left(\mathbb{E}[e^{\lambda Y_\theta(A,X)}]\right) - \frac{1}{\lambda(1 - \gamma)\mathbb{E}[e^{\lambda Y_\theta(A,X)}]} \sqrt{\frac{4|\lambda|^{1+\epsilon}\nu \log(2/\delta)}{n}} \\
&\quad - \frac{4\log(2/\delta)}{(1 - \gamma)\lambda 3\mathbb{E}[e^{\lambda Y_\theta(A,X)}]n}.
\end{aligned}$$

Equation 39 can be considered as quadratic equation in terms of $\frac{1}{\sqrt{n}}$. Then, using lemma B.6, we have,

$$\frac{\left(2|\lambda|^{1+\epsilon}\nu + \frac{4}{3}\gamma\right) \log(2/\delta)}{\gamma^2 \exp(2\lambda\nu^{1/(1+\epsilon)})} \leq n. \tag{40}$$

Using Lemma B.11, with probability at least $(1 - \delta)$ we have

$$\begin{aligned}
\widehat{V}_{\mathrm{LSE}}^{\lambda}(S, \pi_\theta) &\leq \mathbb{E}[Y_\theta(A, X)] - \frac{1}{\lambda(1 - \gamma)\mathbb{E}[e^{\lambda Y_\theta(A,X)}]} \sqrt{\frac{4|\lambda|^{1+\epsilon}\nu \log(2/\delta)}{n}} - \frac{4\log(2/\delta)}{3(1 - \gamma)\lambda\mathbb{E}[e^{\lambda Y_\theta(A,X)}]n} \\
&\leq \mathbb{E}[Y_\theta(A, X)] - \frac{1}{(1 - \gamma)\lambda\mathbb{E}[e^{\lambda Y_\theta(A,X)}]} \sqrt{\frac{4|\lambda|^{1+\epsilon}\nu \log(2/\delta)}{n}} - \frac{4\log(2/\delta)}{3(1 - \gamma)\lambda\mathbb{E}[e^{\lambda Y_\theta(A,X)}]n} \\
&\leq \mathbb{E}[Y_\theta(A, X)] - \frac{1}{\lambda(1 - \gamma)} \sqrt{\frac{4|\lambda|^{1+\epsilon}\nu \log(2/\delta)}{n \exp(2\lambda\nu^{1/(1+\epsilon)})}} - \frac{4\log(2/\delta)}{3(1 - \gamma)\lambda \exp(\lambda\nu^{1/(1+\epsilon)})n}.
\end{aligned}$$

The final result holds by applying Lemma B.10 to $\mathbb{E}[e^{\lambda Y_\theta(A,X)}] \geq \exp(\lambda\nu^{1/(1+\epsilon)})$. $\qquad\square$

Using the previous upper and lower bounds on estimation error, we can provide an upper bound on the regret of the LSE estimator.

> **Theorem 5.3 (Restated).** *For any $\gamma \in (0,1)$, given Assumption 5.1, assuming finite policy set $|\Pi_\theta| < \infty$ and $n \geq \frac{(2|\lambda|^{1+\epsilon}\nu + \frac{4}{3}\gamma)\log\frac{1}{\delta}}{\gamma^2\exp(2\lambda\nu^{1/(1+\epsilon)})}$, with probability at least $(1-\delta)$, the following upper bound holds on the regret of the LSE estimator,*
>
> $$0 \leq \Re_\lambda(\pi_{\widehat{\theta}}, S) \leq \frac{|\lambda|^\epsilon}{1+\epsilon}\nu - \frac{4(2-\gamma)}{3(1-\gamma)}\frac{\log\frac{4|\Pi_\theta|}{\delta}}{n\lambda\exp(\lambda\nu^{1/(1+\epsilon)})} - \frac{(2-\gamma)}{(1-\gamma)\lambda}\sqrt{\frac{4|\lambda|^{1+\epsilon}\nu\log\frac{4|\Pi_\theta|}{\delta}}{n\exp(2\lambda\nu^{1/(1+\epsilon)})}}.$$

*Proof.* We have,

$$V(\pi_{\theta^*}) - V(\pi_{\widehat{\theta}}) = \underbrace{V(\pi_{\theta^*}) - \widehat{V}^\lambda_{\text{LSE}}(S, \pi_{\theta^*})}_{I_1} + \underbrace{\widehat{V}^\lambda_{\text{LSE}}(S, \pi_{\theta^*}) - \widehat{V}^\lambda_{\text{LSE}}(S, \pi_{\widehat{\theta}})}_{I_2} + \underbrace{\widehat{V}^\lambda_{\text{LSE}}(S, \pi_{\widehat{\theta}}) - V(\pi_{\widehat{\theta}})}_{I_3}. \tag{41}$$

Using upper bound on estimation error, Theorem D.2, and union bound (Shalev-Shwartz & Ben-David, 2014), with probability at least $1 - \delta$, the following upper bound holds on term $I_1$,

$$V(\pi_{\theta^*}) - \widehat{V}^\lambda_{\text{LSE}}(S, \pi_{\theta^*}) \leq \sup_{\pi_\theta \in \Pi_\theta} V(\pi_\theta) - \widehat{V}^\lambda_{\text{LSE}}(S, \pi_\theta)$$

$$\leq \frac{1}{1+\epsilon}|\lambda|^\epsilon\nu - \frac{1}{\lambda}\sqrt{\frac{4|\lambda|^{1+\epsilon}\nu\log(2|\Pi_\theta|/\delta)}{n\exp(2\lambda\nu^{1/(1+\epsilon)})}} - \frac{4\log(2|\Pi_\theta|/\delta)}{3\lambda\exp(\lambda\nu^{1/(1+\epsilon)})n}. \tag{42}$$

Using lower bound on estimation error, Theorem D.3, and union bound (Shalev-Shwartz & Ben-David, 2014), with probability at least $1 - \delta$, the following upper bound holds on term $I_3$,

$$\widehat{V}^\lambda_{\text{LSE}}(S, \pi_{\widehat{\theta}}) - V(\pi_{\widehat{\theta}}) \leq \sup_{\pi_\theta \in \Pi_\theta} \widehat{V}^\lambda_{\text{LSE}}(S, \pi_\theta) - V(\pi_\theta)$$

$$\leq \frac{-1}{\lambda(1-\gamma)}\sqrt{\frac{4|\lambda|^{1+\epsilon}\nu\log(2|\Pi_\theta|/\delta)}{n\exp(2\lambda\nu^{1/(1+\epsilon)})}} - \frac{4\log(2|\Pi_\theta|/\delta)}{3(1-\gamma)\lambda\exp(\lambda\nu^{1/(1+\epsilon)})n}. \tag{43}$$

Note that the term $I_2$ is negative as the $\pi_{\widehat{\theta}}$ is the maximizer of the LSE estimator over $\Pi_\theta$. Combining equation 42 and equation 43 with equation 41, and applying the union bound, completes the proof.

$\square$

In the following, we provide a full version of Proposition 5.4.

> **Proposition 5.4 (Full Version).** *Given Assumption 5.1, for any $0 < \gamma < 1$, assuming $n \geq \frac{(2\nu + \frac{4}{3}\gamma)\log\frac{1}{\delta}}{\gamma^2\exp(2\nu^{1/(1+\epsilon)})}$ and setting $\lambda = -n^{-\zeta}$ for $\zeta \in \mathbb{R}^+$, then the overall convergence rate of the regret upper bound is $\max(O(n^{-1+\zeta}), O(n^{-\epsilon\zeta}), O(n^{(-\zeta\epsilon-1)/2}))$ for finite policy set.*

*Proof.* Without loss of generality, we can assume that $\lambda \geq -1$. Therefore, we have $|\lambda|^{1+\epsilon} \leq 1$ and $\nu^{1/(1+\epsilon)} \geq 0$, which results in $n \geq \frac{(2\nu + \frac{4}{3}\gamma)\log\frac{1}{\delta}}{\gamma^2\exp(-2\nu^{1/(1+\epsilon)})} \geq \frac{(2|\lambda|^{1+\epsilon}\nu + \frac{4}{3}\gamma)\log\frac{1}{\delta}}{\gamma^2\exp(2\lambda\nu^{1/(1+\epsilon)})}$. Using Theorem 5.3, with probability at least $(1-\delta)$, we have

$$\Re_\lambda(\pi_{\widehat{\theta}}, S)$$

$$\leq \frac{|\lambda|^\epsilon}{1+\epsilon}\nu - \frac{4(2-\gamma)}{3(1-\gamma)}\frac{\log\frac{4|\Pi_\theta|}{\delta}}{n\lambda\exp(\lambda\nu^{1/(1+\epsilon)})} - \frac{(2-\gamma)}{(1-\gamma)\lambda}\sqrt{\frac{4|\lambda|^{1+\epsilon}\nu\log\frac{4|\Pi_\theta|}{\delta}}{n\exp(2\lambda\nu^{1/(1+\epsilon)})}} \tag{44}$$

$$\leq \frac{|\lambda|^\epsilon}{1+\epsilon}\nu - \frac{4(2-\gamma)}{3(1-\gamma)}\frac{\log\frac{4|\Pi_\theta|}{\delta}}{n\lambda\exp(\lambda\nu^{1/(1+\epsilon)})} + \frac{(2-\gamma)}{(1-\gamma)\exp(\lambda\nu^{1/(1+\epsilon)})}\sqrt{\frac{4|\lambda|^\epsilon\nu\log\frac{4|\Pi_\theta|}{\delta}}{n}}. \tag{45}$$

Since $\lambda \geq -1$, we have $\exp(\lambda\nu^{1/(1+\epsilon)}) \geq \exp(-\nu^{1/(1+\epsilon)})$ (note that $\nu^{1/(1+\epsilon)} \geq 0$ and $-1 < \lambda < 0$). Replacing $\lambda$ with $\lambda^\star = -n^{-\zeta}$ and $\exp(\lambda\nu^{1/(1+\epsilon)})$ with $\exp(-\nu^{1/(1+\epsilon)})$, then we have the overall convergence rate of $\max(O(n^{-\epsilon\zeta}), O(n^{-1+\zeta}), O(n^{(-\zeta\epsilon-1)/2}))$. $\qquad\square$

### D.3. Proofs and Details of Bias and Variance

**Proposition 5.5 (Restated).** *Given Assumption 5.1, the following lower and upper bounds hold on the bias of the LSE estimator,*

$$\frac{(n-1)}{2n|\lambda|}\mathbb{V}(e^{\lambda w_\theta(A,X)R}) \leq \mathbb{B}(\widehat{V}^\lambda_{\text{LSE}}(S,\pi_\theta)) \leq \frac{1}{1+\epsilon}|\lambda|^\epsilon\nu + \frac{1}{2n\lambda}\mathbb{V}(e^{\lambda w_\theta(A,X)R}).$$

*Proof.* In the proof, for the sake of simplicity of notation, we consider $Y_\theta(A, X) = w_\theta(A, X)R$. For the lower bound we need to prove the following,

$$V(\pi_\theta) - \mathbb{E}\left[\widehat{V}^\lambda_{\text{LSE}}(S,\pi_\theta)\right] \geq \frac{n-1}{n|\lambda|}\mathbb{V}\left(e^{\lambda w_\theta(A,X)R}\right).$$

Setting $y_\theta(a_i, x_i) = r_i w_\theta(a_i, x_i)$, according to Lemma B.7 for b = 1, $f(x) = \log(x) + \frac{1}{2}x^2$ is concave. So we have,

$$\log\left(\frac{\sum_{i=1}^n e^{\lambda y_\theta(a_i,x_i)}}{n}\right) + \frac{1}{2}\left(\frac{\sum_{i=1}^n e^{\lambda y_\theta(a_i,x_i)}}{n}\right)^2 \geq \frac{1}{n}\left(\sum_{i=1}^n \log\left(e^{\lambda y_\theta(a_i,x_i)}\right) + \frac{1}{2}e^{2\lambda y_\theta(a_i,x_i)}\right)$$

$$= \frac{\lambda}{n}\sum_{i=1}^n y_\theta(a_i, x_i) + \frac{1}{2n}\sum_{i=1}^n e^{2\lambda y_\theta(a_i,x_i)}.$$

Hence,

$$\mathbb{E}\left[\frac{1}{\lambda}\log\left(\frac{\sum_{i=1}^n e^{\lambda y_\theta(a_i,x_i)}}{n}\right)\right]$$

$$\leq \mathbb{E}\left[\frac{1}{n}\sum_{i=1}^n y_\theta(a_i, x_i) + \frac{1}{2n\lambda}\sum_{i=1}^n e^{2\lambda y_\theta(a_i,x_i)} - \frac{1}{2\lambda}\left(\frac{\sum_{i=1}^n e^{\lambda y_\theta(a_i,x_i)}}{n}\right)^2\right]$$

$$= \mathbb{E}[Y_\theta(A,X)] + \frac{1}{2\lambda}\left(\mathbb{E}\left[e^{2\lambda Y_\theta(A,X)}\right] - \mathbb{E}\left[\left(\frac{\sum_{i=1}^n e^{\lambda y_\theta(a_i,x_i)}}{n}\right)^2\right]\right)$$

$$= \mathbb{E}[Y_\theta(A,X)] + \frac{1}{2\lambda}\left(\mathbb{E}\left[e^{2\lambda Y_\theta(A,X)}\right] - \mathbb{V}\left(\frac{\sum_{i=1}^n e^{\lambda y_\theta(a_i,x_i)}}{n}\right) - \mathbb{E}\left[\frac{\sum_{i=1}^n e^{\lambda y_\theta(a_i,x_i)}}{n}\right]^2\right)$$

$$= \mathbb{E}[Y_\theta(A,X)] + \frac{1}{2\lambda}\left(\mathbb{E}\left[e^{2\lambda Y_\theta(A,X)}\right] - \frac{1}{n}\mathbb{V}\left(e^{\lambda Y_\theta(A,X)}\right) - \mathbb{E}\left[e^{\lambda Y_\theta(A,X)}\right]^2\right)$$

$$= \mathbb{E}[Y_\theta(A,X)] + \frac{n-1}{2n\lambda}\mathbb{V}\left(e^{\lambda Y_\theta(A,X)}\right).$$

Note that $\mathbb{E}[Y_\theta(A, X)] = V(\pi_\theta)$. It completes the proof for the lower bound.

For the upper bound, we need to prove the following

$$\frac{1}{2n\lambda}\mathbb{V}(e^{\lambda w_\theta(A,X)R}) \geq \frac{1}{\lambda}\log\left(\mathbb{E}\left[e^{\lambda Y_\theta(A,X)}\right]\right) - \mathbb{E}\left[\widehat{V}^\lambda_{\text{LSE}}(S,\pi_\theta)\right]. \tag{46}$$

Note that, an upper bound 1 on $\frac{\sum_{i=1}^{n} e^{\lambda r_i w_\theta(a_i, x_i)}}{n}$ holds. Now, we have,

$$\mathbb{E}[\widehat{V}_{\text{LSE}}^\lambda(S, \pi_\theta)] = \frac{1}{\lambda}\mathbb{E}\left[\log\left(\frac{\sum_{i=1}^{n} e^{\lambda y_\theta(a_i, x_i)}}{n}\right)\right]$$

$$= \frac{1}{\lambda}\mathbb{E}\left[\log\left(\frac{\sum_{i=1}^{n} e^{\lambda y_\theta(a_i, x_i)}}{n}\right) + \frac{1}{2}\left(\frac{\sum_{i=1}^{n} e^{\lambda y_\theta(a_i, x_i)}}{n}\right)^2 - \frac{1}{2}\left(\frac{\sum_{i=1}^{n} e^{\lambda y_\theta(a_i, x_i)}}{n}\right)^2\right]$$

$$\geq \frac{1}{\lambda}\left(\log\left(\mathbb{E}\left[\frac{\sum_{i=1}^{n} e^{\lambda y_\theta(a_i, x_i)}}{n}\right]\right) + \frac{1}{2}\mathbb{E}\left[\frac{\sum_{i=1}^{n} e^{\lambda y_\theta(a_i, x_i)}}{n}\right]^2 - \frac{1}{2}\mathbb{E}\left[\left(\frac{\sum_{i=1}^{n} e^{\lambda y_\theta(a_i, x_i)}}{n}\right)^2\right]\right)$$

$$= \frac{1}{\lambda}\log\left(\mathbb{E}\left[e^{\lambda Y_\theta(A, X)}\right]\right) - \frac{1}{2\lambda}\mathbb{V}\left(\frac{\sum_{i=1}^{n} e^{\lambda y_\theta(a_i, x_i)}}{n}\right)$$

$$= \frac{1}{\lambda}\log\left(\mathbb{E}\left[e^{\lambda Y_\theta(A, X)}\right]\right) - \frac{1}{2n\lambda}\mathbb{V}\left(e^{\lambda Y_\theta(A, X)}\right),$$

where the first inequality is derived by applying Jensen inequality on function

$$\log\left(\frac{\sum_{i=1}^{n} e^{\lambda y_\theta(a_i, x_i)}}{n}\right) + \frac{1}{2}\left(\frac{\sum_{i=1}^{n} e^{\lambda y_\theta(a_i, x_i)}}{n}\right)^2,$$

which is concave based on Lemma B.7 for $b = 1$. Then, we have,

$$\frac{1}{\lambda}\log\left(\mathbb{E}\left[e^{\lambda Y_\theta(A, X)}\right]\right) - \mathbb{E}[\widehat{V}_{\text{LSE}}^\lambda(S, \pi_\theta)] \leq \frac{1}{2n\lambda}\mathbb{V}\left(e^{\lambda Y_\theta(A, X)}\right).$$

Finally, we combine the upper bound in equation 46 .

$$\mathbb{E}[\widehat{V}_{\text{LSE}}^\lambda(S, \pi_\theta)] - \frac{1}{\lambda}\log\left(\mathbb{E}\left[e^{\lambda Y_\theta(A, X)}\right]\right) \geq -\frac{1}{2n\lambda}\mathbb{V}\left(e^{\lambda Y_\theta(A, X)}\right),$$

and the upper bound in Lemma B.11,

$$\frac{1}{\lambda}\log\left(\mathbb{E}\left[e^{\lambda Y_\theta(A, X)}\right]\right) \geq \mathbb{E}[Y_\theta(A, X)] - \frac{1}{1+\epsilon}|\lambda|^\epsilon\mathbb{E}[|Y_\theta(A, X)|^{1+\epsilon}].$$

Therefore, we have,

$$\mathbb{E}[\widehat{V}_{\text{LSE}}^\lambda(S, \pi_\theta)] \geq \mathbb{E}[Y_\theta(A, X)] - \frac{1}{1+\epsilon}|\lambda|^\epsilon\mathbb{E}[|Y_\theta(A, X)|^{1+\epsilon}]$$
$$- \frac{1}{2n\lambda}\mathbb{V}\left(e^{\lambda w_\theta(A, X)R}\right). \tag{47}$$

It completes the proof. $\qquad\square$

---

**Proposition 5.7.** *Assume that $\mathbb{E}[(w_\theta(A, X)R)^2] \leq \nu_2$ (Assumption 5.1 for $\epsilon = 1$) holds. Then the variance of the LSE estimator with $\lambda < 0$, satisfies,*

$$\mathbb{V}(\widehat{V}_{\text{LSE}}^\lambda(S, \pi_\theta)) \leq \frac{1}{n}\mathbb{V}(w_\theta(A, X)R) \leq \frac{1}{n}\nu_2. \tag{48}$$

---

*Proof.* Let $Y_\theta(A, X) = w_\theta(A, X)R$ and $Y_\theta^{(c)} = Y_\theta(A, X) - \mathbb{E}[Y_\theta(A, X)]$ be the centered $Y_\theta(A, X)$. We have,

$$\widehat{V}_{\text{LSE}}^\lambda(S, \pi_\theta) = \frac{1}{\lambda}\ln\left(\frac{\sum_{i=1}^{n} e^{\lambda y_{i,\theta}}}{n}\right) = \frac{1}{\lambda}\ln\left(\frac{\sum_{i=1}^{n} e^{\lambda(y_{i,\theta}^{(c)} - m_\theta)}}{n}\right) = \frac{1}{\lambda}\ln\left(\frac{\sum_{i=1}^{n} e^{\lambda y_{i,\theta}^{(c)}}}{n}\right) + m_\theta$$

where $m_\theta = \mathbb{E}[Y_\theta(A, X)]$. Note that, we also have $\mathbb{V}(Y_\theta^{(c)}) = \mathbb{V}(Y_\theta)$.

Now, setting $Z = \frac{\sum_{i=1}^{n} e^{\lambda y_{i,\theta}^{(c)}}}{n}$, we have,

$$\mathbb{V}(\widehat{V}_{\mathrm{LSE}}^{\lambda}(S, \pi_\theta)) = \mathbb{V}\left(\frac{1}{\lambda} \log Z\right)$$

Furthermore, using Jensen's inequality for $\lambda < 0$, we have,

$$\frac{1}{\lambda} \log Z \leq \frac{\sum_{i=1}^{n} y_{i,\theta}^{(c)}}{n}.$$

Hence we have,

$$
\begin{aligned}
\mathbb{V}(\widehat{V}_{\mathrm{LSE}}^{\lambda}(S, \pi_\theta)) &= \mathbb{E}\left[\frac{1}{\lambda^2} \log^2 Z\right] - \left(\mathbb{E}\left[\frac{1}{\lambda} \log Z\right]\right)^2 \\
&\leq \mathbb{E}\left[\left(\frac{\sum_{i=1}^{n} y_{i,\theta}^{(c)}}{n}\right)^2\right] \\
&= \mathbb{V}\left(\frac{\sum_{i=1}^{n} y_{i,\theta}^{(c)}}{n}\right) + \mathbb{E}\left[\frac{\sum_{i=1}^{n} y_{i,\theta}^{(c)}}{n}\right]^2 \\
&= \mathbb{V}\left(\frac{\sum_{i=1}^{n} y_{i,\theta}^{(c)}}{n}\right) + \mathbb{E}\left[Y_\theta^{(c)}\right]^2 \\
&= \mathbb{V}\left(\frac{\sum_{i=1}^{n} y_{i,\theta}^{(c)}}{n}\right) + 0 \\
&= \frac{1}{n}\mathbb{V}(Y_\theta^{(c)}) = \frac{1}{n}\mathbb{V}(Y_\theta).
\end{aligned}
$$

It completes the proof. $\qquad\square$

For the moment of the LSE estimator, we provide the following upper bound.

> **Proposition D.4** (Moment bound). *Given Assumption 5.1, the following upper bound hold on the moment of the LSE estimator,*
> $$\mathbb{E}\left[\left|\frac{1}{\lambda} \log\left(\frac{\sum_{i=1}^{n} e^{\lambda w_\theta(a_i, x_i) r_i}}{n}\right)\right|^{1+\epsilon}\right] \leq \nu. \tag{49}$$

*Proof.* Suppose that $Z = \frac{\sum_{i=1}^{n} e^{\lambda r_i w_\theta(a_i, x_i)}}{n}$. Also set $y_{i,\theta}(a_i, x_i) = r_i(a_i, x_i) w_\theta(a_i, x_i)$. For negative $\lambda < 0$ and $Z > 0$, we have,

$$
\begin{aligned}
\widehat{V}_{\mathrm{LSE}}^{\lambda}(S, \pi_\theta) &= \frac{1}{\lambda} \log(Z) \\
&\leq \frac{\sum_{i=1}^{n} r_i w_\theta(a_i, x_i)}{n}.
\end{aligned}
$$

Since $\log Z < 0$ for $0 < Z < 1$, we have,

$$
\begin{aligned}
\mathbb{E}\left[\left|\frac{1}{\lambda} \log Z\right|^{1+\epsilon}\right] &\leq \mathbb{E}\left[\left|\frac{1}{n}\sum_{i=1}^{n} w_\theta(a_i, x_i) r_i\right|^{1+\epsilon}\right] \\
&\leq \mathbb{E}[|w_\theta(A, X) R|^{1+\epsilon}] \\
&\leq \nu,
\end{aligned}
$$

where the second inequality holds due to Jensen inequality. $\qquad\square$

## D.4. Proof and Details of Robustness of the LSE estimator: Noisy Reward

Using the functional derivative (Cardaliaguet et al., 2019), we can provide the following results.

> **Proposition D.5.** *Given Assumption 5.1, then the following holds,*
>
> $$\frac{1}{\lambda} \log(\mathbb{E}_{P_1}[\exp(\lambda w_\theta(A, X)R)]) - \frac{1}{\lambda} \log(\mathbb{E}_{P_2}[\exp(\lambda w_\theta(A, X)R)])$$
>
> $$\leq \frac{\mathbb{TV}_c(P_{R|X,A}, \widetilde{P}_{R|X,A})}{\lambda^2} \frac{\left( \exp(|\lambda|\tilde{\nu}^{1/(1+\epsilon)}) - \exp(|\lambda|\nu^{1/(1+\epsilon)}) \right)}{\tilde{\nu}^{1/(1+\epsilon)} - \nu^{1/(1+\epsilon)}}, \tag{50}$$
>
> *where $P_1 = P_X \otimes \pi_0(A|X) \otimes P_{R|X,A}$ and $P_2 = P_X \otimes \pi_0(A|X) \otimes \widetilde{P}_{R|X,A}$.*

*Proof.* We have that

$$\frac{1}{\lambda} \log(\mathbb{E}_{P_1}[\exp(\lambda w_\theta(A, X)R)]) - \frac{1}{\lambda} \log(\mathbb{E}_{P_2}[\exp(\lambda w_\theta(A, X)R)])$$

$$\stackrel{(a)}{=} \int_0^1 \int_{\mathbb{R} \times \mathcal{X} \times \mathcal{A}} \frac{\exp(\lambda w_\theta(A, X)R)}{|\lambda| \mathbb{E}_{P_\gamma}[\exp(\lambda w_\theta(A, X)R)]} P_X \otimes \pi_0(A|X)(P_{R|X,A} - \widetilde{P}_{R|X,A})(\mathrm{d}a\mathrm{d}x\mathrm{d}r)\mathrm{d}\gamma$$

$$\stackrel{(b)}{\leq} \frac{\mathbb{TV}_c(P_{R|X,A}, \widetilde{P}_{R|X,A})}{|\lambda|} \int_0^1 \frac{1}{\mathbb{E}_{P_\gamma}[\exp(\lambda w_\theta(A, X)R)]} \mathrm{d}\gamma \tag{51}$$

$$\stackrel{(c)}{\leq} \frac{\mathbb{TV}_c(P_{R|X,A}, \widetilde{P}_{R|X,A})}{\lambda^2} \frac{\left( \exp(|\lambda|\tilde{\nu}^{1/(1+\epsilon)}) - \exp(|\lambda|\nu^{1/(1+\epsilon)}) \right)}{\tilde{\nu}^{1/(1+\epsilon)} - \nu^{1/(1+\epsilon)}}.$$

where $P_\gamma = P_X \otimes \pi_0(A|X) \otimes \left( \widetilde{P}_{R|X,A} + \gamma(P_{R|X,A} - \widetilde{P}_{R|X,A}) \right)$, (a), (b) and (c) follow from the functional derivative and Lemma B.2 and Jensen-inequality. □

Combining Proposition D.5 with estimation error bounds, Theorem D.3 and Theorem D.2, we derive the upper bound on the regret under noisy reward scenario.

> **Theorem 5.9.** *Given Assumption 5.1, Assumption 5.8 and assuming $n \geq \frac{\left(2|\lambda|^{1+\epsilon}\nu + \frac{4}{3}\gamma\right) \log \frac{|\Pi_\theta|}{\delta}}{\gamma^2 \exp(2\lambda\nu^{1/(1+\epsilon)})}$, with probability at least $(1 - \delta)$, then there exists $\gamma \in (0, 1)$ such that the following upper bound holds on the regret of the LSE estimator under noisy reward logged data,*
>
> $$0 \leq \mathfrak{R}_\lambda(\pi_{\widehat{\theta}}(\widetilde{S}), \widetilde{S}) \leq \frac{|\lambda|^\epsilon}{1 + \epsilon} \nu$$
>
> $$- \frac{4(2 - \gamma)}{3(1 - \gamma)} \frac{\log \frac{4|\Pi_\theta|}{\delta}}{n\lambda \exp(\lambda\tilde{\nu}^{1/(1+\epsilon)})} - \frac{(2 - \gamma)}{(1 - \gamma)\lambda} \sqrt{\frac{4|\lambda|^{1+\epsilon}\tilde{\nu} \log \frac{4|\Pi_\theta|}{\delta}}{n \exp(2\lambda\tilde{\nu}^{1/(1+\epsilon)})}}$$
>
> $$+ \frac{2\mathbb{TV}_c(P_{R|X,A}, \widetilde{P}_{R|X,A})}{\lambda^2} \frac{\left( \exp(|\lambda|\tilde{\nu}^{1/(1+\epsilon)}) - \exp(|\lambda|\nu^{1/(1+\epsilon)}) \right)}{\tilde{\nu}^{1/(1+\epsilon)} - \nu^{1/(1+\epsilon)}},$$
>
> *where $\pi_{\widehat{\theta}}(\widetilde{S}) = \arg\max_{\pi_\theta \Pi_\theta} \widehat{V}_{\mathrm{LSE}}^\lambda(\pi_\theta, \widetilde{S})$.*

*Proof.* We have,

$$V(\pi_{\theta^*}) - V(\pi_{\widehat{\theta}}(\widetilde{S})) = \underbrace{V(\pi_{\theta^*}) - \widehat{V}_{\mathrm{LSE}}^\lambda(\widetilde{S}, \pi_{\theta^*})}_{I_1} + \underbrace{\widehat{V}_{\mathrm{LSE}}^\lambda(\widetilde{S}, \pi_{\theta^*}) - \widehat{V}_{\mathrm{LSE}}^\lambda(\widetilde{S}, \pi_{\widehat{\theta}}(\widetilde{S}))}_{I_2}$$

$$+ \underbrace{\widehat{V}_{\mathrm{LSE}}^\lambda(\widetilde{S}, \pi_{\widehat{\theta}}(\widetilde{S})) - V\left(\pi_{\widehat{\theta}}(\widetilde{S})\right)}_{I_3}. \tag{52}$$

Using upper bound on estimation error, Theorem D.2, and union bound (Shalev-Shwartz & Ben-David, 2014), with probability at least $1 - \delta$, the following upper bound holds on term $I_1$,

$$
\begin{aligned}
V(\pi_{\theta^*}) &- \widehat{V}_{\mathrm{LSE}}^{\lambda}(\widetilde{S}, \pi_{\theta^*}) \\
&= V(\pi_{\theta^*}) - \frac{1}{\lambda} \log(\mathbb{E}_{P_1}[\exp(\lambda w_\theta(A, X)R)]) \\
&\quad + \frac{1}{\lambda} \log(\mathbb{E}_{P_1}[\exp(\lambda w_\theta(A, X)R)]) - \frac{1}{\lambda} \log(\mathbb{E}_{P_2}[\exp(\lambda w_\theta(A, X)R)]) \\
&\quad + \frac{1}{\lambda} \log(\mathbb{E}_{P_2}[\exp(\lambda w_\theta(A, X)R)]) - \widehat{V}_{\mathrm{LSE}}^{\lambda}(\widetilde{S}, \pi_{\theta^*}) \\
&\leq \frac{1}{1+\epsilon} |\lambda|^\epsilon \nu \\
&\quad + \frac{\mathbb{TV}_c(P_{R|X,A}, \widetilde{P}_{R|X,A})}{\lambda^2} \frac{\left( \exp(|\lambda|\tilde{\nu}^{1/(1+\epsilon)}) - \exp(|\lambda|\nu^{1/(1+\epsilon)}) \right)}{\tilde{\nu}^{1/(1+\epsilon)} - \nu^{1/(1+\epsilon)}} \\
&\quad - \frac{1}{\lambda} \sqrt{\frac{4|\lambda|^{1+\epsilon}\widetilde{\nu} \log(2|\Pi_\theta|/\delta)}{n \exp(2\lambda\widetilde{\nu}^{1/(1+\epsilon)})}} - \frac{4 \log(2|\Pi_\theta|/\delta)}{3\lambda \exp(\lambda\nu^{1/(1+\epsilon)})n}.
\end{aligned}
\tag{53}
$$

Using lower bound on estimation error, Theorem D.3, and union bound (Shalev-Shwartz & Ben-David, 2014), with probability at least $1 - \delta$, the following upper bound holds on term $I_3$,

$$
\begin{aligned}
\widehat{V}_{\mathrm{LSE}}^{\lambda}(\widetilde{S}, &\pi_{\widehat{\theta}}(\widetilde{S})) - V(\pi_{\widehat{\theta}}(\widetilde{S})) \\
&= \widehat{V}_{\mathrm{LSE}}^{\lambda}(\widetilde{S}, \pi_{\widehat{\theta}}(\widetilde{S})) - \frac{1}{\lambda} \log(\mathbb{E}_{P_2}[\exp(\lambda w_{\widehat{\theta}}(A, X)R)]) \\
&\quad + \frac{1}{\lambda} \log(\mathbb{E}_{P_2}[\exp(\lambda w_{\widehat{\theta}}(A, X)R)]) - \frac{1}{\lambda} \log(\mathbb{E}_{P_1}[\exp(\lambda w_{\widehat{\theta}}(A, X)R)]) \\
&\quad + \frac{1}{\lambda} \log(\mathbb{E}_{P_1}[\exp(\lambda w_{\widehat{\theta}}(A, X)R)]) - V(\pi_{\widehat{\theta}}(\widetilde{S})) \\
&\leq \frac{-1}{\lambda(1-\gamma)} \sqrt{\frac{4|\lambda|^{1+\epsilon}\widetilde{\nu} \log(2|\Pi_\theta|/\delta)}{n \exp(2\lambda\widetilde{\nu}^{1/(1+\epsilon)})}} - \frac{4 \log(2|\Pi_\theta|/\delta)}{3(1-\gamma)\lambda \exp(\lambda\widetilde{\nu}^{1/(1+\epsilon)})n} \\
&\quad + \frac{\mathbb{TV}_c(P_{R|X,A}, \widetilde{P}_{R|X,A})}{\lambda^2} \frac{\left( \exp(|\lambda|\tilde{\nu}^{1/(1+\epsilon)}) - \exp(|\lambda|\nu^{1/(1+\epsilon)}) \right)}{\tilde{\nu}^{1/(1+\epsilon)} - \nu^{1/(1+\epsilon)}}.
\end{aligned}
\tag{54}
$$

Note that the term $I_2$ is negative as the $\pi_{\widehat{\theta}}(\widetilde{S})$ is the maximizer of the LSE estimator over $\Pi_\theta$. Combining equation 53 and equation 54 with equation 52, and applying the union bound, completes the proof. $\square$

### D.5. Data-driven selection of $\lambda$

In Theorem 5.3, we assume a fixed value of $\lambda$. However, it is often important in practical applications to have a method for adjusting $\lambda$ dynamically based on the data.

Recall the following regret bound proposed by Theorem 5.3,

$$
\begin{aligned}
\mathfrak{R}_\lambda(\pi_{\widehat{\theta}}, S) &\leq \frac{|\lambda|^\epsilon}{1+\epsilon} \nu + \frac{4(2-\gamma)}{3(1-\gamma)} \frac{\log \frac{4|\Pi_\theta|}{\delta} \exp(|\lambda|\nu^{1/(1+\epsilon)})}{n|\lambda|} \\
&\quad + \frac{(2-\gamma)}{(1-\gamma)|\lambda|} \sqrt{\frac{4|\lambda|^{1+\epsilon}\nu \log \frac{4|\Pi_\theta|}{\delta} \exp(2|\lambda|\nu^{1/(1+\epsilon)})}{n}}
\end{aligned}
$$

which is true for any $\gamma$. If $\gamma$ tends to zero, we have,

$$
\mathfrak{R}_\lambda(\pi_{\widehat{\theta}}, S) \leq \frac{|\lambda|^\epsilon}{1+\epsilon} \nu + \frac{8}{3} \frac{\exp(|\lambda|\nu^{1/(1+\epsilon)}) \log \frac{4|\Pi_\theta|}{\delta}}{n|\lambda|} + \frac{2}{|\lambda|} \sqrt{\frac{4|\lambda|^{1+\epsilon}\nu \log \frac{4|\Pi_\theta|}{\delta} \exp(2|\lambda|\nu^{1/(1+\epsilon)})}{n}}.
$$

Let the upper bound be $U_R$ and $x = \sqrt{\nu |\lambda|^{1+\epsilon}}$. We have,

$$
\begin{aligned}
U_R &= \frac{x^{\frac{2\epsilon}{1+\epsilon}}}{(1+\epsilon)\nu^{\frac{\epsilon}{1+\epsilon}}}\nu + \frac{8}{3}\frac{\nu^{\frac{1}{1+\epsilon}}\exp(x^{\frac{2}{1+\epsilon}})\log\frac{4|\Pi_\theta|}{\delta}}{nx^{\frac{2}{1+\epsilon}}} + 2\sqrt{\frac{4\nu\log\frac{4|\Pi_\theta|}{\delta}\exp(2x^{\frac{2}{1+\epsilon}})}{n(x^{\frac{2}{1+\epsilon}}\nu^{\frac{-1}{1+\epsilon}})^{1-\epsilon}}} \\
&= \nu^{\frac{1}{1+\epsilon}}\left(\frac{x^{\frac{2\epsilon}{1+\epsilon}}}{(1+\epsilon)} + \frac{8}{3}\frac{\exp(x^{\frac{2}{1+\epsilon}})\log\frac{4|\Pi_\theta|}{\delta}}{nx^{\frac{2}{1+\epsilon}}} + 2\sqrt{\frac{4\log\frac{4|\Pi_\theta|}{\delta}\exp(2x^{\frac{2}{1+\epsilon}})}{nx^{\frac{2(1-\epsilon)}{1+\epsilon}}}}\right).
\end{aligned}
\tag{55}
$$

Finally, we assume that $|\lambda| \le 1$ and bound and replace the exponential $\exp(x^{\frac{2}{1+\epsilon}})$ by $e$. Minimizing the upper bound in equation 55, we derive the following optimum $\lambda$ for the optimization of the upper bound in Theorem 5.3,

$$
\lambda_{\mathrm{D}} = \max\left\{-f(\epsilon) \cdot \left(\frac{\ln\left(\frac{1}{\delta}\right)}{vn}\right)^{\frac{1}{1+\epsilon}}, -1\right\},
\tag{56}
$$

where $f(\epsilon) = \left(\frac{e(1+\epsilon)}{\epsilon}\left(1 - \epsilon + \sqrt{(1-\epsilon)^2 + \frac{8\epsilon}{3e(1+\epsilon)}}\right)\right)^{\frac{2}{1+\epsilon}}$. Note that, we can compute the empirical value of $\nu$ based on the available LBF dataset,

$$
\hat{\nu} = \frac{1}{n}\sum_{i=1}^{n}\left(w_\theta(a_i, x_i)r_i\right)^{1+\epsilon}.
\tag{57}
$$

Using empirical $\hat{\nu}$ in equation 56, we derive the value for data driven $\lambda$. Note that, in our experiments, we consider $\epsilon = 1$.

**Data-driven $\lambda$ under noisy reward:** As discussed in Section 5.4, we can also derive a data-driven under noisy reward for asymptotic regime, where $n \to \infty$. For this purpose, we need to solve the following objective function,

$$
\underset{\lambda \in (-\infty, 0)}{\arg\min} \frac{|\lambda|^\epsilon}{1+\epsilon}\nu + \frac{2\mathbb{TV}_{\mathrm{c}}(P_{R|X,A}, \widetilde{P}_{R|X,A})}{\lambda^2}\frac{\exp(|\lambda|\tilde{\nu}^{1/(1+\epsilon)}) - \exp(|\lambda|\nu^{1/(1+\epsilon)})}{\tilde{\nu}^{1/(1+\epsilon)} - \nu^{1/(1+\epsilon)}},
$$

To solve this objective, we use the following estimation,

$$
\frac{e^x - e^y}{x - y} \approx e^x
$$

which is true when $x$ and $y$ are close. Using this estimation, we replace the term $\frac{\exp(|\lambda|\tilde{\nu}^{1/(1+\epsilon)}) - \exp(|\lambda|\nu^{1/(1+\epsilon)})}{\tilde{\nu}^{1/(1+\epsilon)} - \nu^{1/(1+\epsilon)}}$ with $|\lambda|\exp(|\lambda|\tilde{\nu}^{1/(1+\epsilon)})$, and estimate $\nu$ with $\tilde{\nu}$, which is observed from the data and we get a simplified objective function as,

$$
\lambda_{\mathrm{ND}} := \underset{\lambda \in (-\infty, 0)}{\arg\min} \frac{|\lambda|^\epsilon}{1+\epsilon}\tilde{\nu} + \frac{2\mathbb{TV}_{\mathrm{c}}(P_{R|X,A}, \widetilde{P}_{R|X,A})}{|\lambda|}\exp(|\lambda|\tilde{\nu}^{1/(1+\epsilon)}),
\tag{58}
$$

We further studied the performance of data-driven $\lambda$ selection in App. G.7.

### D.6. PAC-Bayesian Discussion

In this section, we explore the PAC-Bayesian approach and its application in extending our previous results. Given that the methodology for deriving these results closely resembles our earlier approach, we will outline the key steps in the derivation process rather than providing a full detailed analysis.

For this purpose, we introduce several additional definitions inspired by Gabbianelli et al. (2023). For the PAC-Bayesian approach, we focus on randomized algorithms that output a distribution $\widehat{Q}_n \in \mathcal{P}(\Pi_\theta)$ over policies. Our interest lies in performance guarantees that satisfy two conditions: (1) they hold in expectation with respect to the random selection of $\widehat{\pi}_n \sim \widehat{Q}_n$, and (2) they maintain high probability with respect to the realization of the LBF dataset. For this purpose, we define the following integral forms of our previous formulation,

$$V(Q) = \int V(\pi_\theta)\mathrm{d}Q(\pi_\theta),$$

$$\widehat{\mathrm{V}}_{\mathrm{LSE}}^\lambda(S, Q) = \int \widehat{\mathrm{V}}_{\mathrm{LSE}}^\lambda(S, \pi)\mathrm{d}Q(\pi), \tag{59}$$

$$\mathfrak{R}(Q, S) = \int \mathfrak{R}(\pi, S)\mathrm{d}Q(\pi).$$

These expressions capture relevant quantities evaluated in expectation under the distribution $\mathbb{Q} \in \mathcal{P}(\Pi_\theta)$ where $\mathcal{P}(\Pi_\theta)$ is the set of distributions over policy set. Let $\mathbb{P} \in \mathcal{P}(\Pi_\theta)$ a prior distribution over policy class.

We can relax the uniform assumption on $(1 + \epsilon)$-th moment Assumption 5.1, as follows,

**Assumption D.6.** The reward distribution $P_{R|X,A}$ and $P_X \otimes \pi_0(A|X)$ are such that for a posterior distribution $Q$ over the set of policies $\Pi_\theta$ and some $\epsilon \in (0, 1]$, the $(1 + \epsilon)$-th moment of the weighted reward is bounded,

$$\mathbb{E}_{\pi_\theta \sim \mathbb{Q}}\mathbb{E}_{P_X \otimes \pi_0(A|X) \otimes P_{R|X,A}}\left[\left(w_\theta(A, X)R\right)^{1+\epsilon}\right] \leq \nu_q. \tag{60}$$

In order to derive the upper bound on regret, we need to derive the upper and lower PAC-Bayesian bound on estimation error. For this purpose, we can apply the following bound from (Tolstikhin & Seldin, 2013, Theorem 2) which holds with probability $1 - \delta$ and for a fixed $c_1 > 1$,

$$\left| \int_{\pi_\theta \sim \mathbb{Q}} \mathbb{E}[\exp(\lambda Y_\theta(A, X))] - \int_{\pi_\theta \sim \mathbb{Q}} \frac{1}{n}\sum_{i=1}^n \exp(\lambda Y_\theta(a_i, x_i)) \right|$$

$$\leq (1 + c_1)\sqrt{\frac{(e - 2)\mathbb{E}_Q[\mathbb{V}(\exp(\lambda Y_\theta(A, X)))]\left(\mathrm{KL}(\mathbb{Q}\|\mathbb{P}) + \ln\frac{\nu_1}{\delta}\right)}{n}}, \tag{61}$$

where $Y_\theta(a_i, x_i) = w_\theta(a_i, x_i)r_i$ and

$$\nu_1 = \left\lceil \frac{1}{\ln c_1}\ln\left(\sqrt{\frac{(e - 2)n}{4\ln(1/\delta)}}\right)\right\rceil + 1. \tag{62}$$

Similar to Theorem D.3 and Theorem D.2, we can replace $\mathbb{E}_Q[\mathbb{V}(\exp(\lambda Y_\theta(A, X)))]$ with $|\lambda|^{1+\epsilon}\mathbb{E}[Y_\theta(A, X)^{1+\epsilon}]$. Given Assumption D.6, the following upper bounds holds on estimation error,

$$\mathbb{E}_{\pi_\theta \sim \mathbb{Q}}[\mathrm{Est}_\lambda(\pi_\theta)] \leq \frac{1}{1 + \epsilon}|\lambda|^\epsilon\nu_q - \frac{(1 + c_1)}{\lambda}\sqrt{\frac{(e - 2)|\lambda|^{1+\epsilon}\nu_q\left(\mathrm{KL}(\mathbb{Q}\|\mathbb{P}) + \ln\frac{2\nu_1}{\delta}\right)}{\exp(2\lambda\nu_q^{1/(1+\epsilon)})n}}. \tag{63}$$

For lower bound, given Assumption D.6, there exists $n_0$ such that for $n \geq n_0$ and $\gamma_q \in (0, 1)$ the following holds with probability $(1 - \delta)$,

$$\mathbb{E}_{\pi_\theta \sim \mathbb{Q}}[\mathrm{Est}_\lambda(\pi_\theta)] \geq \frac{(1 + c_1)}{\lambda(1 - \gamma_q)}\sqrt{\frac{(e - 2)|\lambda|^{1+\epsilon}\nu_q\left(\mathrm{KL}(\mathbb{Q}\|\mathbb{P}) + \ln\frac{2\nu_1}{\delta}\right)}{\exp(2\lambda\nu_q^{1/(1+\epsilon)})n}}. \tag{64}$$

Combining equation 64 and equation 63, we can derive an upper bound on $\mathfrak{R}(\widehat{Q}, S)$ in a similar approach to Theorem 5.3 under Assumption D.6 and assuming $\widehat{Q}_n := \arg\max_{Q \in \mathcal{P}(\Pi_\theta)} \widehat{\mathrm{V}}_{\mathrm{LSE}}^\lambda(S, Q)$.

$$\mathfrak{R}(\widehat{Q}_n, S) \leq \frac{1}{1 + \epsilon}|\lambda|^\epsilon\nu_q - \frac{(1 + c_1)(2 - \gamma_q)}{(1 - \gamma_q)\lambda}\sqrt{\frac{(e - 2)|\lambda|^{1+\epsilon}\nu_q\left(\mathrm{KL}(\mathbb{Q}\|\mathbb{P}) + \ln\frac{2\nu_1}{\delta}\right)}{\exp(2\lambda\nu_q^{1/(1+\epsilon)})n}}. \tag{65}$$

Note that, the PAC-Bayesian approach in (London & Sandler, 2019; Sakhi et al., 2023; 2024; Aouali et al., 2023) is different. However, their PAC-Bayesian model can also be applied to our LSE estimator.

### D.7. Sub-Gaussian Discussion

In this section, we investigate the sub-Gaussianity concentration inequality (estimation error) under LSE estimator.

We first present the following general result.

**Proposition D.7.** *Given Assumption 5.1, for any $0 < \gamma < 1$, assuming $n \geq \max\left( \frac{\left(2\nu + \frac{4}{3}\gamma\right) \log \frac{1}{\delta}}{\gamma^2 \exp(2\nu^{1/(1+\epsilon)})}, \frac{\log \frac{2}{\delta}}{\nu} \right)$ and setting*

$$\lambda = -\left( \frac{\log \frac{2}{\delta}}{\nu n} \right)^{\frac{1}{1+\epsilon}},$$

*then with a probability at least $(1 - \delta)$ for $\delta \in (0, 1)$, the absolute of estimation error of the LSE estimator satisfies for a fixed $\pi_\theta \in \Pi_\theta$,*

$$\left| \mathrm{Est}_\lambda(\pi_\theta) \right| \leq \left( \frac{1}{1+\epsilon} + \frac{4}{(1-\gamma) \exp(\nu^{1/(1+\epsilon)})} \right) \nu^{\frac{1}{1+\epsilon}} \left( \frac{\log \frac{2}{\delta}}{n} \right)^{\frac{\epsilon}{1+\epsilon}}.$$

*Proof.* Choosing $n \geq \frac{2 \log \frac{2}{\delta}}{\nu}$, we have $\lambda \geq -1$, $|\lambda|^{1+\epsilon} \leq 1$ and $\nu \geq 0$, which results in $n \geq \frac{\left(2\nu + \frac{4}{3}\gamma\right) \log \frac{1}{\delta}}{\gamma^2 \exp(2\nu^{1+\epsilon})} \geq \frac{\left(2|\lambda|^{1+\epsilon}\nu + \frac{4}{3}\gamma\right) \log \frac{1}{\delta}}{\gamma^2 \exp(2\lambda\nu^{1+\epsilon})}$. Using Theorem D.2 and Theorem D.3, we have with probability at least $(1 - \delta)$,

$$\left| \mathrm{Est}_\lambda(\pi_\theta) \right|$$
$$\leq \frac{1}{1+\epsilon} |\lambda|^\epsilon \nu - \frac{1}{\lambda(1-\gamma)} \sqrt{\frac{4|\lambda|^{1+\epsilon}\nu \log(2/\delta)}{n \exp(2\lambda\nu^{1/(1+\epsilon)})}} - \frac{4 \log(2/\delta)}{3(1-\gamma)\lambda \exp(\lambda\nu^{1/(1+\epsilon)})n} \tag{66}$$

Since $\lambda \geq -1$, we have $\exp(\lambda\nu^{1+\epsilon}) \geq \exp(-\nu^{1+\epsilon})$ (note that $\nu \geq 0$). Replacing $\lambda$ with $\lambda^\star = -\left( \frac{\log \frac{2}{\delta}}{\nu n} \right)^{\frac{1}{1+\epsilon}}$ and $\exp(\lambda\nu^{1+\epsilon})$ with $\exp(\nu^{1+\epsilon})$, we have,

$$\left| \mathrm{Est}_\lambda(\pi_\theta) \right| \leq \frac{\nu^{\frac{1}{1+\epsilon}}}{1+\epsilon} \left( \frac{\log \frac{2}{\delta}}{n} \right)^{\frac{\epsilon}{1+\epsilon}} + \frac{4\nu^{\frac{1}{1+\epsilon}}}{3(1-\gamma) \exp(\nu^{1/(1+\epsilon)})} \left( \frac{\log \frac{2}{\delta}}{n} \right)^{\frac{\epsilon}{1+\epsilon}}$$
$$+ \frac{2\nu^{\frac{1}{1+\epsilon}}}{(1-\gamma) \exp(\nu^{1/(1+\epsilon)})} \left( \frac{\log \frac{2}{\delta}}{n} \right)^{\frac{\epsilon}{1+\epsilon}}$$
$$\leq \left( \frac{1}{1+\epsilon} + \frac{4}{(1-\gamma) \exp(\nu^{1/(1+\epsilon)})} \right) \nu^{\frac{1}{1+\epsilon}} \left( \frac{\log \frac{2}{\delta}}{n} \right)^{\frac{\epsilon}{1+\epsilon}}$$

with a probability at least $(1 - \delta)$. As the upper bound on absolute value of the estimation error holds. $\qquad\square$

*Remark* D.8. Suppose that the second moment of weighted reward is bounded which is equal to Assumption 5.1 with $\epsilon = 1$. As a result, using Proposition D.7 for $\epsilon = 1$, we can establish a concentration inequality (estimation bound) for the LSE even in cases where the rewards are unbounded.

### D.8. Implicit Shrinkage

Su et al. (2020) proposed the optimistic shrinkage where the weights are less than the main weights of the IPS estimator. Other transformations of weights in other estimators are also lower bound to the main weights of IPS estimators. For example, in TR-IPS, we have $\min(M, w_\theta(a, x))$ which is a lower bound to $w_\theta(a, x)$. Our LSE estimator is also a lower bound to the IPS estimator,

$$\frac{1}{\lambda} \log\left( \frac{1}{n} \sum_{i=1}^n \exp(\lambda w_\theta(a_i, x_i) r_i) \right) \leq \frac{1}{n} \sum_{i=1}^n w_\theta(a_i, x_i) r_i, \tag{67}$$

which can be interpreted as implicit shrinkage. Furthermore, note that the LSE is not separable with respect to the samples, so instead of per-sample shrinkage, we investigate LSE's shrinkage effect on the entire output, which is the estimated

average reward. It can be derived by simple calculation that for $\lambda < 0$,

$$\frac{1}{n}\sum_{i=1}^{n} y_i - \frac{1}{\lambda}\log\left(\frac{\sum_{i=1}^{n} e^{\lambda y_i}}{n}\right) = \frac{1}{|\lambda|}D_{\mathrm{KL}}\left(\frac{1}{n}\mathbb{1}_n, \mathrm{softmax}(\lambda y_i)\right),$$

where $\mathbb{1}_n$ is all-one vector with size $n$. Hereby we see that LSE shrinks the Monte-Carlo average by the KL-divergence between the uniform vector and softmax of the samples (with temperature $1/\lambda$). This way, when outlier values or large values are out of the normal range of the data are observed, the amount of shrinkage increases. Also when the variance is high or we have heavy-tailed distributions, the softmax of $\lambda y_i$ goes further from the uniform vector and more shrinkage is applied.

### D.9. Robustness Discussion

In this section, we study the convergence rate LSE estimator under $m$ outlier samples. Without loss of generality, assume that $\widetilde{P}_{R|X,A} = \frac{n}{n+m}P_{R|X,A} + \frac{m}{n+m}P_{RO}$, where $P_{RO}$ is the distribution of outlier samples. Then, we have

$$\mathbb{TV}_c(P_{R|X,A}, \widetilde{P}_{R|X,A}) = \int_{\mathbb{R}} |P_{R|X,A}(r) - \widetilde{P}_{R|X,A}(r)|\mathrm{d}r \tag{68}$$

$$= \frac{m}{n+m}\int_{\mathbb{R}} |P_{R|X,A}(r) - P_{RO}(r)|\mathrm{d}r \tag{69}$$

$$\leq \frac{m}{n+m}\mathbb{TV}_c(P_{R|X,A}, P_{RO}) \tag{70}$$

$$\leq \frac{2m}{n+m}. \tag{71}$$

Note that $D(\tilde{\nu}, \nu) = \frac{\exp(|\lambda|\tilde{\nu}^{1/(1+\epsilon)}) - \exp(|\lambda|\nu^{1/(1+\epsilon)})}{\tilde{\nu}^{1/(1+\epsilon)} - \nu^{1/(1+\epsilon)}} \sim O(|\lambda|)$. Therefore, we have,

$$\frac{2\mathbb{TV}_c(P_{R|X,A}, \widetilde{P}_{R|X,A})}{\lambda^2}D(\tilde{\nu}, \nu) \sim O\left(\frac{2m}{|\lambda|(n+m)}\right). \tag{72}$$

Choosing $\lambda = O(n^{-1/(1+\epsilon)})$, we have the overall convergence rate of $\max\left(O(n^{-\epsilon/(1+\epsilon)}), O(\frac{2mn^{1/(1+\epsilon)}}{n+m})\right)$. Note that, for $m = 1$ and large enough $n$, we have the convergence rate of $O(n^{-\epsilon/(1+\epsilon)})$.

### D.10. Relaxing the lower bound on $n$ in the regret bound

We can relax the lower bound on $n$, by selecting $\gamma$ appropriately. The lower bound on $n$, given $\gamma > 0$, becomes equivalent to the following inequality of $\gamma$,

$$\gamma \geq \frac{B + \sqrt{B^2 + 4An}}{2n}$$

, where $A = \frac{2|\lambda|^{1+\epsilon}\nu\log\frac{|\Pi_\theta|}{\delta}}{\exp(2\lambda\nu^{1/(1+\epsilon)})}$ and $B = \frac{\frac{4}{3}\log\frac{|\Pi_\theta|}{\delta}}{\exp(2\lambda\nu^{1/(1+\epsilon)})}$. If,

$$1 \geq \frac{B + \sqrt{B^2 + 4An}}{2n}$$

existence of such a $\gamma$ is guaranteed. This is equivalent to,

$$n \geq \frac{\left(2|\lambda|^{1+\epsilon}\nu + \frac{4}{3}\right)\log\frac{|\Pi_\theta|}{\delta}}{\exp(2\lambda\nu^{1/(1+\epsilon)})}$$

Hence, setting $\gamma = \frac{B+\sqrt{B^2+4An}}{2n}$ which is $O\left(\frac{1}{\sqrt{n}}\right)$, results in a lower bound on $n$ which is independent of $\gamma$ and is weaker than the original lower bound for any $\gamma$. This only changes the regret bound at most by a constant factor since $2 - \gamma \leq 2$ and $\frac{1}{1-\gamma} = O\left(\frac{1}{1-\frac{1}{\sqrt{n}}}\right) = O\left(\frac{\sqrt{n}}{\sqrt{n}-1}\right) = O(1)$, and remove the dependence of the bound on the extra parameter $\gamma$.

# E. Robustness of the LSE estimator: Estimated Propensity Scores

In this section, we study the robustness of the LSE estimator with respect to estimated (noisy) propensity scores.

To model the estimated propensity scores, we consider $\widehat{\pi}_0(a|x)$ as the noisy version of the logging policy $\pi_0(a|x)$. Similarly, we define $\widehat{V}_{\mathrm{LSE}}^{\lambda}(\widehat{S}, \pi_\theta)$ for the LSE estimator on the noisy data samples $\widehat{S}$, with estimated propensity scores. In this section, we made the following definitions.

**Definition E.1** (Discrepancy metric). We define the general discrepancy metric between $\widehat{w}_\theta(A, X)R$ and $w_\theta(A, X)R$ with bounded $1 + \epsilon$-th moment as,

$$d_{\pi_0}(\widehat{w}_\theta(A, X)R, w_\theta(A, X)R) := \mathbb{E}\big[\big(\widehat{w}_\theta(A, X) - w_\theta(A, X)\big)R\big]. \tag{73}$$

**Definition E.2.** The log-sum error of the noisy (or estimated) propensity score $\widehat{\pi}_0(a|x)$ is defined as

$$\Delta_{\pi_\theta}(\widehat{\pi}_0, \pi_0) = \frac{1}{\lambda} \log \mathbb{E}_{P_1}[\exp(\lambda \widehat{w}_\theta(A, X)R)] - \frac{1}{\lambda} \log \mathbb{E}_{P_1}[\exp(\lambda w_\theta(A, X)R)]. \tag{74}$$

where $\widehat{w}_\theta(A, X) = \frac{\pi_\theta(A|X)}{\widehat{\pi}_0(A|X)}$ and where $P_1 = P_X \otimes \pi_0(A|X) \otimes P_{R|X,A}$.

Definition E.2 captures a notion of bias in the noise that is applied to the propensity score. It indicates the change in the population form of the LSE estimator. Similarly, for the Monte Carlo estimator, the change in the expected value shows the bias of the noise, and for additive noise, the zero-mean assumption ensures that in expectation, the noisy value is the same as the original value. For the LSE estimator instead, we require the exponential forms to be close to each other. It is also inspired by influence function definition and robust statistic (Ronchetti & Huber, 2009; Christmann & Steinwart, 2004).

We made the following assumption on estimated propensity scores.

**Assumption E.3** (Bounded moment under noise). The reward function $r(A, X)$ and $P_X$ are such that for all learning policy $\pi_\theta(A|X) \in \Pi_\theta$, the moment of weighted reward is bounded under estimated propensity score scenario, $\mathbb{E}_{P_X \otimes \pi_0(A|X) \otimes P_{R|X,A}}[(\widehat{w}_\theta(A, X)R)^{1+\epsilon}] \leq \widehat{\nu}$.

*Remark* E.4. Under Assumption E.3 and Assumption 5.1 and using Lemma B.10, it can be shown that the discrepancy metric in Definition E.1 is bounded,

$$-\nu^{1/(1+\epsilon)} \leq d_{\pi_0}(\widehat{w}_\theta(A, X)R, w_\theta(A, X)R) \leq \widehat{\nu}^{1/(1+\epsilon)}. \tag{75}$$

We define the achieved policy under the estimated propensity scores as

$$\pi_{\widehat{\theta}}(S) := \arg\max_{\pi_\theta \in \Pi_\Theta} \widehat{V}_{\mathrm{LSE}}^{\lambda}(\widehat{S}, \pi_\theta).$$

In order to derive an upper bound on the regret under the noisy propensity score, the following results are needed.

---

**Proposition E.5.** *Given Assumption 5.1 and Assumption E.3, the following upper and lower bound hold on* $\Delta_{\pi_\theta}(\widehat{\pi}_0, \pi_0)$,

$$d_{\pi_0}(w_\theta(A, X)R, \widehat{w}_\theta(A, X)R) - \frac{|\lambda|^\epsilon \widehat{\nu}}{1+\epsilon} \leq \Delta_{\pi_\theta}(\widehat{\pi}_0, \pi_0),$$

*and,*

$$\Delta_{\pi_\theta}(\widehat{\pi}_0, \pi_0) \leq \frac{|\lambda|^\epsilon \nu}{1+\epsilon} + d_{\pi_0}(\widehat{w}_\theta(A, X)R, w_\theta(A, X)R).$$

---

*Proof.* It follows directly from applying Lemma B.11 to $\frac{1}{\lambda} \log \mathbb{E}_{P_1}[\exp(\lambda \widehat{w}_\theta(A, X)R)]$ and $\frac{1}{\lambda} \log \mathbb{E}_{P_1}[\exp(\lambda w_\theta(A, X)R)]$ and combining the lower and upper bounds. Then, we have,

$$\mathbb{E}\big[\big(w_\theta(A, X) - \widehat{w}_\theta(A, X)\big)R\big] - \frac{|\lambda|^\epsilon \widehat{\nu}}{1+\epsilon} \leq \Delta_{\pi_\theta}(\widehat{\pi}_0, \pi_0) \leq \frac{|\lambda|^\epsilon \nu}{1+\epsilon} + \mathbb{E}\big[\big(\widehat{w}_\theta(A, X) - w_\theta(A, X)\big)R\big].$$

$\square$

**Proposition E.6.** *Given Assumption E.3, and assuming* $n > \frac{\frac{4}{3}\mu_{\min}+4}{\mu_{\min}^2}\log\frac{4}{\delta}$ *where* $\mu_{\min} = \min\left(e^{\lambda\nu^{1/(1+\epsilon)}}, e^{\lambda\widehat{\nu}^{1/(1+\epsilon)}}\right)$, *then with probability at least* $(1-\delta)$ *for a fixed* $\pi_\theta \in \Pi_\theta$, *we have,*

$$\left|\widehat{V}_{\mathrm{LSE}}^\lambda(\widehat{S}, \pi_\theta) - \widehat{V}_{\mathrm{LSE}}^\lambda(S, \pi_\theta) - \Delta_{\pi_\theta}(\widehat{\pi}_0, \pi_0)\right| \le \frac{2\upsilon(\delta)}{\lambda}\left(\frac{1}{e^{\lambda\widehat{\nu}^{1/(1+\epsilon)}}} + \frac{1}{e^{\lambda\nu^{1/(1+\epsilon)}}}\right),$$

*where,* $\upsilon(\delta) = \frac{\log\frac{4}{\delta}}{3n} + \sqrt{\frac{\log\frac{4}{\delta}}{n}}$.

*Proof.* Set $Y_\theta(A, X) = w_\theta(A, X)R$, $\widehat{Y}_\theta(A, X) = \widehat{w}_\theta(A, X)r(A, X)$, $u_i = \frac{1}{\lambda}\left(e^{\widehat{y}_i} - e^{\lambda\Delta_{\pi_\theta}(\widehat{\pi}_0, \pi_0)}\mu\right)$ and $v_i = \frac{1}{\lambda}(e^{y_\theta(a_i, x_i)} - \mu)$, where $\mu = \mathbb{E}[e^{\lambda Y_\theta(A, X)}]$. We have $-\frac{\mu}{\lambda} \le v_i \le \frac{1}{\lambda} - \frac{\mu}{\lambda}$ and $-\frac{e^{\lambda\Delta_{\pi_\theta}(\widehat{\pi}_0, \pi_0)}\mu}{\lambda} \le u_i \le \frac{1}{\lambda} - \frac{e^{\lambda\Delta_{\pi_\theta}(\widehat{\pi}_0, \pi_0)}\mu}{\lambda}$. Then, using the one-sided Bernstein's inequality (Lemma B.4), and changing variables (Lemma B.5), we have:

$$\mathbb{P}\left(\frac{1}{n}\sum_{i=1}^n e^{\lambda y_\theta(a_i, x_i)} - \mathbb{E}[e^{\lambda Y_\theta(A, X)}] > \frac{(1-\mu)\log\frac{1}{\delta}}{3n} + \sqrt{\frac{\mathbb{V}\left(e^{\lambda Y_\theta(A, X)}\right)\log\frac{1}{\delta}}{n}}\right) \le \delta,$$

$$\mathbb{P}\left(\frac{1}{n}\sum_{i=1}^n e^{\lambda y_\theta(a_i, x_i)} - \mathbb{E}[e^{\lambda Y_\theta(A, X)}] < -\frac{\mu\log\frac{1}{\delta}}{3n} - \sqrt{\frac{\mathbb{V}\left(e^{\lambda Y_\theta(A, X)}\right)\log\frac{1}{\delta}}{n}}\right) \le \delta,$$

$$\mathbb{P}\left(\frac{1}{n}\sum_{i=1}^n e^{\lambda\widehat{y}_i} - e^{\lambda\Delta_{\pi_\theta}(\widehat{\pi}_0, \pi_0)}\mathbb{E}[e^{\lambda Y_\theta(A, X)}] > \frac{(1 - e^{\lambda\Delta_{\pi_\theta}(\widehat{\pi}_0, \pi_0)}\mu)\log\frac{1}{\delta}}{3n} + \sqrt{\frac{\mathbb{V}\left(e^{\lambda\widehat{Y}_\theta(A, X)}\right)\log\frac{1}{\delta}}{n}}\right) \le \delta,$$

$$\mathbb{P}\left(\frac{1}{n}\sum_{i=1}^n e^{\lambda\widehat{y}_i} - e^{\lambda\Delta_{\pi_\theta}(\widehat{\pi}_0, \pi_0)}\mathbb{E}[e^{\lambda Y_\theta(A, X)}] < -\frac{e^{\lambda\Delta_{\pi_\theta}(\widehat{\pi}_0, \pi_0)}\mu\log\frac{1}{\delta}}{3n} - \sqrt{\frac{\mathbb{V}\left(e^{\lambda\widehat{Y}_\theta(A, X)}\right)\log\frac{1}{\delta}}{n}}\right) \le \delta.$$

Therefore, with probability at least $1 - 2\delta$, for $\upsilon_2 < \frac{1}{2}\mathbb{E}[e^{\lambda Y_\theta(A, X)}]$, we have,

$$\widehat{V}_{\mathrm{LSE}}^\lambda(\widehat{S}, \pi_\theta) - \widehat{V}_{\mathrm{LSE}}^\lambda(S, \pi_\theta)$$

$$= \frac{1}{\lambda}\log\left(\frac{\sum_{i=1}^n e^{\lambda\widehat{y}_i}}{\sum_{i=1}^n e^{\lambda y_\theta(a_i, x_i)}}\right)$$

$$\le \frac{1}{\lambda}\log\left(\frac{e^{\lambda\Delta_{\pi_\theta}(\widehat{\pi}_0, \pi_0)}\mathbb{E}[e^{\lambda Y_\theta(A, X)}] + \upsilon_1}{\mathbb{E}[e^{\lambda Y_\theta(A, X)}] - \upsilon_2}\right)$$

$$= \frac{1}{\lambda}\left(\log\left(e^{\lambda\Delta_{\pi_\theta}(\widehat{\pi}_0, \pi_0)}\mathbb{E}[e^{\lambda Y_\theta(A, X)}] + \upsilon_1\right) - \log\left(\mathbb{E}[e^{\lambda Y_\theta(A, X)}] - \upsilon_2\right)\right)$$

$$\le \frac{1}{\lambda}\left(\log\left(e^{\lambda\Delta_{\pi_\theta}(\widehat{\pi}_0, \pi_0)}\mathbb{E}[e^{\lambda Y_\theta(A, X)}]\right) + \frac{\upsilon_1}{e^{\lambda\Delta_{\pi_\theta}(\widehat{\pi}_0, \pi_0)}\mathbb{E}[e^{\lambda Y_\theta(A, X)}]}\right.$$

$$\left. - \left(\log\left(\mathbb{E}[e^{\lambda Y_\theta(A, X)}]\right) - \frac{\upsilon_2}{\mathbb{E}[e^{\lambda Y_\theta(A, X)}] - \upsilon_2}\right)\right)$$

$$\le \Delta_{\pi_\theta}(\widehat{\pi}_0, \pi_0) + \frac{1}{\lambda}\left(\frac{\upsilon_1}{\mathbb{E}[e^{\lambda\widehat{Y}_\theta(A, X)}]} + \frac{2\upsilon_2}{\mathbb{E}[e^{\lambda Y_\theta(A, X)}]}\right)$$

$$\le \Delta_{\pi_\theta}(\widehat{\pi}_0, \pi_0) + \frac{2}{\lambda}\left(\frac{\upsilon_1}{\mathbb{E}[e^{\lambda\widehat{Y}_\theta(A, X)}]} + \frac{\upsilon_2}{\mathbb{E}[e^{\lambda Y_\theta(A, X)}]}\right).$$

where

$$v_1 = \frac{(1 - \mathbb{E}[e^{\lambda \widehat{Y}_\theta(A,X)}]) \log \frac{1}{\delta}}{3n} + \sqrt{\frac{\mathbb{V}\left(e^{\lambda \widehat{Y}_\theta(A,X)}\right) \log \frac{1}{\delta}}{n}},$$

$$v_2 = \frac{\mathbb{E}[e^{\lambda Y_\theta(A,X)}] \log \frac{1}{\delta}}{3n} + \sqrt{\frac{\mathbb{V}\left(e^{\lambda Y_\theta(A,X)}\right) \log \frac{1}{\delta}}{n}}.$$

Similarly, with probability at least $1 - 2\delta$ we have, given $v_3 < \frac{1}{2}\mathbb{E}[e^{\lambda \widehat{Y}_\theta(A,X)}]$,

$$\widehat{V}^\lambda_{\text{LSE}}(\widehat{S}, \pi_\theta) - \widehat{V}^\lambda_{\text{LSE}}(S, \pi_\theta) \geq \Delta_{\pi_\theta}(\widehat{\pi}_0, \pi_0) - \frac{2}{\lambda}\left(\frac{v_3}{\mathbb{E}[e^{\lambda \widehat{Y}_\theta(A,X)}]} + \frac{v_4}{\mathbb{E}[e^{\lambda Y_\theta(A,X)}]}\right),$$

where,

$$v_3 = \frac{\mathbb{E}[e^{\lambda \widehat{Y}_\theta(A,X)}] \log \frac{1}{\delta}}{3n} + \sqrt{\frac{\mathbb{V}\left(e^{\lambda \widehat{Y}_\theta(A,X)}\right) \log \frac{1}{\delta}}{n}},$$

$$v_4 = \frac{(1 - \mathbb{E}[e^{\lambda Y_\theta(A,X)}]) \log \frac{1}{\delta}}{3n} + \sqrt{\frac{\mathbb{V}\left(e^{\lambda Y_\theta(A,X)}\right) \log \frac{1}{\delta}}{n}}.$$

Therefore, with probability at least $1 - 4\delta$ we have,

$$\Delta_{\pi_\theta}(\widehat{\pi}_0, \pi_0) - \frac{2}{\lambda}\left(\frac{v_3}{\mathbb{E}[e^{\lambda \widehat{Y}_\theta(A,X)}]} + \frac{v_4}{\mathbb{E}[e^{\lambda Y_\theta(A,X)}]}\right) \leq \widehat{V}^\lambda_{\text{LSE}}(\hat{S}, \pi_\theta) - \widehat{V}^\lambda_{\text{LSE}}(S, \pi_\theta)$$

$$\leq \Delta_{\pi_\theta}(\widehat{\pi}_0, \pi_0) + \frac{2}{\lambda}\left(\frac{v_1}{\mathbb{E}[e^{\lambda \widehat{Y}_\theta(A,X)}]} + \frac{v_2}{\mathbb{E}[e^{\lambda Y_\theta(A,X)}]}\right).$$

We have for $i \in [4]$,

$$v_i \leq \frac{\log \frac{1}{\delta}}{3n} + \sqrt{\frac{\log \frac{1}{\delta}}{n}}.$$

So, replacing $\delta$ with $\delta/4$, we have with probability at least $(1 - \delta)$,

$$\left|\widehat{V}^\lambda_{\text{LSE}}(\widehat{S}, \pi_\theta) - \widehat{V}^\lambda_{\text{LSE}}(S, \pi_\theta) - \Delta_{\pi_\theta}(\widehat{\pi}_0, \pi_0)\right|$$

$$\leq \frac{2}{\lambda}\left(\frac{\log \frac{4}{\delta}}{3n} + \sqrt{\frac{\log \frac{4}{\delta}}{n}}\right)\left(\frac{1}{\mathbb{E}[e^{\lambda \widehat{Y}_\theta(A,X)}]} + \frac{1}{\mathbb{E}[e^{\lambda Y_\theta(A,X)}]}\right)$$

$$\leq \frac{2}{\lambda}\left(\frac{\log \frac{4}{\delta}}{3n} + \sqrt{\frac{\log \frac{4}{\delta}}{n}}\right)\frac{2\epsilon}{\lambda}\left(\frac{1}{e^{\lambda \widehat{\nu}^{1/(1+\epsilon)}}} + \frac{1}{e^{\lambda \nu^{1/(1+\epsilon)}}}\right),$$

which is true given $\frac{\log \frac{4}{\delta}}{3n} + \sqrt{\frac{\log \frac{4}{\delta}}{n}} < \frac{1}{2}\min\left(\mathbb{E}[e^{\lambda Y_\theta(A,X)}], \mathbb{E}[e^{\lambda \widehat{Y}_\theta(A,X)}]\right)$. According to Lemma B.6, this is satisfied by

$$n > \frac{\frac{4}{3}\mu_{\min} + 4}{\mu_{\min}^2} \log \frac{4}{\delta}.$$

$\square$

In the following theorem, we study the regret of the LSE estimator under $\pi_{\widetilde{\theta}}(S)$ policy.

**Theorem E.7.** *Suppose that,*

$$\pi_{\widetilde{\theta}}(\widehat{S}) = \arg\max_{\pi_\theta \in \Pi_\Theta} \widehat{V}^\lambda_{\mathrm{LSE}}(\widehat{S}, \pi_\theta),$$

*where $\widehat{S}$ is the data with noisy propensity scores. For any $\gamma \in (0,1)$, given Assumption 5.1, and E.3, and assuming that $n \geq \max\left(\frac{\frac{4}{3}\mu_{\min}+4}{\mu_{\min}^2}\log\frac{4|\Pi_\theta|}{\delta}, \frac{(2|\lambda|^{1+\epsilon}\nu+\frac{4}{3}\gamma)\log\frac{4|\Pi_\theta|}{\delta}}{\gamma^2\exp(2\lambda\nu^{1/(1+\epsilon)})}\right)$ where $\mu_{\min} = \min\left(e^{\lambda\nu^{1/(1+\epsilon)}}, e^{\lambda\widehat{\nu}^{1/(1+\epsilon)}}\right)$, the following upper bound holds on the regret of the LSE estimator under $\pi_{\widetilde{\theta}}(S)$ with probability at least $(1-\delta)$ for $\delta \in (0,1)$,*

$$
\begin{aligned}
\mathfrak{R}_\lambda(\pi_{\widetilde{\theta}}, S) &\leq \frac{2|\lambda|^\epsilon}{1+\epsilon}\nu + \frac{|\lambda|^\epsilon}{1+\epsilon}\widehat{\nu} \\
&\quad - \frac{4(2-\gamma)}{3(1-\gamma)}\frac{\log\frac{4|\Pi_\theta|}{\delta}}{n\lambda\exp(\lambda\nu^{1/(1+\epsilon)})} - \frac{(2-\gamma)}{(1-\gamma)\lambda}\sqrt{\frac{4|\lambda|^{1+\epsilon}\nu\log\frac{4|\Pi_\theta|}{\delta}}{n\exp(2\lambda\nu^{1/(1+\epsilon)})}} \\
&\quad + d_{\pi_0}(\widehat{w}_{\widehat{\theta}}(A,X)R, w_{\widehat{\theta}}(A,X)R) + d_{\pi_0}(\widehat{w}_{\widetilde{\theta}}(A,X)R, w_{\widetilde{\theta}}(A,X)R) \\
&\quad + \frac{4v(\frac{\delta}{4|\Pi_\theta|})}{\lambda}\left(\frac{1}{e^{\lambda\nu^{1/(1+\epsilon)}}} + \frac{1}{e^{\lambda\widehat{\nu}^{1/(1+\epsilon)}}}\right),
\end{aligned}
\tag{76}
$$

*where, $v(\delta) = \frac{\log\frac{4}{\delta}}{3n} + \sqrt{\frac{\log\frac{4}{\delta}}{n}}$.*

*Proof.* Let $\widehat{\theta}$ be,

$$\pi_{\widehat{\theta}}(S) = \arg\max_{\pi_\theta \in \Pi_\Theta} \widehat{V}^\lambda_{\mathrm{LSE}}(S, \pi_\theta).$$

We decompose the regret as follows,

$$
\begin{aligned}
&\mathfrak{R}_\lambda(\pi_{\widetilde{\theta}}, S) \\
&= V(\pi_{\theta^*}) - V(\pi_{\widetilde{\theta}}) \\
&= \widehat{V}^\lambda_{\mathrm{LSE}}(S, \pi_{\widetilde{\theta}}) - V(\pi_{\widetilde{\theta}}) \\
&\quad - \widehat{V}^\lambda_{\mathrm{LSE}}(S, \pi_{\widetilde{\theta}}) + \widehat{V}^\lambda_{\mathrm{LSE}}(\widehat{S}, \pi_{\widetilde{\theta}}) \\
&\quad - \widehat{V}^\lambda_{\mathrm{LSE}}(\widehat{S}, \pi_{\widetilde{\theta}}) + \widehat{V}^\lambda_{\mathrm{LSE}}(\widehat{S}, \pi_{\widehat{\theta}}) \\
&\quad - \widehat{V}^\lambda_{\mathrm{LSE}}(\widehat{S}, \pi_{\widehat{\theta}}) + \widehat{V}^\lambda_{\mathrm{LSE}}(S, \pi_{\widehat{\theta}}) \\
&\quad - \widehat{V}^\lambda_{\mathrm{LSE}}(S, \pi_{\widehat{\theta}}) + \widehat{V}^\lambda_{\mathrm{LSE}}(S, \pi_{\theta^*}) \\
&\quad - \widehat{V}^\lambda_{\mathrm{LSE}}(S, \pi_{\theta^*}) + V(\pi_{\theta^*}).
\end{aligned}
$$

Using the estimation error bounds at Theorem D.3 and Theorem D.2 and using the union bound, with probability $(1-\delta)$ we have,

$$\widehat{V}^\lambda_{\mathrm{LSE}}(S, \pi_{\widetilde{\theta}}) - V(\pi_{\widetilde{\theta}}) \leq -\frac{1}{\lambda(1-\gamma)}\sqrt{\frac{4|\lambda|^{1+\epsilon}\nu\log(2|\Pi_\theta|/\delta)}{n\exp(2\lambda\nu^{1/(1+\epsilon)})}} - \frac{4\log(2|\Pi_\theta|/\delta)}{3(1-\gamma)\lambda\exp(\lambda\nu^{1/(1+\epsilon)})n}, \tag{77}$$

$$V(\pi_{\theta^*}) - \widehat{V}^\lambda_{\mathrm{LSE}}(S, \pi_{\theta^*}) \leq \frac{1}{1+\epsilon}|\lambda|^\epsilon\nu - \frac{1}{\lambda}\sqrt{\frac{4|\lambda|^{1+\epsilon}\nu\log(2|\Pi_\theta|/\delta)}{n\exp(2\lambda\nu^{1/(1+\epsilon)})}} - \frac{4\log(2|\Pi_\theta|/\delta)}{3\lambda\exp(\lambda\nu^{1/(1+\epsilon)})n}. \tag{78}$$

In addition, using Proposition E.6, we have,

$$\widehat{V}^\lambda_{\mathrm{LSE}}(\widehat{S}, \pi_{\widetilde{\theta}}) - \widehat{V}^\lambda_{\mathrm{LSE}}(S, \pi_{\widetilde{\theta}}) \leq \Delta_{\pi_{\widetilde{\theta}}}(\widehat{\pi}_0, \pi_0) + \frac{2v(\delta/|\Pi_\theta|)}{\lambda}\left(\frac{1}{e^{\lambda\widehat{\nu}^{1/(1+\epsilon)}}} + \frac{1}{e^{\lambda\nu^{1/(1+\epsilon)}}}\right), \tag{79}$$

$$\widehat{V}^\lambda_{\mathrm{LSE}}(S, \pi_{\widehat{\theta}}) - \widehat{V}^\lambda_{\mathrm{LSE}}(\widehat{S}, \pi_{\widehat{\theta}}) \leq \Delta_{\pi_{\widehat{\theta}}}(\widehat{\pi}_0, \pi_0) + \frac{2v(\delta/|\Pi_\theta|)}{\lambda}\left(\frac{1}{e^{\lambda\widehat{\nu}^{1/(1+\epsilon)}}} + \frac{1}{e^{\lambda\nu^{1/(1+\epsilon)}}}\right). \tag{80}$$

As $\pi_{\widetilde{\theta}}$ is the maximizer of $\widehat{V}^\lambda_{\mathrm{LSE}}(\widehat{S}, \pi_\theta)$, we have,

$$\widehat{V}^\lambda_{\mathrm{LSE}}(\widehat{S}, \pi_{\widehat{\theta}}) - \widehat{V}^\lambda_{\mathrm{LSE}}(\widehat{S}, \pi_{\widetilde{\theta}}) \leq 0, \tag{81}$$

and as $\pi_{\widehat{\theta}}$ is the maximizer of $\widehat{V}^\lambda_{\mathrm{LSE}}(S, \pi_\theta)$ we have,

$$\widehat{V}^\lambda_{\mathrm{LSE}}(S, \pi_{\theta^*}) - \widehat{V}^\lambda_{\mathrm{LSE}}(S, \pi_{\widehat{\theta}}) \leq 0. \tag{82}$$

So putting all together, using the union bound we have with probability at least $1 - \delta$,

$$
\begin{aligned}
V(\pi_{\widetilde{\theta}}) - V(\pi_{\theta^*}) \leq{} & \frac{|\lambda|^\epsilon}{1+\epsilon}\nu - \frac{4(2-\gamma)}{3(1-\gamma)}\frac{\log\frac{4|\Pi_\theta|}{\delta}}{n\lambda\exp(\lambda\nu^{1/(1+\epsilon)})} - \frac{(2-\gamma)}{(1-\gamma)\lambda}\sqrt{\frac{4|\lambda|^{1+\epsilon}\nu\log\frac{4|\Pi_\theta|}{\delta}}{n\exp(2\lambda\nu^{1/(1+\epsilon)})}}. \\
& + \Delta_{\pi_{\widehat{\theta}}}(\widehat{\pi}_0, \pi_0) - \Delta_{\pi_{\widetilde{\theta}}}(\widehat{\pi}_0, \pi_0) \\
& + \frac{2\upsilon\left(\frac{\delta}{4|\Pi_\theta|}\right)}{\lambda}\left(\frac{1}{e^{\lambda\nu^{1/(1+\epsilon)}}} + \frac{1}{e^{\lambda\widehat{\nu}^{1/(1+\epsilon)}}}\right),
\end{aligned}
$$

where $\upsilon\left(\frac{\delta}{4|\Pi_\theta|}\right) = \frac{\log\left(\frac{16\Pi_\theta}{\delta}\right)}{3n} + \sqrt{\frac{\log\left(\frac{16\Pi_\theta}{\delta}\right)}{n}}$. The final result holds by applying Proposition E.5 to $\Delta_{\pi_{\widehat{\theta}}}(\widehat{\pi}_0, \pi_0) - \Delta_{\pi_{\widetilde{\theta}}}(\widehat{\pi}_0, \pi_0)$. $\qquad\square$

**Discussion:** The term $d_{\pi_0}(\widehat{w}_{\widehat{\theta}}(A, X)R, w_{\widehat{\theta}}(A, X)R) + d_{\pi_0}(\widehat{w}_{\widetilde{\theta}}(A, X)R, w_{\widetilde{\theta}}(A, X)R)$ in equation 76 can be interpreted as the cost of estimated propensity scores which is independent from $n$. Note that, we have the convergence rate of $O(n^{-\epsilon/(1+\epsilon)})$ for all remaining terms in equation 76.

In the following Corollary, we discuss that the small range of variation of the noise gives an upper bound on the variance of the LSE estimator under estimated propensity score.

---

**Corollary E.8.** *Under the same assumptions in Proposition E.6, then the following upper bound holds on the variance of the LSE estimator underestimated propensity scores with probability at least $(1 - \delta)$,*

$$\mathbb{V}(\widehat{V}^\lambda_{\mathrm{LSE}}(\widehat{S}, \pi_\theta)) \leq 2\mathbb{V}(\widehat{V}^\lambda_{\mathrm{LSE}}(S, \pi_\theta)) + 2B^2\varepsilon^2,$$

*where $\varepsilon = \frac{2}{\lambda}\left(\frac{\log\frac{1}{\delta}}{3n} + \sqrt{\frac{\log\frac{1}{\delta}}{n}}\right)$, and $B = \left(\frac{1}{e^{\lambda\widehat{\nu}^{1/(1+\epsilon)}}} + \frac{1}{e^{\lambda\nu^{1/(1+\epsilon)}}}\right)$.*

---

*Proof.* As $\Delta_{\pi_\theta}(\widehat{\pi}_0, \pi_0)$ is a constant with respect to $\widehat{V}^\lambda_{\mathrm{LSE}}(\widehat{S}, \pi_\theta)$ and $\widehat{V}^\lambda_{\mathrm{LSE}}(S, \pi_\theta)$, then we have,

$$\mathbb{V}(\widehat{V}^\lambda_{\mathrm{LSE}}(\widehat{S}, \pi_\theta) - \widehat{V}^\lambda_{\mathrm{LSE}}(S, \pi_\theta)) \leq \left(\frac{2B\varepsilon}{2}\right)^2 = B^2\epsilon^2.$$

Therefore,

$$
\begin{aligned}
\mathbb{V}(\widehat{V}^\lambda_{\mathrm{LSE}}(\widehat{S}, \pi_\theta)) &= \mathbb{V}(\widehat{V}^\lambda_{\mathrm{LSE}}(\widehat{S}, \pi_\theta) - \widehat{V}^\lambda_{\mathrm{LSE}}(S, \pi_\theta) + \widehat{V}^\lambda_{\mathrm{LSE}}(S, \pi_\theta)) \\
&= \mathbb{V}(\widehat{V}^\lambda_{\mathrm{LSE}}(\widehat{S}, \pi_\theta) - \widehat{V}^\lambda_{\mathrm{LSE}}(S, \pi_\theta)) + \mathbb{V}(\widehat{V}^\lambda_{\mathrm{LSE}}(S, \pi_\theta)) \\
&\quad + 2\mathrm{Cov}(\widehat{V}^\lambda_{\mathrm{LSE}}(\widehat{S}, \pi_\theta) - \widehat{V}^\lambda_{\mathrm{LSE}}(S, \pi_\theta), \widehat{V}^\lambda_{\mathrm{LSE}}(S, \pi_\theta)) \\
&\leq \mathbb{V}(\widehat{V}^\lambda_{\mathrm{LSE}}(\widehat{S}, \pi_\theta) - \widehat{V}^\lambda_{\mathrm{LSE}}(S, \pi_\theta)) + \mathbb{V}(\widehat{V}^\lambda_{\mathrm{LSE}}(S, \pi_\theta)) \\
&\quad + 2\sqrt{\mathbb{V}(\widehat{V}^\lambda_{\mathrm{LSE}}(S, \pi_\theta))\mathbb{V}(\widehat{V}^\lambda_{\mathrm{LSE}}(\widehat{S}, \pi_\theta) - \widehat{V}^\lambda_{\mathrm{LSE}}(S, \pi_\theta))} \\
&= \left(\sqrt{\mathbb{V}(\widehat{V}^\lambda_{\mathrm{LSE}}(S, \pi_\theta))} + \sqrt{\mathbb{V}(\widehat{V}^\lambda_{\mathrm{LSE}}(\widehat{S}, \pi_\theta) - \widehat{V}^\lambda_{\mathrm{LSE}}(S, \pi_\theta))}\right)^2 \\
&\leq \left(\sqrt{\mathbb{V}(\widehat{V}^\lambda_{\mathrm{LSE}}(S, \pi_\theta))} + B\varepsilon\right)^2 \leq 2\mathbb{V}(\widehat{V}^\lambda_{\mathrm{LSE}}(S, \pi_\theta)) + 2B^2\varepsilon^2.
\end{aligned}
$$

$\qquad\square$

From Corollary E.8, we have an upper bound on the variance of the LSE estimator underestimated propensity scores, in terms of the variance of the LSE estimator under true propensity scores. Therefore, if $\mathbb{V}(\widehat{V}_{\mathrm{LSE}}^{\lambda}(S, \pi_\theta))$ is bounded, then we expect bounded $\mathbb{V}(\widehat{V}_{\mathrm{LSE}}^{\lambda}(\widehat{S}, \pi_\theta))$.

### E.1. Gamma Noise Discussion

For statistical modeling of the estimated propensity scores, as discussed in (Zhang et al., 2023b), suppose that the logging policy is a softmax policy with respect to $a$.

$$\pi_0(A|X) = \mathrm{softmax}(f_{\theta^*}(X, A)), \tag{83}$$

where $f_\theta$ is a function parameterized by $\theta$ that indicates the policy's function output before softmax operation and $\theta^*$ is the parameter of this function for the true logging policy.

We have an estimation of the function $f_{\theta^*}(X, A)$, as $f_{\widehat{\theta}}(X, A)$ and we model the error in the estimation of $f_{\theta^*}(X, A)$ as a random variable $Z$ which is a function of $X$ and $A$,

$$f_{\widehat{\theta}}(X, A) = f_{\theta^*}(X, A) + Z(X, A).$$

Then we have,

$$\begin{aligned} \widehat{\pi}_0 &= \mathrm{softmax}(f_{\widehat{\theta}}(X, A)) \\ &= \mathrm{softmax}(f_{\theta^*} + Z) \\ &\propto e^Z \pi_0. \end{aligned}$$

Motivated by Halliwell (2018), we use a negative log-gamma distribution for $Z$, which results in an inverse Gamma multiplicative noise on the propensity scores. Negative log-gamma distribution is skewed towards negative values, resulting in inverse gamma noise on the logging policy which is skewed towards values less than one. This pushes the propensity scores $\frac{\pi_\theta}{\pi_0}$ towards the higher variance, i.e., the logging policy is near zero and the importance weight becomes large.

In particular, we consider a model-based setting in which the noise is modelled with an inverse Gamma distribution. We use inverse gamma distribution $1/U$ as a multiplicative noise, so we have,

$$\widehat{\pi}_0 = \frac{1}{U}\pi_0 \rightarrow \widehat{w}_\theta(A, X) = U w_\theta(A, X).$$

which results in a multiplicative gamma noise on the importance-weighted reward. We choose $U \sim \mathrm{Gamma}(b, b)$, so $\mathbb{E}[U] = 1$. Hence, the expected value of the noisy version is the same as the original noiseless variable.

$$\mathbb{E}[U w_\theta(A, X) R] = \mathbb{E}[U]\mathbb{E}[w_\theta(A, X) R] = \mathbb{E}[w_\theta(A, X) R].$$

Note that we have

$$\mathbb{E}\left[e^{\lambda w_\theta(A, X) R U}\right] = \mathbb{E}\left[\left(\frac{1}{1 - \lambda w_\theta(A, X) R/b}\right)^b\right],$$

Therefore, $\mathbb{E}[e^{\lambda U w_\theta(A, X) R}]$ converges to $\mathbb{E}[e^{\lambda w_\theta(A, X) R}]$ for $b \to \infty$. Furthermore, we assume that for a large value $b$, $\Delta_{\pi_\theta}(\widehat{\pi}_0, \pi_0) \approx 0$ and using Proposition E.6, with a probability at least $(1 - \delta)$, we have,

$$\left|\widehat{V}_{\mathrm{LSE}}^{\lambda}(\widehat{S}, \pi_\theta) - \widehat{V}_{\mathrm{LSE}}^{\lambda}(S, \pi_\theta)\right| \leq \epsilon\left(\frac{1}{\mathbb{E}[e^{\lambda \widehat{w}_\theta(A, X) R}]} + \frac{1}{\mathbb{E}[e^{\lambda w_\theta(A, X) R}]}\right). \tag{84}$$

The impact of inverse Gamma noise on the LSE estimator is constrained when the noise's domain is sufficiently small. This property ensures that the LSE remains relatively stable under certain noise conditions. Furthermore, we can reduce the deviation from the original noiseless LSE by increasing the size of the Logged Bandit Feedback (LBF) dataset. This relationship demonstrates the estimator's robustness and scalability in practical applications.

Table 7: Statistics of the datasets used in our experiments. For image datasets the 2048-dimensional features from pretrained ResNet-50 are used.

| DATA SET | IPS-TRAINING SAMPLES | TEST SAMPLES | NUMBER OF ACTIONS | DIMENSION |
|---|---|---|---|---|
| FMNIST | $60,000$ | $10000$ | $10$ | $2048$ |
| EMNIST | $60,000$ | $10000$ | $10$ | $2048$ |
| KUAIREC | $12,530,806$ | $4,676,570$ | $10,728$ | $1555$ |

# F. Experiment Details

**Datasets:** In addition to dataset EMNIST, we also run our estimator over Fashion-MNIST (FMNIST) (Xiao et al., 2017).

**Setup Details:** We use mini-batch SGD as an optimizer for all experiments. The learning rate used for EMNIST and FMNIST datasets is 0.001. Furthermore, we use early stopping in our training phase and the maximum number of epochs is 300. For the image datasets, EMNIST and FMNIST, we use the last layer features from ResNet-50 model pretrained on the ImageNet dataset (Deng et al., 2009).

## F.1. Hyper-parameter Tuning

All experiments can be categorised into 4 classes,

- Supervised2Bandit OPL

- Synthetic OPE

- Supervised2Bandit OPE

- Real-world OPL

From different aspects, experiments have different setups.

1. Evaluation: For OPE experiments, multiple instances of the experiment are conducted and the empirical average squared error of the estimator is calculated as the estimation of MSE. For OPL, a separate test set is used to evaluate the estimator's performance.

2. Hyper-parameter tuning: Each estimator may have one or no hyper-parameter. For all experiments except Supervised2Bandit OPE, and the ones that the selection of hyperparameter is explicitly specified, the selection of this hyperparameter is conducted by grid-search. Other Table 8 indicates the search grid for each estimator. For supervised2Bandit OPE, for the estimators that provided a selection method (PM, LS, IX, OS, TR) we used their suggested value. For other estimators we used grid-search. For ES and LSE we used the following grids $\{0.0, 0.3, 0.5, 0.7, 1.0\}$, and $\{0.0, 0.001, 0.01, 0.1, 1.0\}$, respectively.

In order to find the value for each hyper-parameter, we put aside a part of the training dataset as a validation set and find the parameter that results in the highest accuracy on the validation set, and then we report the method's performance on the test set. In order to tune $\lambda$ we use grid search over the values in $\{0.01, 0.1, 1, 10, 100\}$.

**Hyper-Parameter Tuning for PM, ES, and IX Estimators**: For the PM, ES, and IX estimators, grid search will be used for hyper-parameter tuning. To tune the PM parameter $\lambda$, we will use data-driven approach proposed in (Metelli et al., 2021). For the ES estimator, the parameter $\alpha$ will be varied across $\alpha \in \{0.1, 0.4, 0.7, 1\}$. For the IX estimator, the $\gamma$ parameter will be tested with values in the set $\gamma \in \{0.01, 0.1, 1, 10, 100\}$.

## F.2. Code

The code for this study is written in Python. We use Pytorch for the training of our model. The supplementary material includes a zip file named rl_without_reward.zip with the following files:

Table 8: Hyperparameter of Different Estimators

| Estimator | Grid |
|---|---|
| ES | $\alpha \in \{0.0, 0.1, 0.4, 0.7, 1.0\}$ |
| IX | $\eta \in \{0.01, 0.1, 1.0, 10.0, 100.0\}$ |
| PM | $\hat{\lambda} \in \{0.0, 0.1, 0.3, 0.5, 0.8\}$ |
| OS | $\tau \in \{0.01, 0.1, 1.0, 10.0, 100.0\}$ |
| IPS-TR | $M \in \{2.0, 5.0, 10.0, 50.0\}$ |
| LS | $\tilde{\lambda} \in \{0.01, 0.1, 1.0, 10.0, 100.0\}$ |
| LSE | $\lambda \in \{-0.01, -0.1, -1.0, -10.0, -100.0\}$ |

- **preprocess_raw_dataset_from_model.py**: The code to generate the base pre-processed version of the datasets with raw input values.

- The **data** folder consists of any potentially generated bandit dataset (which can be generated by running the scripts in code).

- The **code** folder contains the scripts and codes written for the experiments.

  - **requirements.txt** contains the Python libraries required to reproduce our results.
  - **readme.md** includes the syntax of different commands in the code.
  - **accs**: A folder containing the result reports of different experiments.
  - **data.py** code to load data for image datasets.
  - **eval.py** code to evaluate estimators for image datasets and open bandit dataset.
  - **config**: Contains different configuration files for different setups.
  - **runs**: Folder containing different batch running scripts.
  - **loss.py**: Script of our loss functions including LSE.
  - **train_logging_policy.py**: Script to train the logging policy.
  - **train_reward_estimator.py**: Script to train the reward estimator for DM and DR methods.
  - **create_bandit_dataset.py**: Code for the generation of the bandit dataset using the logging policy.
  - **main_semi_ot.py**: Main training code which implements different methods proposed by our paper.
  - **synthetic_experiment_v3.py**: Code for synthetic experiments.
  - **motivation.ipynb**: Code for motivating example.
  - **OPE_classification**: The codes for the OPE experiments on real-world datasets from UCI repository.
    * **train_on_uci.ipynb**: Main code running experiments on UCI datasets.
    * **faulty_policy.py**: The code for the faulty policy model for the logging and training polices.
    * **UCI**: The folder containing UCI datasets used in the experiments.

- The **real_world** folder contains the scripts and codes written for Kuai-Rec dataset.

  - **preprocess_data.ipynb**: The code that preprocess the KuaiRec dataset and makes it ready for training.
  - **run_kuairec_experiments.py**: The main code for real dataset experiments. It contains the training of the logging policy as well as the learning policy
  - **eval.py**: Code containing the implementation of the evaluation metrics.

To use this code, the user needs to download and store the dataset using *preprocess_raw_dataset_from_model.py* script. All downloaded data will be stored in *data* directory. Then, to train the logging policy, the *code/train_logging_policy.py* should be run. Then, by using *code/create_bandit_dataset.py*, the LBF dataset corresponding to the experiment setup, will be created. Finally, to train the desired estimator, the user should use *code/main_semi_ot.py* script.

For OPE synthetic experiments, the code *synthetic_experiment_v3.py* should be run. For real-world OPL experiments, the Kuairec (version 2) dataset should be downloaded and put in *real_world/KuaiRec 2.0/* folder and first *real_world/preprocess_data.ipynb* notebook should be run and then *real_world/run_kuairec_experiments.py* code will train the estimators on Kuairec dataset. The code itself trains and stores a logging policy before the main training phase. For OPE real-world experiments, the notebook *OPE_classification/train_on_uci.ipynb* would train the estimators on the UCI datasets in the folder *OPE_classification/UCI*. The final version of the code is available at the following link: `https://github.com/armin-behnamnia/lse-offpolicy-learning`.

**Computational resources:** We have taken all our experiments using 3 servers, one with a nvidia 1080 Ti and one with two nvidia GTX 4090, and one with three nvidia 2070-Super GPUs.

# G. Additional Experiments

This section presents supplementary experiments to further validate our LSE approach in off-policy learning and evaluation. We extend our experiments as follows:

1. Comparison with the Model-based estimators: We conduct a series of experiments to assess the performance of model-based estimators in comparison with our LSE estimator.

2. Combined method: We investigate the efficacy of combining the LSE estimator with the Doubly Robust (DR) estimator, exploring potential synergies between these methods.

3. Real-world application: To demonstrate the practical relevance of our approach, we apply our methods to a real-world dataset, providing insights into their performance under real world datasets in off-policy learning scenarios.

4. $\lambda$ Effect: We study the effect of $\lambda$ in different scenarios.

5. Sample number effect: We study the performance of the LSE estimator with different number of samples $n$.

6. Off-policy evaluation: We conduct more off-policy evaluation using Lomax distribution.

7. Off-policy learning: We run more experiments for off-policy learning scenarios under FMNIST dataset.

8. Selection of $\lambda$: Different methods of the selection of $\lambda$, data-driven selection of $\lambda$ and sensitivity of $\lambda$ are explored.

9. Distributional properties: In the OPE scenario under heavy-tailed assumption, the distributional properties of LSE are studied.

10. Comparison with LS estimator: More Comparison with the LS estimator in the OPE setting based on choosing $\lambda$ is provided.

These additional experiments aim to provide a comprehensive evaluation of our proposed LSE estimator.

## G.1. Off-policy evaluation experiment

We conduct synthetic experiments to test our model's performance and behavior compared to other models and the effectiveness of our approach in the case of heavy-tailed rewards. We have two different settings. Gaussian setting in which the distributions are Gaussian random variables, having exponential tails, and Lomax setting in which the distributions are Lomax random variables, with polynomial tails. In all experiments we run 10K trials to estimate the bias, variance and MSE of each method, given MSE as the main criteria to compare the performance of different approaches. We conduct experiments on our method (LSE), power-mean estimator (PM) (Metelli et al., 2021), exponential smoothing (ES) (Aouali et al., 2023), IX estimator (Gabbianelli et al., 2023), truncated IPS (IPS-TR) (Ionides, 2008b), self-normalized IPS (SNIPS) (Swaminathan & Joachims, 2015b), OS estimator (Su et al., 2020) and LS estimator (Sakhi et al., 2024). The number of samples changes in different settings. In each setting, we grid search the hyperparameter of each method with 5 different values and select the one that leads to the least estimated MSE value. Note that the hyperparameter for each method is selected independently in each setting, but the candidate values are fixed throughout all settings.

**Gaussian:** In this setting, as explained in section 6, we have $\pi_\theta(\cdot|x_0) \sim \mathcal{N}(\mu_1, \sigma^2), \pi_0(\cdot|x_0) \sim \mathcal{N}(\mu_2, \sigma^2)$ and $r(x_0, u) = -e^{\alpha u^2}$. Given $2\alpha\sigma^2 < 1$, with simple calculations we have,

$$\mathbb{E}_{\pi_\theta}[r] = -\frac{1}{\sqrt{1 - 2\alpha\sigma^2}} \exp\left(\frac{\alpha\mu_1^2}{1 - 2\alpha\sigma^2}\right) \tag{85}$$

$$\mathbb{E}_{\pi_0}\left[\left|\frac{\pi_\theta}{\pi_0}r\right|^{1+\epsilon}\right] = |\mathbb{E}_{\pi_\theta}[r]| \exp\left(\frac{\epsilon(\mu_1 - \mu_2)((1 + \epsilon + 2\alpha\sigma^2)\mu_1 - (1 + \epsilon - 2\alpha\sigma^2)\mu_2)}{2\sigma^2(1 - 2\alpha\sigma^2)}\right) \tag{86}$$

We fix $\mu_1 = 0.5, \mu_2 = 1, \sigma^2 = 0.25$, but we change $\alpha$ as it increases the $1 + \epsilon$-moment of the weighted reward variable as it tends to $\frac{1}{2\sigma^2}$ and (given $\mu_1 > 0, \epsilon \leq \frac{\mu_1}{|\mu_1 - \mu_2|}$ or $\mu_1 > \mu_2$) leads to unbounded $1 + \epsilon$-moment for $\alpha = \frac{1}{2\sigma^2}$. We report the experiment results in Tables 9 and 10 As we can observe LSE effectively keeps the variance low without significant side-effects on bias, leading to an overall low MSE, making it a viable choice with general unbounded reward functions.

We also try different values for the number of samples, and observe the estimator's capability to work well on small number of samples and their performance growth with the number of samples. For $\alpha = 1.4$, the results of different methods for $n = 100, 1K, 10K, 100K$ are illustrated in Table 11.

**Discussion:** We observe that either in small sample size or large sample size, LSE beats other methods with a significant gap. Inspecting the bias of LSE through different sample sizes, the bias becomes fixed and doesn't decrease as the number of samples in the LBF dataset goes beyond 1K. This is due to the fixed candidate set for the parameter $\lambda$ in LSE and the presence of $\lambda$ in our derived bias upper bound in Proposition 5.5. This shows that the dependence of the bias on $\lambda$ that appears in the bias upper bound is tight and with a fixed $\lambda$, the bias doesn't vanish, no matter how much data we have and for a large number of samples it is critical to select $\lambda$ as a function of $n$. Furthermore, we can see that the variance of LSE effectively decreases as the number of samples increases. Here we can observe the decrease rate of $1/n$ in the variance, as it is proved in Proposition 5.7 under bounded second-moment assumption. We also observe that as $\alpha$ increases and the reward function's growth becomes bigger PM, IPS-TR, SNIPS, and OS suffer from a very large variance, while ES, LSE, IX, and LS-LIN manage to keep the variance relatively low. Among these low-variance methods, LSE achieves the lowest bias, indicating a better bias-variance trade-off. We hypothesize that this is due to the fact that LS-LIN, along LSE, is the only method that is not linear with respect to reward and compresses the reward besides the importance weight.

**Lomax:** In the Lomax setting, we use Lomax distributions with scale 1 for the learning and logging policies, $\pi_\theta(u|x_0) \sim \frac{\alpha}{(u+1)^{\alpha+1}}, \pi_0(u|x_0) \sim \frac{\alpha}{(u+1)^{\alpha'+1}}, \alpha, \alpha' > 0$. We use a polynomial function for the reward, $r(u) = (1 + u)^\beta, \beta > 0$. The main difference in this setting compared to the Gaussian setting is that here the tails of the distributions are polynomial, in contrast to the Gaussian setting in which the tails are exponential. In this setting, for $\alpha > \beta$, we have,

$$\mathbb{E}_{\pi_\theta}[r] = \frac{\alpha}{\alpha - \beta},$$

$$\mathbb{E}_{\pi_0}\left[\left|\frac{\pi_\theta}{\pi_0}r\right|^{1+\epsilon}\right] = \left(\frac{\alpha}{\alpha - \beta}\right)^{1+\epsilon} k^{-\epsilon}(1 + \epsilon(1 - k))^{-1},$$

where $k = \frac{\alpha'}{\alpha - \beta}$ and for the second inequality to hold we should have $1 + \epsilon(1 - k) > 0$. The condition $\alpha > \beta$ is sufficient for the weighted reward function to be $\epsilon$-heavy-tailed for some $\epsilon > 0$ (either $k < 1$ or $\epsilon < \frac{1}{|1-k|}$. We change the value of $\beta$ to $0.5, 1, 2$. We also fix $\alpha - \beta = 0.5$, to keep the value function in an appropriate range. We change $k$ to get different values for $\alpha' = k(\alpha - \beta)$ which determines the tail of the weighted reward variable. We set $k = 2, 3, 4$. The results are shown in Tables 13 and 14. We observe the superior performance of LSE compared to other methods.

**Discussion:** In Lomax experiments the LSE estimator has the best performance in most of the settings. In two settings, i.e., $\beta = 0.5$ and $\alpha' \in \{1.5, 2.0\}$, IPS-TR does better than LSE with a very small margin, yet LSE is the second-best model in these two settings. Similar to the Gaussian setting, we also run the experiments for different numbers of samples to inspect the effect of the number of samples on the performance of the models. We fix $\alpha = 2.5$, $\beta = 2$ and $\alpha' = 1.5$ in this scenario. Table 15 reports the performance of LSE across different number of samples. The same conclusions as the Gaussian setting are also observable in the Lomax setting. We can observe that LSE has better performance for $n = 100, 10K, 100K$.

In order to have an overall picture of our estimator's performance in OPE, for each estimator we report the number of experiments in which it becomes the first and second best-performing estimator. This is illustrated in Table 12.

We observe that in 13 out of 25 experiments, the LSE estimator outperforms other estimators. Additionally, it ranks second in 11 of the remaining 12 experiments. The overall report shows that LSE and LS dominate other methods in OPE, and both perform well with LSE winning with a small margin.

### G.1.1. HEAVY-TAILED DISTRIBUTION FAMILIES

To confirm our method's superior performance in heavy-tailed scenarios, we conduct experiments on different families of heavy-tailed reward distributions. Other than Lomax, we test on **Generalized Extreme Value (GEV)**, **Frechet**, and **Student's** $t$ distributions. For GEV, we set $c = -0.9$, for Student's $t$ distribution, we set $df = 1.2$, for Frechet (inv-weibull) we set $c = 1.2$, and for Lomax we set $\alpha = 1.2$. To keep the values positive, we consider the absolute value of the reward values. Note that this does not change the tail behavior of the distribution. Table 16 shows compare the performance of LSE compared to other methods in OPE on these heavy-tailed reward distributions.

Table 9: Bias, variance, and MSE of LSE, ES, PM, IX, IPS-TR, SNIPS, LS-LIN, and OS estimators with Gaussian distributions for $\alpha = 1.0, 1.1, 1.2, 1.3$. The experiment was run 10000 times and the variance, bias, and MSE of the estimations are reported. The best-performing result is highlighted in **bold** text, while the second-best result is colored in red for each scenario.

| $\alpha$ | Estimator | Bias | Variance | MSE |
|---|---|---|---|---|
| | PM | 0.037 | 0.004 | 0.006 |
| | ES | −0.001 | 0.006 | 0.006 |
| | LSE | 0.021 | 0.003 | **0.003** |
| | IPS-TR | 0.019 | 0.004 | 0.004 |
| 1.0 | IX | 0.168 | 0.001 | 0.029 |
| | SNIPS | −0.003 | 0.008 | 0.008 |
| | LS-LIN | 0.151 | 0.001 | 0.024 |
| | LS | 0.006 | 0.005 | 0.005 |
| | OS | 0.505 | 0.005 | 0.260 |
| | PM | 0.004 | 0.063 | 0.063 |
| | ES | −0.001 | 0.054 | 0.054 |
| | LSE | 0.052 | 0.006 | **0.009** |
| | IPS-TR | 0.020 | 0.052 | 0.052 |
| 1.1 | IX | 0.237 | 0.002 | 0.058 |
| | SNIPS | −0.005 | 0.059 | 0.059 |
| | LS-LIN | 0.284 | 0.001 | 0.082 |
| | LS | 0.082 | 0.007 | 0.0135 |
| | OS | 0.521 | 0.020 | 0.292 |
| | PM | −0.043 | 0.435 | 0.437 |
| | ES | 0.000 | 0.357 | 0.357 |
| | LSE | 0.152 | 0.014 | **0.037** |
| | IPS-TR | 0.024 | 0.353 | 0.354 |
| 1.2 | IX | 0.373 | 0.005 | 0.144 |
| | SNIPS | −0.003 | 0.366 | 0.366 |
| | LS-LIN | 0.545 | 0.002 | 0.299 |
| | LS | 0.183 | 0.016 | 0.050 |
| | OS | 0.541 | 0.116 | 0.409 |
| | PM | −0.121 | 1.731 | 1.746 |
| | ES | 1.162 | 0.026 | 1.377 |
| | LSE | 0.158 | 0.124 | **0.148** |
| | IPS-TR | 0.030 | 1.404 | 1.405 |
| 1.3 | IX | 0.662 | 0.016 | 0.453 |
| | SNIPS | −0.000 | 1.491 | 1.491 |
| | LS-LIN | 1.069 | 0.003 | 1.145 |
| | LS | 0.155 | 0164 | 0.188 |
| | OS | 0.463 | 56.581 | 56.796 |

Table 10: Bias, variance, and MSE of LSE, ES, PM, IX, IPS-TR, SNIPS, LS-LIN, and OS estimators with Gaussian distributions for $\alpha = 1.4, 1.5, 1.6, 1.7$. The experiment was run 10000 times and the variance, bias, and MSE of the estimations are reported. The best-performing result is highlighted in **bold** text, while the second-best result is colored in red for each scenario.

| $\alpha$ | Estimator | Bias | Variance | MSE |
|---|---|---|---|---|
| | PM | −0.301 | 164.951 | 165.041 |
| | ES | 1.959 | 0.396 | 4.232 |
| | LSE | 0.615 | 0.292 | **0.670** |
| | IPS-TR | 0.053 | 133.688 | 133.691 |
| 1.4 | IX | 1.340 | 0.048 | 1.842 |
| | SNIPS | −0.029 | 133.520 | 133.521 |
| | LS-LIN | 2.164 | 0.005 | 4.687 |
| | LS | 0.564 | 0.458 | 0.776 |
| | OS | 0.623 | 23.589 | 23.977 |
| | PM | −0.205 | 222.003 | 222.045 |
| | ES | 3.850 | 1.505 | 16.324 |
| | LSE | 2.132 | 0.645 | 5.190 |
| | IPS-TR | 0.349 | 179.990 | 180.112 |
| 1.5 | IX | 3.116 | 0.153 | 9.865 |
| | SNIPS | 0.315 | 194.830 | 194.929 |
| | LS-LIN | 4.682 | 0.009 | 21.927 |
| | LS | 1.968 | 1.156 | **5.028** |
| | OS | 1.096 | 504.001 | 505.205 |
| | PM | 0.726 | 5095.725 | 5096.252 |
| | ES | 9.420 | 22.685 | 111.416 |
| | LSE | 7.541 | 1.233 | 58.105 |
| | IPS-TR | 1.903 | 4131.016 | 4134.636 |
| 1.6 | IX | 8.665 | 0.502 | 75.589 |
| | SNIPS | 1.860 | 4426.166 | 4429.625 |
| | LS-LIN | 11.547 | 0.015 | 133.349 |
| | LS | 7.148 | 2.595 | **53.689** |
| | OS | 3.669 | 1303.684 | 1317.146 |
| | PM | 9.943 | 125126.550 | 125225.418 |
| | ES | 38.531 | 0.301 | 1484.959 |
| | LSE | 32.107 | 2.244 | 1033.093 |
| | IPS-TR | 12.880 | 101427.776 | 101593.680 |
| 1.7 | IX | 32.923 | 1.802 | 1085.753 |
| | SNIPS | 12.704 | 102027.853 | 102189.250 |
| | LS-LIN | 38.112 | 0.024 | 1452.556 |
| | LS | 31.227 | 5.267 | **980.41** |
| | OS | 29.171 | 17767.954 | 18618.899 |

Table 11: Bias, variance, and MSE of LSE, ES, PM, IX, IPS-TR, SNIPS, LS-LIN, and OS estimators with Gaussian distributions setup. The experiment was run 10000 times fixing $\alpha = 1.4$ and different number of samples $n \in \{100, 1000, 10000, 100000\}$. The variance, bias, and MSE of the estimations are reported. The best-performing result is highlighted in **bold** text, while the second-best result is colored in red for each scenario.

| $n$ | Estimator | Bias | Variance | MSE |
|---|---|---|---|---|
| | PM | −0.1288 | 203.5015 | 203.5181 |
| | ES | 1.9769 | 1.7696 | 5.6775 |
| | LSE | 1.2210 | 0.5015 | **1.9925** |
| | IPS-TR | 0.1617 | 164.9972 | 165.0234 |
| 100 | IX | 1.3459 | 0.4783 | 2.2897 |
| | SNIPS | 0.0074 | 196.8881 | 196.8881 |
| | LS-LIN | 2.1683 | 0.0568 | 4.7585 |
| | LS | 1.1817 | 0.8115 | 2.2079 |
| | OS | 0.7661 | 10.2588 | 10.8458 |
| | PM | −0.1963 | 18.3363 | 18.3749 |
| | ES | 1.9587 | 0.1694 | 4.0058 |
| | LSE | 0.6030 | 0.2999 | **0.6635** |
| | IPS-TR | 0.1007 | 14.8696 | 14.8798 |
| 1000 | IX | 1.3375 | 0.0486 | 1.8376 |
| | SNIPS | 0.0594 | 15.0741 | 15.0776 |
| | LS-LIN | 2.1646 | 0.0056 | 4.6910 |
| | LS | 0.5640 | 0.4580 | 0.7761 |
| | OS | 0.6432 | 8.7698 | 9.1835 |
| | PM | −0.2282 | 10.4458 | 10.4979 |
| | ES | 1.9625 | 0.0285 | 3.8800 |
| | LSE | 0.6159 | 0.0296 | 0.4089 |
| | IPS-TR | 0.0464 | 8.4660 | 8.4681 |
| 10000 | IX | 1.3410 | 0.0048 | 1.8031 |
| | SNIPS | 0.0435 | 8.5986 | 8.6005 |
| | LS-LIN | 2.1644 | 0.0005 | 4.6852 |
| | LS | 0.5606 | 0.0466 | **0.3609** |
| | OS | 0.5564 | 4.8936 | 5.2032 |
| | PM | −0.2505 | 1.8148 | 1.8775 |
| | ES | 0.0246 | 1.4707 | 1.4713 |
| | LSE | 0.6160 | 0.0029 | 0.3823 |
| | IPS-TR | 0.0250 | 1.4706 | 1.4712 |
| 100000 | IX | 1.3408 | 0.0005 | 1.7982 |
| | SNIPS | 0.0246 | 1.4757 | 1.4763 |
| | LS-LIN | 2.1629 | 5.6014 | 4.6783 |
| | LS | 0.5584 | 0.0049 | **0.3167** |
| | OS | 0.5823 | 0.8251 | 1.1643 |

Table 12: Comparison of different estimators in terms of the number of best|second rank performances of all Lomax and Gaussian experiment setups in OPE scenario.

| Estimator | Gaussian | Lomax | Total |
|:---------:|:--------:|:-----:|:-----:|
| LSE | 7\|5 | 6\|6 | 13\|11 |
| OS | 0\|0 | 0\|0 | 0\|0 |
| PM | 0\|0 | 0\|0 | 0\|0 |
| ES | 0\|0 | 0\|0 | 0\|0 |
| LS | 5\|7 | 6\|6 | 11\|13 |
| IPS-TR | 0\|0 | 1\|1 | 1\|1 |
| IX | 0\|0 | 0\|0 | 0\|0 |

### G.2. Off-policy learning experiment

We present the results of our experiments for EMNIST and FMNIST in Table 18.

As we can observe in the results for different scenarios and datasets, our estimator shows dominant performance among other baselines. The details of the number of best-performing and second-rank estimators are provided in Table 17. We observe that in 21 out of 30 experiments, the LSE estimator outperforms other estimators. Additionally, it ranks second in 7 of the remaining 9 experiments.

In the noisy scenario, where noise robustness is critical, increasing the noise on the propensity scores by reducing the $b$ value results in a marked decrease in the performance of all estimators, with the notable exception of LSE, which exhibits superior noise robustness.

In all two datasets, without noise, increasing $\tau$ has a negligible impact on the estimators. However, in noisy scenarios, a higher $\tau$ leads to decreased performance. This happens because as $\tau$ increases, the logging policy distribution approaches a uniform distribution, making it easier for noise to affect the argmax value, thereby reducing the estimators' performance. Notably, the LSE estimator demonstrates better robustness compared to other estimators, consistently showing superior performance in all noisy setups when $b = 0.01$.

Table 13: Bias, variance, and MSE of LSE, ES, PM, IX, IPS-TR, SNIPS, LS-LIN, and OS estimators with Lomax distributions setup for $\beta = 0.5, 1.5$. The experiment was run 10000 times with different values of $\alpha$, $\alpha'$ and $\beta$. The variance, bias, and MSE of the estimations are reported. The best-performing result is highlighted in **bold** text, while the second-best result is colored in red for each scenario.

| $\beta$ | $\alpha$ | $\alpha'$ | Method | Bias | Variance | MSE |
|---|---|---|---|---|---|---|
| 0.5 | 1.0 | 1.0 | PM | $-0.0004$ | 0.0197 | 0.0197 |
| | | | ES | $-0.0004$ | 0.0197 | 0.0197 |
| | | | LSE | 0.0361 | 0.0047 | 0.0060 |
| | | | IPS-TR | $-0.0004$ | 0.0197 | 0.0197 |
| | | | IX | 0.6958 | 0.0001 | 0.4842 |
| | | | SNIPS | $-0.0004$ | 0.0197 | 0.0197 |
| | | | LS-LIN | 0.4475 | 0.0002 | 0.2005 |
| | | | LS | 0.0266 | 0.0046 | **0.0053** |
| | | | OS | 0.3332 | 0.0094 | 0.1204 |
| | | 1.5 | PM | 0.2191 | 0.0154 | 0.0634 |
| | | | ES | 0.0145 | 0.2011 | 0.2013 |
| | | | LSE | 0.1702 | 0.0117 | 0.0407 |
| | | | IPS-TR | 0.1341 | 0.0146 | 0.0326 |
| | | | IX | 0.7815 | 0.0003 | 0.6111 |
| | | | SNIPS | 0.0181 | 0.1668 | 0.1671 |
| | | | LS-LIN | 0.5303 | 0.0011 | 0.2822 |
| | | | LS | 0.0697 | 0.0346 | **0.0395** |
| | | | OS | 0.7636 | 0.0007 | 0.5838 |
| | | 2.0 | PM | 0.4784 | 0.0084 | 0.2372 |
| | | | ES | 0.9554 | 0.0020 | 0.9147 |
| | | | LSE | 0.1586 | 0.0801 | 0.1052 |
| | | | IPS-TR | 0.2965 | 0.0171 | **0.1050** |
| | | | IX | 0.8641 | 0.0006 | 0.7472 |
| | | | SNIPS | 0.0580 | 1.1500 | 1.1533 |
| | | | LS-LIN | 0.6106 | 0.0023 | 0.3751 |
| | | | LS | 0.3086 | 0.0238 | 0.1190 |
| | | | OS | 1.0176 | 0.0003 | 1.0358 |
| 1 | 1.5 | 1.0 | PM | $-0.0823$ | 0.0440 | 0.0508 |
| | | | ES | 0.0006 | 0.0357 | 0.0357 |
| | | | LSE | 0.0731 | 0.0092 | 0.0146 |
| | | | IPS-TR | 0.0006 | 0.0357 | 0.0357 |
| | | | IX | 1.0438 | 0.0002 | 1.0897 |
| | | | SNIPS | $-0.0003$ | 0.0418 | 0.0418 |
| | | | LS-LIN | 0.8513 | 0.0004 | 0.7252 |
| | | | LS | 0.0429 | 0.0104 | **0.0122** |
| | | | OS | 0.3566 | 0.0364 | 0.1635 |
| | | 1.5 | PM | 0.0167 | 0.7885 | 0.7888 |
| | | | ES | 0.0167 | 0.7885 | 0.7888 |
| | | | LSE | 0.1122 | 0.0820 | 0.0946 |
| | | | IPS-TR | 0.0167 | 0.7885 | 0.7888 |
| | | | IX | 1.1723 | 0.0006 | 1.3749 |
| | | | SNIPS | 0.0167 | 0.7885 | 0.7888 |
| | | | LS-LIN | 0.9551 | 0.0014 | 0.9136 |
| | | | LS | 0.1183 | 0.0717 | **0.0857** |
| | | | OS | 0.5122 | 0.6815 | 0.9439 |
| | | 2.0 | PM | 0.3839 | 0.3198 | 0.4672 |
| | | | ES | 1.4337 | 0.0035 | 2.0589 |
| | | | LSE | 0.2731 | 0.1353 | **0.2099** |
| | | | IPS-TR | 0.2280 | 0.2424 | 0.2944 |
| | | | IX | 1.2957 | 0.0013 | 1.6801 |
| | | | SNIPS | 0.0614 | 2.3202 | 2.3239 |
| | | | LS-LIN | 1.0580 | 0.0030 | 1.1223 |
| | | | LS | 0.2548 | 0.1785 | 0.2434 |
| | | | OS | 1.2544 | 0.0059 | 1.5793 |

Table 14: Bias, variance, and MSE of LSE, ES, PM, IX, IPS-TR, SNIPS, LS-LIN, and OS estimators with Lomax distributions setup for $\beta = 2$. The experiment was run 10000 times with different values of $\alpha$, $\alpha'$ and $\beta$. The variance, bias, and MSE of the estimations are reported. The best-performing result is highlighted in **bold** text, while the second-best result is colored in red for each scenario.

| $\beta$ | $\alpha$ | $\alpha'$ | **Method** | **Bias** | **Variance** | **MSE** |
|---|---|---|---|---|---|---|
| 2 | 2.5 | 1.0 | PM | $-0.2267$ | 0.1913 | 0.2427 |
| | | | ES | $-0.0049$ | 0.1540 | 0.1540 |
| | | | LSE | 0.0304 | 0.0461 | 0.0471 |
| | | | IPS-TR | $-0.0049$ | 0.1540 | 0.1540 |
| | | | IX | 1.7392 | 0.0007 | 3.0256 |
| | | | SNIPS | $-0.0100$ | 0.1858 | 0.1859 |
| | | | LS-LIN | 1.9231 | 0.0011 | 3.6995 |
| | | | LS | 0.0819 | 0.0281 | **0.0348** |
| | | | OS | 0.5571 | 0.0849 | 0.3953 |
| | | 1.5 | PM | $-0.2510$ | 17.7398 | 17.8028 |
| | | | ES | 2.2891 | 0.0024 | 5.2425 |
| | | | LSE | 0.2266 | 0.1688 | **0.2201** |
| | | | IPS-TR | $-0.0042$ | 14.3693 | 14.3694 |
| | | | IX | 1.9546 | 0.0018 | 3.8224 |
| | | | SNIPS | $-0.0062$ | 14.4548 | 14.4549 |
| | | | LS-LIN | 2.0374 | 0.0016 | 4.1529 |
| | | | LS | 0.2330 | 0.1699 | 0.2242 |
| | | | OS | 0.3995 | 13.5957 | 13.7553 |
| | | 2.0 | PM | $-0.2114$ | 27.6307 | 27.6754 |
| | | | ES | 2.3886 | 0.0113 | 5.7167 |
| | | | LSE | 0.5334 | 0.2729 | **0.5574** |
| | | | IPS-TR | $-0.0086$ | 22.5415 | 22.5416 |
| | | | IX | 2.1606 | 0.0035 | 4.6717 |
| | | | SNIPS | $-0.0107$ | 22.6954 | 22.6955 |
| | | | LS-LIN | 2.1601 | 0.0034 | 4.6694 |
| | | | LS | 0.4946 | 0.3696 | 0.61424 |
| | | | OS | 0.5158 | 7.4515 | 7.7175 |

Table 15: Bias, variance, and MSE of LSE, ES, PM, IX, IPS-TR, SNIPS, LS-LIN, and OS estimators with Lomax distributions setup. The experiment is conducted for 10000 times and different number of samples $n \in \{100, 1000, 10000, 100000\}$. The variance, bias, and MSE of the estimations are reported. The best-performing result is highlighted in **bold** text, while the second-best result is colored in red for each scenario.

| $n$ | Estimator | Bias | Variance | MSE |
|---|---|---|---|---|
| | PM | $-0.2486$ | 75.480 | 75.542 |
| | ES | 2.2895 | 0.0244 | 5.2663 |
| | LSE | 0.6217 | 0.4035 | **0.7900** |
| 100 | IPS-TR | 0.0021 | 61.140 | 61.140 |
| | IX | 1.9546 | 0.0182 | 3.8388 |
| | SNIPS | $-0.0331$ | 67.583 | 67.583 |
| | LS-LIN | 2.0369 | 0.0168 | 4.1660 |
| | LS | 0.6339 | 0.5402 | 0.9421 |
| | OS | 0.4287 | 61.159 | 61.343 |
| | PM | $-0.2421$ | 10.960 | 11.019 |
| | ES | 2.2889 | 0.0024 | 5.2415 |
| | LSE | 0.2245 | 0.1702 | 0.2206 |
| 1000 | IPS-TR | 0.0037 | 8.8781 | 8.8780 |
| | IX | 1.9540 | 0.0018 | 3.8198 |
| | SNIPS | 0.0010 | 9.0742 | 9.0742 |
| | LS-LIN | 2.0375 | 0.0016 | 4.1531 |
| | LS | 0.2330 | 0.1699 | **0.2242** |
| | OS | 0.4345 | 8.8799 | 9.0687 |
| | PM | $-0.2317$ | 0.6596 | 0.7132 |
| | ES | 0.0131 | 0.5343 | 0.5345 |
| | LSE | 0.2253 | 0.0171 | **0.0679** |
| 10000 | IPS-TR | 0.0131 | 0.5342 | 0.5345 |
| | IX | 1.9539 | 0.0002 | 3.8180 |
| | SNIPS | 0.0133 | 0.5364 | 0.5366 |
| | LS-LIN | 2.0375 | 0.0002 | 4.1517 |
| | LS | 0.2338 | 0.0171 | 0.0717 |
| | OS | 0.4438 | 0.5345 | 0.7315 |
| | PM | $-0.2619$ | 0.6546 | 0.7232 |
| | ES | $-0.0140$ | 0.5302 | 0.5304 |
| | LSE | 0.2267 | 0.0019 | **0.0533** |
| 100000 | IPS-TR | $-0.0140$ | 0.5302 | 0.5304 |
| | IX | 1.9538 | 1.6977 | 3.8175 |
| | SNIPS | $-0.0137$ | 0.5284 | 0.5286 |
| | LS-LIN | 2.0374 | 1.6805 | 4.1509 |
| | LS | 0.2351 | 0.0019 | 0.0572 |
| | OS | 0.4166 | 0.5302 | 0.7038 |

Table 16: Bias, Variance, and MSE for different methods under heavy-tailed reward distributions.

| Distribution | Metric | LSE | LS | IX | ES | PM | OS | SNIPS | IPS-TR |
|---|---|---|---|---|---|---|---|---|---|
| Lomax | Bias | 1.511 | 1.617 | 4.671 | 2.584 | 0.5677 | 4.778 | 0.1058 | 0.9616 |
| | Variance | 0.4641 | 0.5156 | 0.6819 | 84.07 | 187.6 | 0.3136 | 190.4 | 171.5 |
| | MSE | **2.746** | 3.132 | 22.50 | 90.75 | 187.9 | 23.15 | 190.4 | 172.4 |
| GEV | Bias | 0.1204 | 0.2105 | 0.7073 | $-2.861$ | 0.2103 | 0.7220 | $-6.751$ | $-5.4139$ |
| | Variance | 0.0004 | 0.0004 | 0.0657 | 722.1 | 43.68 | 0.0054 | 2209 | 1604 |
| | MSE | **0.0149** | 0.0447 | 0.5660 | 730.2 | 43.72 | 0.5268 | 2255 | 1633 |
| Frechet | Bias | 1.474 | 1.598 | 3.965 | 2.993 | 1.1863 | 5.309 | 0.2678 | 1.235 |
| | Variance | 0.4842 | 0.5161 | 10.31 | 29.80 | 92.08 | 0.1344 | 132.8 | 80.58 |
| | MSE | **2.656** | 3.068 | 26.03 | 38.75 | 93.49 | 28.31 | 132.9 | 82.10 |
| Student's $t$ | Bias | 0.9914 | 1.072 | 3.688 | 2.086 | 0.7545 | 3.766 | 0.0029 | 0.7982 |
| | Variance | 0.3270 | 0.3647 | 0.1986 | 24.69 | 79.09 | 0.2374 | 205.6 | 61.42 |
| | MSE | **1.310** | 1.513 | 13.80 | 29.04 | 79.66 | 14.42 | 205.6 | 62.06 |

Table 17: Comparison of different estimators in terms of the number of best|second rank performances of all true propensity score/ reward, estimated (noisy) propensity scores and noisy reward experiment setups in OPL scenario.

| Estimator | True PS & Reward | Noisy PS | Noisy Reward | Total |
|---|---|---|---|---|
| LSE | 3\|2 | 10\|1 | 8\|4 | 21\|7 |
| OS | 1\|2 | 1\|0 | 3\|3 | 5\|5 |
| PM | 2\|1 | 1\|7 | 1\|5 | 4\|13 |
| ES | 0\|0 | 0\|3 | 0\|0 | 0\|4 |
| LS-LIN | 0\|1 | 0\|0 | 0\|0 | 0\|1 |
| IX | 0\|0 | 0\|1 | 0\|0 | 0\|1 |

Table 18: Comparison of different estimators LSE, PM, ES, IX, BanditNet, LS-LIN and OS accuracy for EMNIST and FMNIST with different qualities of logging policy ($\tau \in \{1, 10, 20\}$) and true / estimated propensity scores with $b \in \{5, 0.01\}$ and noisy reward with $P_f \in \{0.1, 0.5\}$. The best-performing result is highlighted in **bold** text, while the second-best result is colored in red for each scenario.

| Dataset | $\tau$ | $b$ | $P_f$ | LSE | PM | ES | IX | BanditNet | LS-LIN | OS | Logging Policy |
|---------|--------|-----|-------|-----|----|----|-----|-----------|--------|-----|----------------|
| EMNIST | 1 | − | − | $88.49 \pm 0.04$ | $\mathbf{89.19 \pm 0.03}$ | $88.61 \pm 0.06$ | $88.33 \pm 0.13$ | $66.58 \pm 6.39$ | $88.70 \pm 0.02$ | $88.71 \pm 0.26$ | 88.08 |
| | | 5 | − | $\mathbf{89.16 \pm 0.03}$ | $88.94 \pm 0.05$ | $88.48 \pm 0.03$ | $88.51 \pm 0.23$ | $65.10 \pm 0.69$ | $88.38 \pm 0.18$ | $88.70 \pm 0.15$ | 88.08 |
| | | 0.01 | − | $\mathbf{86.07 \pm 0.01}$ | $85.62 \pm 0.10$ | $85.71 \pm 0.04$ | $81.39 \pm 4.02$ | $66.55 \pm 3.11$ | $84.64 \pm 0.17$ | $84.59 \pm 0.09$ | 88.08 |
| | | − | 0.1 | $\mathbf{89.29 \pm 0.04}$ | $89.08 \pm 0.05$ | $88.45 \pm 0.09$ | $88.14 \pm 0.14$ | $59.90 \pm 3.78$ | $88.30 \pm 0.12$ | $88.74 \pm 0.09$ | 88.08 |
| | | − | 0.5 | $88.72 \pm 0.08$ | $\mathbf{88.78 \pm 0.03}$ | $87.27 \pm 0.10$ | $87.08 \pm 0.14$ | $56.95 \pm 3.06$ | $87.20 \pm 0.32$ | $88.06 \pm 0.09$ | 88.08 |
| | 10 | − | − | $88.59 \pm 0.03$ | $\mathbf{88.61 \pm 0.04}$ | $88.38 \pm 0.08$ | $87.43 \pm 0.19$ | $85.48 \pm 3.13$ | $88.58 \pm 0.08$ | $86.88 \pm 0.34$ | 79.43 |
| | | 5 | − | $88.42 \pm 0.07$ | $\mathbf{88.43 \pm 0.07}$ | $88.39 \pm 0.10$ | $88.39 \pm 0.06$ | $84.90 \pm 3.10$ | $88.23 \pm 0.27$ | $86.00 \pm 0.37$ | 79.43 |
| | | 0.01 | − | $\mathbf{82.15 \pm 0.21}$ | $80.85 \pm 0.29$ | $81.07 \pm 0.07$ | $77.49 \pm 2.77$ | $27.02 \pm 1.92$ | $78.43 \pm 3.13$ | $21.70 \pm 4.11$ | 79.43 |
| | | − | 0.1 | $\mathbf{88.29 \pm 0.06}$ | $88.22 \pm 0.02$ | $88.19 \pm 0.08$ | $87.93 \pm 0.35$ | $84.89 \pm 3.21$ | $87.50 \pm 0.17$ | $87.68 \pm 0.16$ | 79.43 |
| | | − | 0.5 | $88.71 \pm 0.16$ | $88.52 \pm 0.07$ | $84.42 \pm 0.34$ | $83.25 \pm 3.45$ | $63.35 \pm 13.39$ | $85.75 \pm 0.04$ | $\mathbf{89.09 \pm 0.05}$ | 79.43 |
| | 20 | − | − | $\mathbf{88.28 \pm 0.05}$ | $88.20 \pm 0.08$ | $87.96 \pm 0.34$ | $86.82 \pm 1.30$ | $83.69 \pm 3.32$ | $88.21 \pm 0.06$ | $80.64 \pm 0.25$ | 14.86 |
| | | 5 | − | $\mathbf{88.42 \pm 0.12}$ | $87.98 \pm 0.05$ | $88.27 \pm 0.33$ | $88.27 \pm 0.07$ | $86.82 \pm 0.17$ | $88.19 \pm 0.11$ | $79.31 \pm 0.61$ | 14.86 |
| | | 0.01 | − | $\mathbf{81.36 \pm 0.14}$ | $75.53 \pm 2.61$ | $73.45 \pm 2.78$ | $72.31 \pm 1.46$ | $26.92 \pm 2.51$ | $72.33 \pm 0.35$ | $11.12 \pm 0.39$ | 14.86 |
| | | − | 0.1 | $\mathbf{88.10 \pm 0.05}$ | $87.93 \pm 0.16$ | $87.69 \pm 0.22$ | $87.67 \pm 0.18$ | $81.73 \pm 3.09$ | $87.08 \pm 0.14$ | $82.95 \pm 0.31$ | 14.86 |
| | | − | 0.5 | $\mathbf{86.83 \pm 0.10}$ | $86.67 \pm 0.19$ | $84.01 \pm 0.32$ | $80.79 \pm 3.06$ | $75.20 \pm 3.01$ | $83.05 \pm 0.75$ | $86.03 \pm 0.48$ | 14.86 |
| FMNIST | 1 | − | − | $76.45 \pm 0.12$ | $73.33 \pm 2.67$ | $72.90 \pm 2.35$ | $69.12 \pm 0.26$ | $60.66 \pm 2.16$ | $69.29 \pm 0.19$ | $\mathbf{77.77 \pm 0.09}$ | 78.38 |
| | | 5 | − | $73.20 \pm 2.43$ | $75.07 \pm 0.27$ | $70.38 \pm 2.59$ | $70.80 \pm 2.38$ | $22.41 \pm 4.50$ | $69.33 \pm 0.20$ | $\mathbf{77.57 \pm 0.10}$ | 78.38 |
| | | 0.01 | − | $\mathbf{74.08 \pm 1.64}$ | $70.35 \pm 0.12$ | $57.93 \pm 2.66$ | $63.34 \pm 3.64$ | $30.20 \pm 8.17$ | $63.86 \pm 3.40$ | $37.57 \pm 3.16$ | 78.38 |
| | | − | 0.1 | $76.07 \pm 0.02$ | $74.54 \pm 0.02$ | $70.42 \pm 2.53$ | $70.58 \pm 2.47$ | $50.37 \pm 5.43$ | $70.41 \pm 2.20$ | $\mathbf{77.71 \pm 0.22}$ | 78.38 |
| | | − | 0.5 | $76.96 \pm 0.23$ | $74.03 \pm 0.30$ | $66.32 \pm 0.44$ | $66.66 \pm 1.41$ | $54.53 \pm 1.32$ | $66.57 \pm 2.76$ | $\mathbf{77.46 \pm 0.11}$ | 78.38 |
| | 10 | − | − | $\mathbf{76.14 \pm 0.11}$ | $74.42 \pm 0.17$ | $69.25 \pm 0.10$ | $70.69 \pm 2.39$ | $65.70 \pm 3.78$ | $69.31 \pm 0.24$ | $74.89 \pm 0.96$ | 21.43 |
| | | 5 | − | $\mathbf{75.42 \pm 0.16}$ | $74.79 \pm 0.15$ | $71.42 \pm 2.53$ | $69.21 \pm 0.25$ | $69.53 \pm 0.29$ | $70.15 \pm 2.53$ | $72.87 \pm 0.47$ | 21.43 |
| | | 0.01 | − | $\mathbf{74.04 \pm 0.15}$ | $60.77 \pm 0.09$ | $53.69 \pm 1.37$ | $63.57 \pm 3.91$ | $26.96 \pm 1.87$ | $60.65 \pm 3.83$ | $13.22 \pm 0.91$ | 21.43 |
| | | − | 0.1 | $\mathbf{76.78 \pm 0.23}$ | $73.91 \pm 0.13$ | $68.58 \pm 0.09$ | $68.07 \pm 0.18$ | $64.05 \pm 2.34$ | $68.10 \pm 0.58$ | $76.24 \pm 0.29$ | 21.43 |
| | | − | 0.5 | $\mathbf{77.66 \pm 0.17}$ | $74.02 \pm 0.05$ | $61.46 \pm 4.72$ | $62.60 \pm 0.16$ | $43.33 \pm 2.83$ | $61.35 \pm 1.83$ | $77.52 \pm 0.26$ | 21.43 |
| | 20 | − | − | $\mathbf{75.12 \pm 0.03}$ | $74.32 \pm 0.12$ | $69.26 \pm 0.09$ | $72.46 \pm 2.14$ | $64.92 \pm 3.82$ | $72.86 \pm 2.32$ | $65.78 \pm 1.10$ | 14.84 |
| | | 5 | − | $\mathbf{75.13 \pm 0.09}$ | $74.17 \pm 0.15$ | $69.23 \pm 0.46$ | $68.72 \pm 0.30$ | $62.41 \pm 4.24$ | $69.06 \pm 0.11$ | $63.53 \pm 1.70$ | 14.84 |
| | | 0.01 | − | $\mathbf{69.16 \pm 0.22}$ | $55.20 \pm 1.14$ | $60.91 \pm 2.75$ | $61.11 \pm 4.92$ | $28.23 \pm 2.18$ | $61.46 \pm 1.96$ | $13.04 \pm 4.76$ | 14.84 |
| | | − | 0.1 | $\mathbf{75.48 \pm 0.09}$ | $71.84 \pm 2.47$ | $65.41 \pm 4.23$ | $67.91 \pm 0.16$ | $65.21 \pm 2.93$ | $68.03 \pm 0.46$ | $70.90 \pm 0.26$ | 14.84 |
| | | − | 0.5 | $\mathbf{75.96 \pm 0.05}$ | $73.12 \pm 0.25$ | $61.79 \pm 3.13$ | $60.19 \pm 3.13$ | $55.13 \pm 0.15$ | $60.51 \pm 3.28$ | $73.32 \pm 0.81$ | 14.84 |

Table 19: Comparison of different model-based estimators DR, DR-OS, MRDR, SWITCH-DR, SWITCH-DR-LSE, DM and DR-LSE with LSE for EMNIST and FMNIST under a logging policy with $\tau = 10$, true / estimated propensity scores with $b \in \{5, 0.01\}$ and noisy reward with $P_f \in \{0.1, 0.5\}$. The best-performing result is highlighted in **bold** text, while the second-best result is colored in red for each scenario.

| Dataset | $\tau$ | $b$ | $P_f$ | DR-LSE | DR | DR-OS | MRDR | DR-Switch | DR-Switch-LSE | LSE | DM | Logging Policy |
|---|---|---|---|---|---|---|---|---|---|---|---|---|
| EMNIST | 10 | – | – | **88.79 ± 0.03** | 88.71 ± 0.07 | 87.79 ± 0.36 | 80.57 ± 4.00 | 79.40 ± 5.21 | 87.73 ± 0.31 | 88.59 ± 0.03 | 76.52 ± 2.68 | 79.43 |
| | | 5 | – | **88.67 ± 0.04** | 88.49 ± 0.13 | 87.83 ± 0.17 | 80.08 ± 4.62 | 79.28 ± 0.65 | 85.80 ± 3.40 | 88.42 ± 0.07 | 76.73 ± 4.95 | 79.43 |
| | | 0.01 | – | **83.30 ± 3.13** | 78.24 ± 0.57 | 80.53 ± 0.32 | 10.00 ± 0.01 | 74.81 ± 0.57 | 41.11 ± 2.87 | 82.15 ± 0.21 | 75.65 ± 0.29 | 79.43 |
| | | – | 0.1 | **88.51 ± 0.02** | 88.32 ± 0.16 | 87.50 ± 0.28 | 45.49 ± 9.14 | 75.28 ± 0.09 | 79.86 ± 0.64 | 88.29 ± 0.06 | 78.85 ± 2.69 | 79.43 |
| | | – | 0.5 | 85.88 ± 0.13 | 83.53 ± 0.54 | 85.46 ± 0.73 | 7.04 ± 4.18 | 72.76 ± 0.56 | 81.73 ± 0.23 | **88.71 ± 0.16** | 75.26 ± 2.39 | 79.43 |
| FMNIST | 10 | – | – | **80.15 ± 0.09** | 68.70 ± 5.12 | 63.66 ± 0.39 | 58.61 ± 3.89 | 54.20 ± 6.27 | 34.47 ± 0.02 | 76.14 ± 0.11 | 51.24 ± 4.16 | 79.43 |
| | | 5 | – | **79.64 ± 0.05** | 66.67 ± 3.50 | 64.80 ± 2.36 | 56.62 ± 1.52 | 56.61 ± 7.37 | 29.59 ± 3.83 | 75.42 ± 0.16 | 59.65 ± 3.13 | 79.43 |
| | | 0.01 | – | 55.10 ± 0.25 | 52.19 ± 3.84 | 60.92 ± 1.81 | 10.00 ± 0.01 | 63.35 ± 1.62 | 41.13 ± 2.84 | **74.04 ± 0.15** | 58.94 ± 4.18 | 79.43 |
| | | – | 0.1 | **79.91 ± 0.11** | 68.94 ± 0.35 | 63.19 ± 1.69 | 10.00 ± 0.01 | 57.54 ± 3.05 | 52.79 ± 4.04 | 76.78 ± 0.23 | 56.33 ± 7.70 | 79.43 |
| | | – | 0.5 | **79.14 ± 0.04** | 56.47 ± 7.08 | 56.72 ± 7.19 | 22.05 ± 4.50 | 59.54 ± 2.95 | 75.31 ± 0.55 | 77.66 ± 0.17 | 53.70 ± 7.19 | 79.43 |

## G.3. Model-based estimators

There are some approaches that utilise the estimation of reward. For example, in the direct method (DM), the reward is estimated from logged data via regression. In particular, an estimation of the reward function, $\widehat{r}(x, a)$, is learning from the LBF dataset $S$ using a regression. The objective function for DM can be represented as,

$$\frac{1}{n} \sum_{i=1}^{n} \sum_{a} \pi_\theta(a|x_i) \widehat{r}(a, x_i). \tag{87}$$

In the doubly-robust (DR) approach (Dudík et al., 2014) DM is combined with the IPS estimator and has a promising performance in the off-policy learning scenarios. The object function for doubly robust can be represented,

$$\frac{1}{n} \sum_{i=1}^{n} \sum_{a} \pi_\theta(a|x_i) \widehat{r}(a, x_i) + \frac{1}{n} \sum_{i=1}^{n} \frac{\pi_\theta(a_i|x_i)}{\pi_0(a_i|x_i)} (r_i - \widehat{r}(a, x_i)). \tag{88}$$

There are also some improvements regarding the DR, including DR based on optimistic Shrinkage (DR-OS) (Su et al., 2020), DR-Switch (Wang et al., 2017) and MRDR (Farajtabar et al., 2018).

As these methods are based on the estimation of reward, we consider them as model-based methods. Inspired by the DR method, we combine the LSE estimator with the DM method (DR-LSE)

$$\frac{1}{n} \sum_{i=1}^{n} \sum_{a} \pi_\theta(a|x_i) \widehat{r}(a, x_i) + \frac{1}{\lambda} \log \left( \frac{1}{n} \sum_{i=1}^{n} \exp \left( \lambda \frac{\pi_\theta(a_i|x_i)}{\pi_0(a_i|x_i)} (r_i - \widehat{r}(a, x_i)) \right) \right). \tag{89}$$

We also combine, LSE with DR-Switch as (DR-Switch-LSE) where the IPS estimator in DR-Switch is replaced with LSE estimator.

In this section, we aim to show that the combination of our LSE estimator with the DR method as a model-based method can improve the performance of these methods. For our experiments, we use the same experiment setup as described in App. F. We compare model-based methods, DM, DR and DR-LSE, DR-Switch, DR-OS, and DR-Switch-LSE with our LSE estimator. The results are shown in Table 19. We observed that DR-LSE outperforms the standard DR in many scenarios.

## G.4. Real-World dataset

We applied our method to two real-world scenarios: one in recommendation systems and the other in Natural Language Processing.

### G.4.1. RECOMMANDATION SYSTEMS

We applied our method to the Kuairec, a public real-world recommendation system dataset ((Gao et al., 2022)). This dataset is gathered from the recommendation logs of the video-sharing mobile app Kuaishou. In each instance, a user watches an item (video) and the watch duration divided by the entire duration of the video is reported. We use the same procedure as (Zhang et al., 2023a) to prepare the logged bandit dataset. We also use the same architecture for the logging policy and the learning policy, with some modifications in the hidden size and number of layers of the deep models. We use separate models for the logging and learning policies. We first train the logging policy using cross-entropy loss and fix it to use as the propensity score estimator for the training of the OPL models. We report Precision@K, and NDCG@K for K=1, 3, 5, 10. Recall@K is very low for small K values because the number of positive items for each use of much more than K. For each method, we use grid search to find the hyperparameter that maximizes the Precision@1 in the validation dataset. The comparison of different estimators is presented in Table 20. We can observe that in Precision@1, Precision@3, Precision@10, NDCG@1, NDCG@3 and NDCG@10, we have the best performance.

Table 20: Comparison of different estimators LSE, PM, ES, IX, LS-LIN, OS and SNIPS in different metrics. The best-performing result is highlighted in **bold** text, while the second-best result is colored in red for each scenario.

| Dataset | Estimator | Precision@1 | Precision@3 | Precision@5 | Precision@10 | NDCG@1 | NDCG@3 | NDCG@5 | NDCG@10 |
|---------|-----------|-------------|-------------|-------------|--------------|--------|--------|--------|---------|
|         | PM        | 0.8885      | 0.5723      | 0.5201      | 0.4275       | 0.8585 | 0.6551 | 0.5932 | 0.4988  |
|         | SNIPS     | 0.0289      | 0.6177      | 0.5995      | 0.6462       | 0.0289 | 0.4981 | 0.5226 | 0.5830  |
|         | IX        | 0.8794      | 0.5824      | 0.6355      | 0.6586       | 0.8794 | 0.6164 | 0.6410 | 0.6548  |
| KuaiRec | ES        | 0.8951      | 0.7495      | **0.7187**  | 0.6644       | 0.8951 | 0.7787 | **0.7483** | 0.7006  |
|         | OS        | 0.8993      | 0.3215      | 0.2015      | 0.1403       | 0.8993 | 0.4381 | 0.3227 | 0.2378  |
|         | LS-LIN    | 0.8836      | 0.6680      | 0.7159      | 0.6904       | 0.8836 | 0.7159 | 0.7368 | 0.7108  |
|         | LSE       | **0.9257**  | **0.7534**  | 0.6999      | **0.7206**   | **0.9257** | **0.7917** | 0.7441 | **0.7431** |

### G.4.2. NLP

We run experiments on **PubMed 200K RCT** dataset, which is a collection of 200,000 abstracts of medical articles. Parts of each abstract are classified into one of the classes of *BACKGROUND, OBJECTIVE, METHOD, RESULT, CONCLUSION*. This is a short text classification that can be viewed as a bandit problem. We apply the same conversion method as we did on image classification datasets in OPL experiments. In Table 21 we can observe the result of our method compared to other state-of-the-art methods.

Table 21: Comparison of different methods LSE, IX, LS, OS, ES, and PM under logging policies with $\tau = 1, 10$ on the PubMed RCT dataset. The best-performing result is highlighted in **bold** text, while the second-best result is colored in red for each scenario.

| Dataset | $\tau$ | LSE | LS | IX | ES | PM | OS |
|---------|--------|-----|-----|-----|-----|-----|-----|
| RCT | 1 | **67.16** | 58.57 | 58.58 | 58.48 | 67.06 | 62.29 |
|     | 10 | 70.17 | 58.59 | 58.56 | 58.60 | **70.32** | 66.15 |

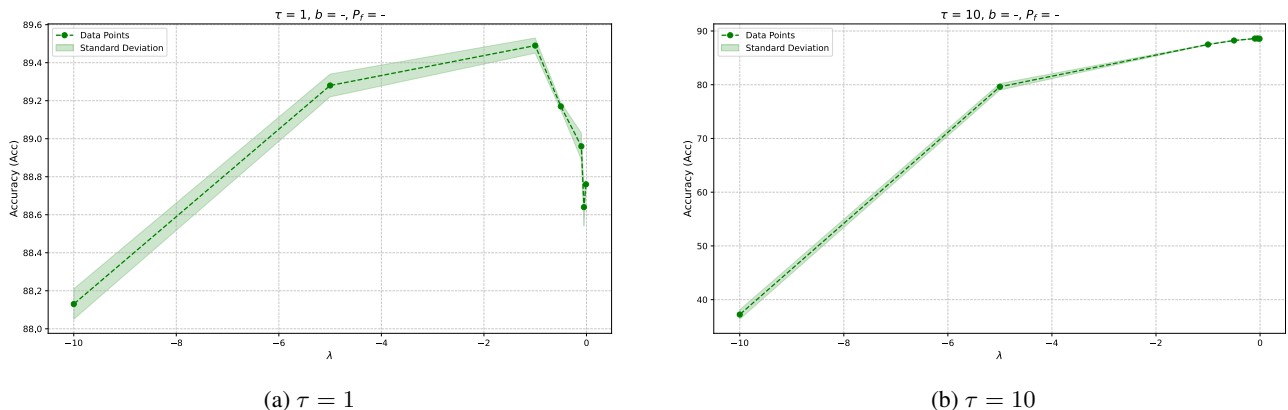

(a) $\tau = 1$

(b) $\tau = 10$

Figure 2: Accuracy of the LSE estimator over different values of $\lambda$ for true propensity score and reward. (a) $\tau = 1$. (b) $\tau = 10$.

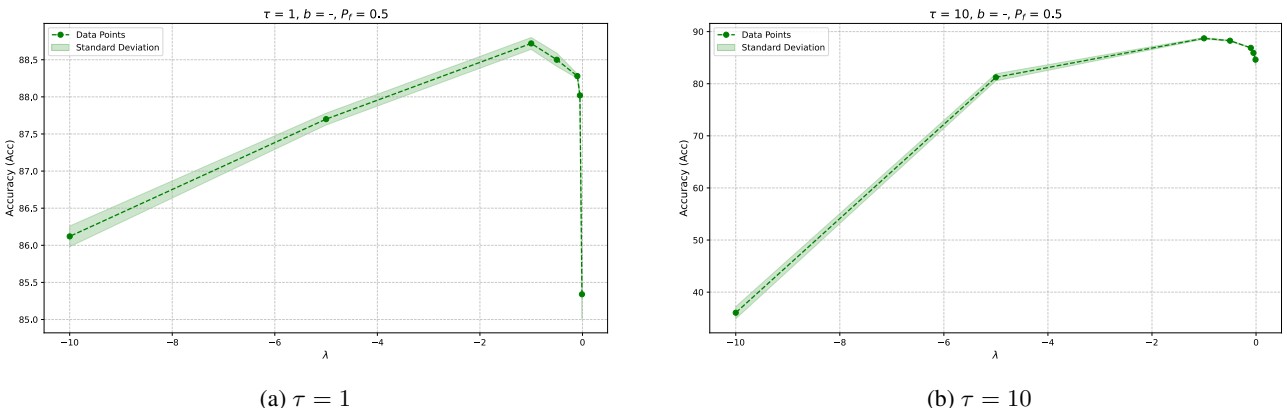

(a) $\tau = 1$

(b) $\tau = 10$

Figure 3: Plots of Accuracy of the LSE estimator over different values of $\lambda$ for true propensity score and noisy reward with $P_f = 0.5$. (a) $\tau = 1$. (b) $\tau = 10$.

### G.5. Sample number effect

We also conduct experiments on our LSE estimator and PM estimator to examine the effect of limited training samples in the OPL scenario. For this purpose, we considered different ratios of training LBF dataset, $R_n \in \{1, 0.5, 0.2, 0.05\}$. The results are shown in Table 22. We observed that reducing $R_n$ decreased the accuracy for both estimators. However, our LSE estimator demonstrated robust performance under different ratios of training LBF dataset, $R_n$. Therefore, for small-size LBF datasets, we can apply the LSE estimator for off-policy learning.

### G.6. $\lambda$ Effect

### G.6.1. OPL

The impact of $\lambda$ across various scenarios and $\tau$ values was investigated using the experimental setup described in Appendix F for the EMNIST dataset. Figure 2 illustrates the accuracy of the LSE estimator for $\tau \in \{1, 10\}$. For $\tau = 1$, corresponding to a logging policy with higher accuracy, an optimal $\lambda$ value of approximately $-1.5$ was observed. In contrast, for $\tau = 10$, representing a logging policy with lower accuracy, the optimal $\lambda$ approached zero. Additionally, in scenarios with noisy rewards Fig.3, both $\tau = 1$ and $\tau = 10$, we observed an optimal $\lambda$ values larger $-2$. As for $\tau = 1$, the logging policy has higher accuracy, the effect of noisy reward should be canceled by larger $|\lambda|$. However, for $\tau = 10$, we need a smaller $|\lambda|$.

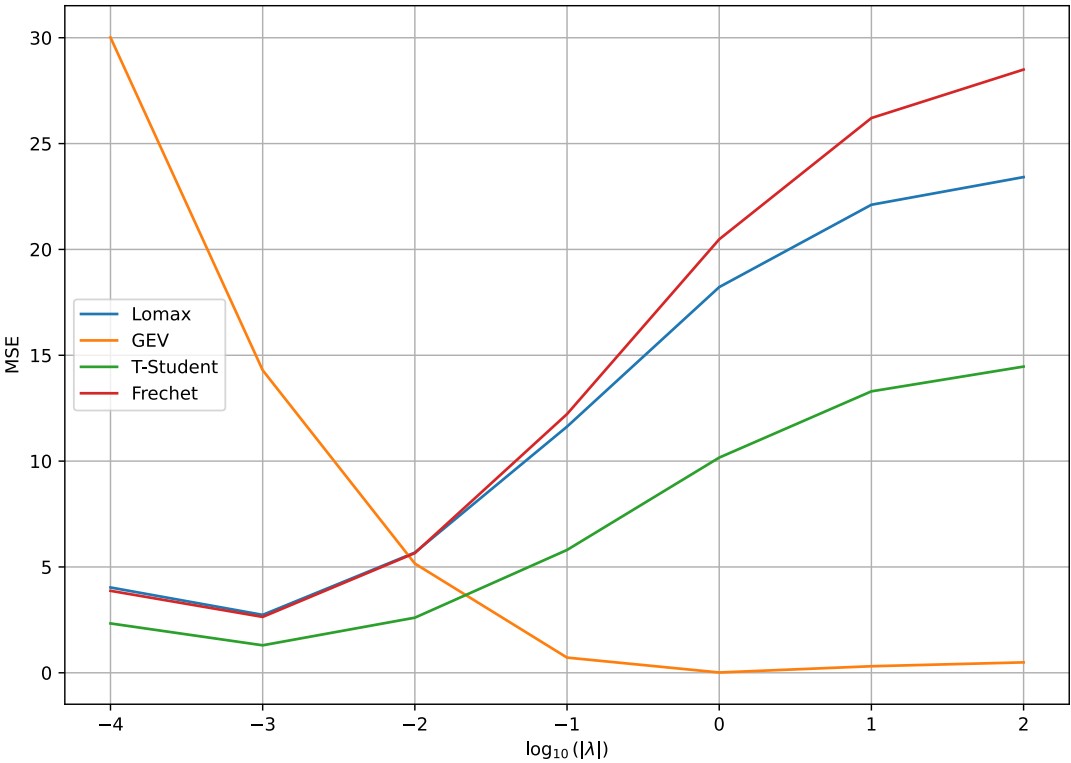

Figure 4: The effect of $\lambda$ on different families of distributions: GEV, Student's t, Frechet, and Lomax

### G.6.2. OPE

We also conduct experiments to investigate the effect of $\lambda$ in OPE through the synthetic setup. In order to analyze the sensitivity w.r.t. $\lambda$ in OPE, we test on different heavy-tailed distributions and keep the same setting as Section G.1.1. Here we change $|\lambda|$ exponentially from $10^{-4}$ to $10^2$ and observe the MSE of LSE with the selected value for $\lambda$. Figure 4 illustrates the MSE value over the different values of $\lambda$ for different families of distributions. As we can see, the trend of MSE value is different for GEV, and MSE monotonically decreases as $\lambda$ increases. However, in other distributions, higher values of $\lambda$ result in relatively worse performance, and there is an intermediate value ($10^{-3}$) for which the lowest MSE is achieved. Nevertheless, $10^{-2}$ is an acceptable choice of $\lambda$ for all distribution families.

### G.7. Data-independent Selection of $\lambda$

Although we use grid search to tune the $\lambda$ in our algorithm, inspired by Proposition 5.4, we can select the following value,

$$\lambda^* = \frac{1}{n^{1/(\epsilon+1)}}, \tag{90}$$

where $n$ is the number of samples. This selection is independent of the data values and ensures only asymptotical guarantees. With such a selection we have a regret rate of $O(n^{-\epsilon/(1+\epsilon)})$. We test and evaluate our selection in OPL and OPE. We also examine a data-driven approach for selecting $\lambda$ in Section D.5.

### G.7.1. $\lambda$ SELECTION FOR OPL

We have tested $\lambda^*$ on EMNIST dataset. In OPL experiments we have truncated the propensity score to $0.001$ in order to avoid numerical overflow. Hence, our distributions are effectively heavy-tailed with $\epsilon = 1$, leading to $\lambda^* = \frac{1}{\sqrt{n}}$. We

change $n = 512, 256, 128, 64, 16$ with corresponding values $\lambda^* \in \{0.044, 0.0625, 0.088, 0.125, 0.25\}$ which its results are presented in the Table 23. Note that because we use stochastic gradient descent in training, here $n$ is the batch size. We can observe that the suggested value of $\lambda^* = \frac{1}{\sqrt{n}}$ does not only have a theoretical estimation bound of $O(\frac{1}{\sqrt{n}})$ (according to Proposition 5.4), but also achieves reasonable performance in experiments.

### G.7.2. $\lambda$ SELECTION FOR OPE

We tested our $\lambda$ selection in the OPE setting with Lomax distributions. We changed the number of samples and set $n = 100, 500, 1K, 5K, 10K, 50K, 500K$ and tested all estimators as we as LSE with selected $\lambda = \lambda^\star$. The results are illustrated at Tables 24, and 25. The first observation is that in all settings, the selected $\lambda^\star$ outperforms all other estimators, except LS which loses in $n \leq 5000$ experiments with a very small margin and is not significantly worse than the $\lambda$ found by grid search.

Another critical observation is that as the number of samples increases, the selected $\lambda$ works better than compared to other methods, even LSE with $\lambda$ found by grid-search. In $n = 100K$, not only $\lambda^\star$ perform the best, but also the $\lambda$ found by grid-search falls behind IPS-TR and ES. This shows the significance of selective $\lambda$ when the number of samples is large.

The third observation is the lower performance of $\lambda^\star$ when we have a very small number of samples, e.g. $n = 100$. This also conforms to our theoretical results, as upper and lower bounds on estimation and regret bounds in Theorem D.2, Theorem D.3 and Theorem 5.3 requires a minimum number of samples as an assumption.

Table 22: Comparison of LSE and PM accuracy for EMNIST dataset with different ratio of training LBF dataset ($R_n \in \{1, 0.5, 0.2, 0.05\}$) and true / estimated propensity scores with $b \in \{5, 0.01\}$ and noisy reward with $P_f \in \{0.1, 0.5\}$. The best-performing result is highlighted in **bold** text.

| Dataset | $\tau$ | $R_n$ | $b$ | $P_f$ | **LSE** | **PM** | **Logging Policy** |
|---|---|---|---|---|---|---|---|
| EMNIST | 1 | 1 | $-$ | $-$ | $\mathbf{88.49 \pm 0.04}$ | $\mathbf{89.19 \pm 0.03}$ | 88.08 |
| | | | 0.01 | $-$ | $\mathbf{86.07 \pm 0.01}$ | $85.62 \pm 0.10$ | 88.08 |
| | | | $-$ | 0.5 | $88.72 \pm 0.08$ | $\mathbf{88.78 \pm 0.03}$ | 88.08 |
| | | 0.5 | $-$ | $-$ | $\mathbf{87.79 \pm 0.08}$ | $86.42 \pm 0.11$ | 88.08 |
| | | | 0.01 | $-$ | $\mathbf{81.13 \pm 0.08}$ | $48.70 \pm 15.46$ | 88.08 |
| | | | $-$ | 0.5 | $\mathbf{86.24 \pm 0.07}$ | $85.17 \pm 0.36$ | 88.08 |
| | | 0.2 | $-$ | $-$ | $\mathbf{83.76 \pm 0.25}$ | $74.57 \pm 1.01$ | 88.08 |
| | | | 0.01 | $-$ | $\mathbf{67.64 \pm 3.89}$ | $23.18 \pm 5.02$ | 88.08 |
| | | | $-$ | 0.5 | $\mathbf{80.39 \pm 0.19}$ | $69.54 \pm 0.65$ | 88.08 |
| | | 0.05 | $-$ | $-$ | $\mathbf{70.16 \pm 2.44}$ | $53.51 \pm 2.77$ | 88.08 |
| | | | 0.01 | $-$ | $\mathbf{36.06 \pm 0.62}$ | $15.56 \pm 3.21$ | 88.08 |
| | | | $-$ | 0.5 | $\mathbf{50.06 \pm 2.10}$ | $47.57 \pm 5.19$ | 88.08 |
| | 10 | 1 | $-$ | $-$ | $88.59 \pm 0.03$ | $\mathbf{88.61 \pm 0.04}$ | 79.43 |
| | | | 0.01 | $-$ | $\mathbf{82.15 \pm 0.21}$ | $80.85 \pm 0.29$ | 79.43 |
| | | | $-$ | 0.5 | $\mathbf{88.71 \pm 0.16}$ | $88.52 \pm 0.07$ | 79.43 |
| | | 0.5 | $-$ | $-$ | $\mathbf{86.30 \pm 0.04}$ | $86.02 \pm 0.06$ | 79.43 |
| | | | 0.01 | $-$ | $\mathbf{75.02 \pm 2.67}$ | $28.12 \pm 1.94$ | 79.43 |
| | | | $-$ | 0.5 | $\mathbf{86.61 \pm 0.08}$ | $83.21 \pm 0.10$ | 79.43 |
| | | 0.2 | $-$ | $-$ | $80.67 \pm 0.35$ | $\mathbf{80.83 \pm 0.22}$ | 79.43 |
| | | | 0.01 | $-$ | $\mathbf{53.32 \pm 1.47}$ | $17.03 \pm 0.30$ | 79.43 |
| | | | $-$ | 0.5 | $\mathbf{80.89 \pm 0.19}$ | $73.42 \pm 1.14$ | 79.43 |
| | | 0.05 | $-$ | $-$ | $\mathbf{48.51 \pm 0.81}$ | $42.27 \pm 1.48$ | 79.43 |
| | | | 0.01 | $-$ | $\mathbf{34.15 \pm 0.61}$ | $14.70 \pm 2.20$ | 79.43 |
| | | | $-$ | 0.5 | $\mathbf{56.64 \pm 2.40}$ | $41.75 \pm 1.95$ | 79.43 |

Table 23: Comparison of accuracy (%) for different $\lambda$ values and sample sizes $n$

| $\lambda \backslash n$ | 16 | 64 | 128 | 256 | 512 |
|---|---|---|---|---|---|
| 0.01 | $92.83 \pm 0.10$ | $91.52 \pm 0.01$ | $90.26 \pm 0.02$ | $88.71 \pm 0.26$ | $85.43 \pm 0.44$ |
| 0.1 | $\mathbf{92.83 \pm 0.01}$ | $91.45 \pm 0.01$ | $90.37 \pm 0.02$ | $88.93 \pm 0.10$ | $85.50 \pm 0.58$ |
| 1 | $92.66 \pm 0.01$ | $\mathbf{91.66 \pm 0.02}$ | $\mathbf{90.76 \pm 0.02}$ | $\mathbf{89.54 \pm 0.01}$ | $\mathbf{87.79 \pm 0.01}$ |
| 10 | $91.33 \pm 0.01$ | $89.48 \pm 0.09$ | $88.86 \pm 0.05$ | $88.03 \pm 0.03$ | $86.73 \pm 0.03$ |
| $\lambda^*$ | $92.78 \pm 0.01$ | $91.52 \pm 0.05$ | $90.38 \pm 0.05$ | $88.83 \pm 0.02$ | $85.09 \pm 0.51$ |

Table 24: Summary of Bias, Variance, and MSE for Different Estimators for Lomax OPE experiments. We change the number of samples $n = 100, 500, 1K, 10K$ and report the metrics for PM, ES, LSE, LSE($\lambda^*$), LS, LS-LIN, OS, IPS-TR, IX, SNIPS

| $n$ | Estimator | Bias | Var | MSE |
|---|---|---|---|---|
| | PM | $-0.2623$ | 30.6419 | 30.7106 |
| | ES | 2.2894 | 0.0247 | 5.2662 |
| | LSE | 0.6194 | 0.3967 | **0.7803** |
| 100 | LSE($\lambda^*$) | 0.9144 | 0.1952 | 1.0314 |
| | LS | 0.6386 | 0.5336 | 0.9414 |
| | LS-LIN | 2.0377 | 0.0167 | 4.1689 |
| | OS | 0.4485 | 22.7449 | 22.9461 |
| | IPS-TR | $-0.0144$ | 24.8212 | 24.8214 |
| | IX | 1.9517 | 0.0171 | 3.8264 |
| | SNIPS | $-0.0483$ | 25.8348 | 25.8371 |
| | PM | $-0.2002$ | 3.1605 | 3.2006 |
| | ES | 0.0415 | 2.5603 | 2.5620 |
| | LSE | 0.2221 | 0.3375 | **0.3869** |
| 500 | LSE($\lambda^*$) | 0.5542 | 0.0984 | 0.4055 |
| | LS | 0.2309 | 0.3449 | 0.3983 |
| | LS-LIN | 2.0377 | 0.0033 | 4.1557 |
| | OS | 0.42724 | 7.6075 | 7.7901 |
| | IPS-TR | 0.0415 | 2.5603 | 2.5620 |
| | IX | 1.9536 | 0.0035 | 3.8200 |
| | SNIPS | 0.0347 | 2.6865 | 2.6877 |
| | PM | $-0.2379$ | 4.8325 | 4.8891 |
| | ES | 0.0076 | 3.9145 | 3.9145 |
| | LSE | 0.2262 | 0.1720 | **0.2231** |
| 1000 | LSE($\lambda^*$) | 0.4335 | 0.0712 | 0.2591 |
| | LS | 0.2270 | 0.1751 | 0.2266 |
| | LS-LIN | 2.0368 | 0.0016 | 4.1502 |
| | OS | 0.4178 | 4.0558 | 4.2303 |
| | IPS-TR | 0.0076 | 3.9145 | 3.9145 |
| | IX | 1.9536 | 0.0018 | 3.8186 |
| | SNIPS | 0.0040 | 4.0054 | 4.0054 |
| | PM | $-0.2428$ | 3.7591 | 3.8180 |
| | ES | 0.0032 | 3.0449 | 3.0449 |
| | LSE | 0.2277 | 0.0343 | **0.0862** |
| 5000 | LSE($\lambda^*$) | 0.2448 | 0.0319 | 0.0919 |
| | LS | 0.2334 | 0.0342 | 0.0887 |
| | LS-LIN | 2.0374 | 0.0003 | 4.1513 |
| | OS | 0.4626 | 0.4477 | 0.6617 |
| | IPS-TR | 0.0032 | 3.0449 | 3.0449 |
| | IX | 1.9535 | 0.0004 | 3.8166 |
| | SNIPS | 0.0025 | 2.9976 | 2.9976 |
| | PM | $-0.2318$ | 0.4702 | 0.5239 |
| | ES | 0.0131 | 0.3809 | 0.3811 |
| | LSE | 0.2254 | 0.0171 | 0.0679 |
| 10000 | LSE($\lambda^*$) | 0.1867 | 0.0212 | **0.0560** |
| | LS | 0.2341 | 0.0173 | 0.0721 |
| | LS-LIN | 2.0376 | 0.0002 | 4.1518 |
| | OS | 0.4336 | 0.5004 | 0.6884 |
| | IPS-TR | 0.0131 | 0.3809 | 0.3811 |
| | IX | 1.9536 | 0.0002 | 3.8168 |
| | SNIPS | 0.0123 | 0.3830 | 0.3832 |

Table 25: Summary of Bias, Variance, and MSE for Different Estimators for Lomax OPE experiments. We change the number of samples $n = 50K, 100K$ and report the metrics for PM, ES, LSE, LSE($\lambda^*$), LS, LS-LIN, OS, IPS-TR, IX, SNIPS

| $n$ | Estimator | Bias | Var | MSE |
|---|---|---|---|---|
| 50000 | PM | $-0.2418$ | 0.2152 | 0.2736 |
| | ES | 0.0040 | 0.1743 | 0.1743 |
| | LSE | 0.2261 | 0.0033 | 0.0544 |
| | LSE($\lambda^*$) | 0.1020 | 0.0085 | **0.0189** |
| | LS | 0.2324 | 0.0035 | 0.0574 |
| | LS-LIN | 2.0374 | 0.0000 | 4.1512 |
| | OS | 0.3872 | 5.0487 | 5.1987 |
| | IPS-TR | 0.0040 | 0.1743 | 0.1743 |
| | IX | 1.9538 | 0.0000 | 3.8172 |
| | SNIPS | 0.0040 | 0.1745 | 0.1746 |
| 100000 | PM | $-0.2347$ | 0.0633 | 0.1184 |
| | ES | 0.0105 | 0.0513 | 0.0514 |
| | LSE | 0.2267 | 0.0017 | 0.0531 |
| | LSE($\lambda^*$) | 0.0790 | 0.0056 | **0.0119** |
| | LS | 0.2338 | 0.0017 | 0.0564 |
| | LS-LIN | 2.0375 | 0.0000 | 4.1516 |
| | OS | 0.4294 | 0.2179 | 0.4021 |
| | IPS-TR | 0.0105 | 0.0513 | 0.0514 |
| | IX | 1.9538 | 0.0000 | 3.8172 |
| | SNIPS | 0.0105 | 0.0515 | 0.0516 |

## G.8. Data-driven $\lambda$ Selection

In this section, we evaluate the effectiveness of our data-driven $\lambda$ approach (presented in Appendix D.5) under both clean and noisy reward conditions and provide a strategy to select $\lambda$ based on data. Our approach automatically determines a data-driven $\lambda$ parameter directly from the data, eliminating the need for manual hyperparameter tuning.

### G.8.1. DATA-DRIVEN $\lambda$ FOR OPL

**Data-driven $\lambda$ on normal setting**: when the reward is observed without any error, we use the derived $\lambda_{\mathrm{D}}$ suggested in App.D.5. We experimented with the EMNIST and FMNIST datasets for $\tau = 1, 10$. Table 26 compares the result of data driven $\lambda_{\mathrm{D}}$ with optimal $\lambda$ found by grid-search.

Table 26: Accuracy of LSE and LSE-$\lambda_{\mathrm{D}}$ on EMNIST and FMNIST datasets for $\tau = 1, 10$

| Dataset | $\tau$ | LSE | LSE-$\lambda_{\mathrm{D}}$ |
|---------|--------|-----|-----------------------------|
| EMNIST | 1.0 | $88.49 \pm 0.04$ | $89.17 \pm 0.03$ |
| | 10 | $88.59 \pm 0.03$ | $88.56 \pm 0.04$ |
| FMNIST | 1.0 | $76.45 \pm 0.12$ | $75.81 \pm 0.08$ |
| | 10 | $76.14 \pm 0.11$ | $74.96 \pm 0.12$ |

**Data-driven $\lambda$ on noisy setting**: When the observed reward has errors, we use suggested $\lambda_{\mathrm{ND}}$ in Equation 58 at App.D.5. We experimented with the EMNIST dataset and $\tau = 1$ and different probabilities of noise $P_f = 0.2, 0.5, 0.8$. Table 27 compares the result of data-driven $\lambda_{\mathrm{ND}}$ with optimal $\lambda$ found by grid-search.

Table 27: Accuracy of LSE and LSE-$\lambda_{\mathrm{ND}}$ on EMNIST for $P_f = 0.2, 0.5, 0.8$

| Dataset | $P_f$ | LSE | LSE-$\lambda_{\mathrm{ND}}$ |
|---------|-------|-----|------------------------------|
| | 0.2 | $88.82 \pm 0.05$ | $88.85 \pm 0.09$ |
| EMNIST | 0.5 | $87.65 \pm 0.10$ | $87.5 \pm 0.06$ |
| | 0.8 | $91.86 \pm 0.05$ | $83.96 \pm 0.22$ |

As we can observe, the data-driven selection for $\lambda$, keeps up with the good performance of grid-search, making it a reliable strategy to choose the hyperparameter of our method, without the requirement of any additional experiments or search. The only exception is for the setting with $P_f = 0.8$, in which we observe that the data-driven falls behind by a significant value. This is due to the high noise in the reward, which makes the estimations in the simplified objective for $\lambda_{\mathrm{ND}}$ less accurate.

### G.8.2. $\lambda$ SELECTION STRATEGY

In this section, we provide a general strategy for the selection of $\lambda$ based on experiments on OPE. Since an accurate and robust estimation of the average reward is necessary for appropriate training of the target policy, methods that can perform well in OPE and provide robust estimators in extreme and heavy-tailed scenarios are more reliable for OPL. We compare three different methods for the selection of $\lambda$. First, $\lambda$ is found by grid search which provides the best MSE. Second, $\lambda^*$ is found by the data-driven suggestion in App.D.5. In the third method, we select $\lambda$ uniformly randomly from $[0, 1]$, $\tilde{\lambda} \sim \mathrm{Uniform}(0, 1)$. This method shows the performance of LSE by choosing random $\lambda$ as a hyperparameter. We test these methods on the Lomax scenario where we have the more challenging heavy-tailed (for $\epsilon \neq 1$) condition. The MSE of each method for the same setting of parameters as in the original OPE experiments and for $n = 1K, 10K, 100K$ is reported in table 28.

Our experimental results demonstrate that LSE with grid-searched $\lambda$ consistently achieves the lowest MSE across all experimental configurations. The data-driven $\lambda$ selection approach exhibits strong performance, ranking second in scenarios with larger sample sizes ($n = 10K, 100K$). For smaller samples ($n = 1K$), random $\lambda$ selection occasionally outperforms the data-driven approach. Notably, LSE maintains robust variance control under heavy-tailed distributions even with randomly selected $\lambda$ values. The performance gap between data-driven and random $\lambda$ selection widens significantly as the

Table 28: MSE of LSE with fine-tuned, data-driven and random $\lambda$ for $\beta = 1.0, 1.5, 2.0$. The experiment was run $100000$ times with different values of $\alpha$, $\alpha'$, and $\beta$.

| $\beta$ | $\alpha$ | $\alpha'$ | **Estimator** | $n = 1K$ | $n = 10K$ | $n = 100K$ |
|---|---|---|---|---|---|---|
| | | | LSE | **0.006** | **0.0009** | **0.0001** |
| | | 1.0 | LSE-$\lambda^*$ | 0.049 | 0.0076 | 0.0009 |
| | | | LSE-$\tilde{\lambda}$ | 0.131 | 0.131 | 0.131 |
| 0.5 | 1.0 | | LSE | **0.041** | **0.0.008** | **0.0039** |
| | | 1.5 | LSE-$\lambda^*$ | 0.463 | 0.138 | 0.03 |
| | | | LSE-$\tilde{\lambda}$ | 0.449 | 0.449 | 0.449 |
| | | | LSE | **0.105** | **0.033** | **0.026** |
| | | 2.0 | LSE-$\lambda^*$ | 1.044 | 0.450 | 0.148 |
| | | | LSE-$\tilde{\lambda}$ | 0.764 | 0.762 | 0.760 |
| | | | LSE | **0.014** | **0.002** | **0.0003** |
| | | 1.0 | LSE-$\lambda^*$ | 0.110 | 0.018 | 0.002 |
| | | | LSE-$\tilde{\lambda}$ | 0.398 | 0.398 | 0.394 |
| 1.0 | 1.5 | | LSE | **0.093** | **0.020** | **0.012** |
| | | 1.5 | LSE-$\lambda^*$ | 1.042 | 0.311 | 0.067 |
| | | | LSE-$\lambda_r$ | 1.227 | 1.226 | 1.223 |
| | | | LSE | **0.211** | **0.088** | **0.0754** |
| | | 2.0 | LSE-$\lambda^*$ | 3.05 | 1.013 | 0.333 |
| | | | LSE-$\tilde{\lambda}$ | 1.991 | 1.99 | 1.985 |
| | | | LSE | **0.0463** | **0.005** | **0.0014** |
| | | 1.0 | LSE-$\lambda^*$ | 0.3071 | 0.048 | 0.0054 |
| | | | LSE-$\tilde{\lambda}$ | 1.550 | 1.548 | 1.552 |
| 2.0 | 2.5 | | LSE | **0.222** | **0.058** | **0.052** |
| | | 1.5 | LSE-$\lambda^*$ | 2.894 | 0.864 | 0.187 |
| | | | LSE-$\tilde{\lambda}$ | 4.242 | 4.236 | 4.246 |
| | | | LSE | **0.548** | **0.313** | **0.289** |
| | | 2.0 | LSE-$\lambda^*$ | 6.530 | 2.817 | 0.928 |
| | | | LSE-$\tilde{\lambda}$ | 6.534 | 6.531 | 6.535 |

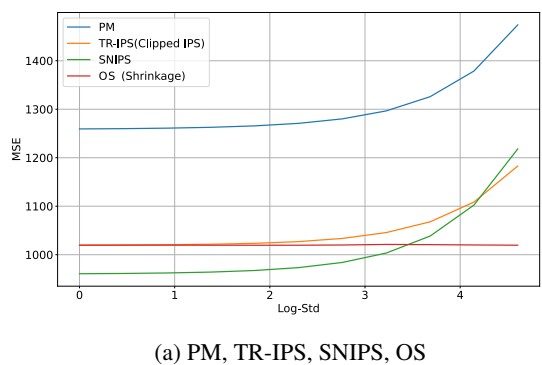

(a) PM, TR-IPS, SNIPS, OS

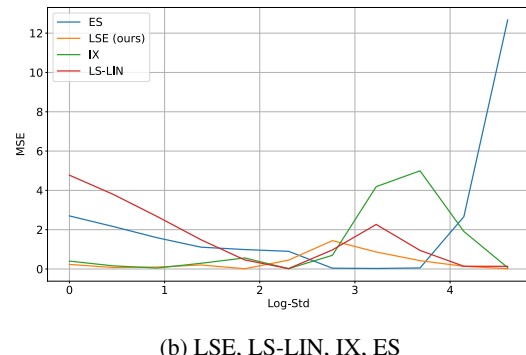

(b) LSE, LS-LIN, IX, ES

Figure 5: MSE of the PM, TR-IPS, SNIPS, OS, LS-LIN, IX, OS, ES, and LSE estimators over different values of $\log \sigma$

sample size increases, suggesting a clear strategy for parameter selection: while the estimator remains robust to arbitrary $\lambda$ choices, the data-driven approach becomes increasingly reliable with larger sample sizes.

- If $n$ is small (e.g. $n \approx 1000$), we have fewer computational concerns, and a grid search based on the performance on a validation set can find an appropriate $\lambda$ for our problem.

- For larger values of $n$, we can hold to the data-driven proposal of $\lambda$ which gives a comparable performance with the grid-search method.

Another hint about the selection of $\lambda$ is that for problems where the variance of the importance weights of the unbounded behaviour of the reward function is not an issue, a very small $\lambda$ (e.g. $\lambda = 0.01$) can be a better option because as $\lambda \to 0$, LSE tends to vanilla IPS. For heavy-tailed problems, selecting bigger $\lambda$ values around $1$ can lead to better performance.

### G.9. OPE with noise

Here we discuss the performance of estimators in OPE when reward noise is available. In all experiments, the number of samples is $1000$ and the number of trials is $100K$.

### G.9.1. GAUSSIAN SETTING

We run the same experiments as mentioned in Section 6.1 by adding noise to the observed reward. We add a positive Gaussian noise,
$$\tilde{R}(S, A) = R(S, A) + |W| : W \sim \mathcal{N}(0, \sigma^2).$$
where $\tilde{R}(S, A)$ is noisy reward function. We increase $\sigma$ from 1 to 100 and observe the behaviour of different estimators under the noise. We report the MSE of different estimators. There is a discrepancy between the performance of different estimators. LSE, LS, LSE-LIN, IX, and ES demonstrated robust performance under high noise conditions, while PM, TR-IPS, SNIPS, and OS exhibited substantially higher MSE values, often differing by several orders of magnitude from the better-performing estimators. We draw the MSE of these two groups against $\log \sigma$ in Figure 5. We observe that ES, LSE, IX, and LS-LIN are better suited for the noisy scenario. Also, we observe that ES is more sensitive to the increase of the variance of the noise. We also investigate the distributional form of the estimators with the same levels of noise. Estimators other than LSE, LS-LIN, and IX keep proposing outlier estimations. But these three estimators stay stable in this setting and are compared in Figure 6 for two levels of noise. Among these three estimators, LSE can keep a low bias with almost the same variance in comparison to IX and LS-LIN, hence leading to the lowest MSE.

### G.9.2. LOMAX SETTING

When we examine the Lomax setting, the estimators' performance deteriorates as we introduce heavier-tailed noise distributions. To test this, we add Pareto-distributed (with parameter $\alpha$) noise to the reward, varying the parameter $\alpha$ from $1.05$ to $2.0$. The parameter $\alpha$ controls the tail weight of the distribution, with values closer to $1$ producing heavier tails. Our

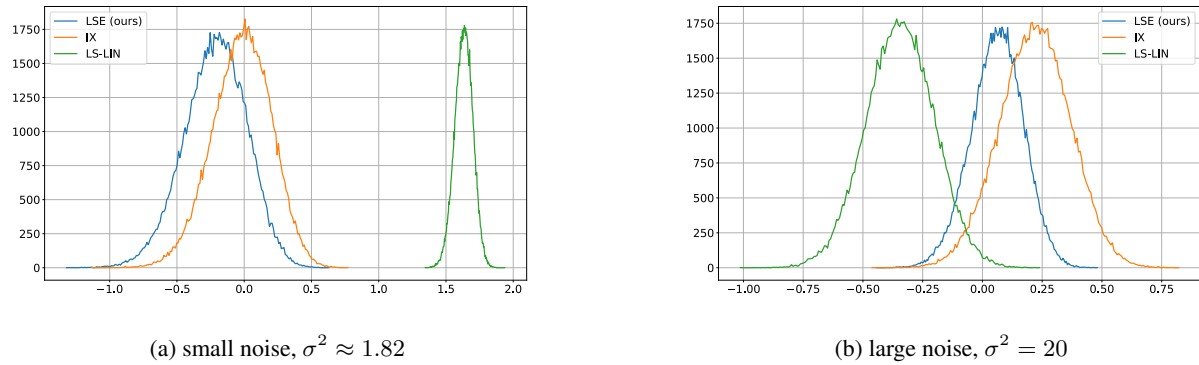

(a) small noise, $\sigma^2 \approx 1.82$

(b) large noise, $\sigma^2 = 20$

Figure 6: The error distribution of the LS-LIN, LSE, and IX estimators

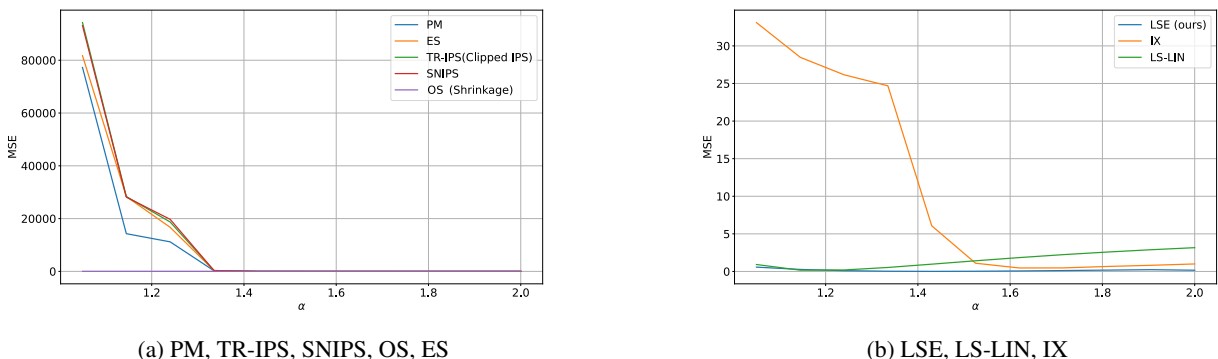

(a) PM, TR-IPS, SNIPS, OS, ES

(b) LSE, LS-LIN, IX

Figure 7: MSE of the PM, TR-IPS, SNIPS, OS, LS-LIN, IX, OS, ES, and LSE estimators over different values of $\alpha$

results, shown in Figure 7, reveal a clear split in estimator performance. The estimators - PM, ES, TR-IPS, OS, and SNIPS - struggle significantly with the heavy-tailed noise and show poor performance based on their MSE. In contrast, the more robust estimators - LSE, LS-LIN, and IX - maintain better performance across different noise levels, similar to what we observed in the Gaussian scenario.

Note that the IX estimator, despite having significantly less error than the poorly performing estimators, compared to LSE and LS-LIN is much worse in the tail of the noise.

For the distributional behavior of the estimators, we observe that except for LSE and LS-LIN, the estimators produce extreme outlier values. Error distribution is the distribution of the difference between the estimated value and the true value. Hence, we plot the error distribution of the LSE and LS-LIN with respect to noise in Figure 8. Here we see that in the small noise scenario LSE despite having more variance, is significantly less biased. Under large noise, LSE keeps the variance lower than LS-LIN, while showing the same bias. Hence, in both cases LSE achieves less MSE than LS-LIN and performs better in both small and large noise scenarios.

## G.10. Distributional properties in OPE

In this section, we investigate the error distribution of different estimators. In both Gaussian and Lomax settings, SNIPS, TR-IPS, OS, ES, and PM show extreme outlier values, but LS-LIN, LSE, and IX avoid outliers. In Figure 9, we show the error distribution of these estimators. We can see the competitive performance of IX and LSE in the Gaussian scenario, while LS-LIN induces a relatively large bias in this setting. In the Lomax setting, LSE has a bigger variance than IX and LS-LIN, while having significantly less bias. LSE has the property that it keeps bias significantly low while trading it for some small variance, leading to less MSE and better performance.

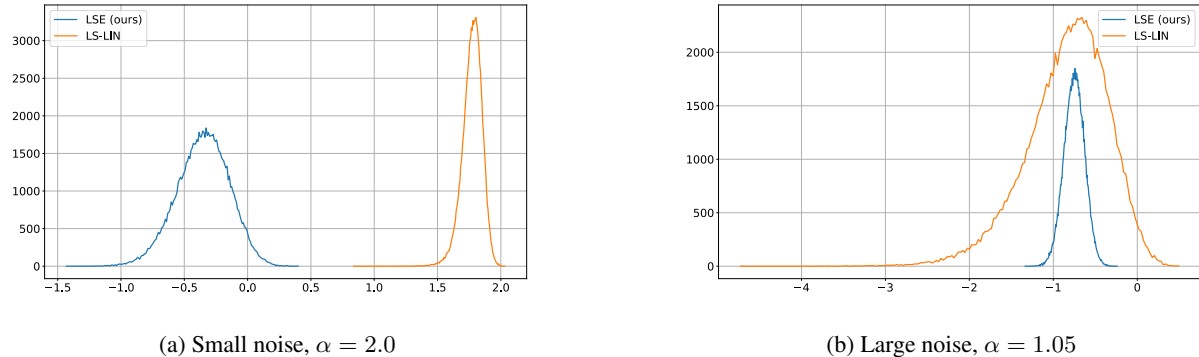

(a) Small noise, $\alpha = 2.0$

(b) Large noise, $\alpha = 1.05$

Figure 8: The error distribution of the LS-LIN and LSE estimators

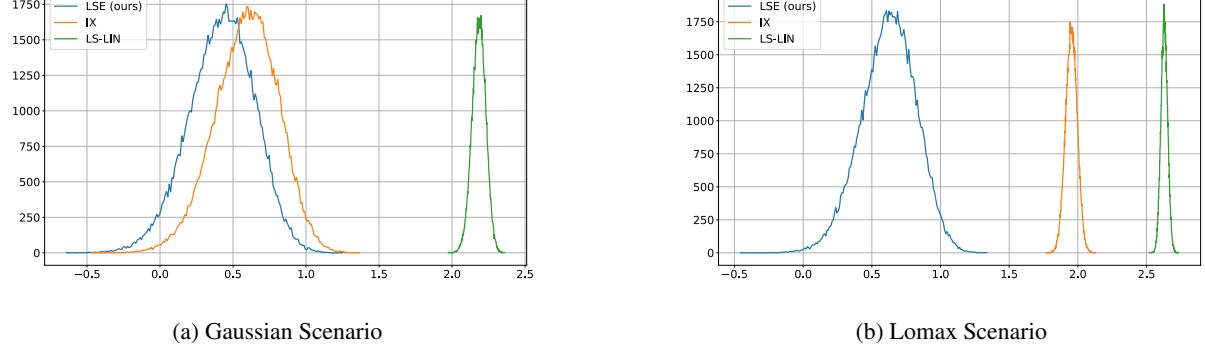

(a) Gaussian Scenario

(b) Lomax Scenario

Figure 9: The error distribution of the LS-LIN, IX, and LSE estimators

### G.11. More Comparison with LS Estimator

We conduct experiments to measure and compare the sensitivity of LSE and LS with respect to the selection of $\lambda$. To measure the sensitivity, we choose grid search method where we test the following set of $\lambda \in \{0.001, 0.01, 0.1, 1.0, 5.0\}$ in Table 29, adaptive method where $\lambda_n := \frac{1}{\sqrt{n}}$ is chosen, Table 30, and random method where $\hat{\lambda}$ chosen uniformly random from $[0, 1]$, Table 31. Then we compare these two estimators among these different methods of selecting $\lambda$. The results are reported below for Lomax setup.

Table 29: MSE of LSE and LS estimators with grid-searched for $\lambda \in 0.001, 0.01, 0.1, 1.0, 5.0$ and $\beta = 1.0, 1.5, 2.0$. The experiment was run 100000 times with different values of $\alpha$, $\alpha'$, and $\beta$.

| $\beta$ | $\alpha$ | $\alpha'$ | **Estimator** | Bias | Variance | MSE |
|---|---|---|---|---|---|---|
| 0.5 | 1.0 | 1.0 | LSE | 0.0362 | 0.0047 | 0.0060 |
| | | | LS | 0.0266 | 0.0047 | **0.0054** |
| | | 1.5 | LSE | 0.1693 | 0.0118 | 0.0404 |
| | | | LS | 0.0697 | 0.0346 | **0.0395** |
| | | 2.0 | LSE | 0.1590 | 0.0813 | **0.1066** |
| | | | LS | 0.3086 | 0.0238 | 0.1190 |
| 1.0 | 1.5 | 1.0 | LSE | 0.0728 | 0.0091 | 0.0144 |
| | | | LS | 0.0429 | 0.0104 | **0.0122** |
| | | 1.5 | LSE | 0.1065 | 0.0829 | 0.0942 |
| | | | LS | 0.1183 | 0.0717 | **0.0857** |
| | | 2.0 | LSE | 0.2726 | 0.1367 | **0.2111** |
| | | | LS | 0.2548 | 0.1785 | 0.2434 |
| 2.0 | 2.5 | 1.0 | LSE | 0.0302 | 0.0452 | 0.0461 |
| | | | LS | 0.0819 | 0.0281 | **0.0348** |
| | | 1.5 | LSE | 0.2245 | 0.1702 | **0.2206** |
| | | | LS | 0.2330 | 0.1699 | 0.2242 |
| | | 2.0 | LSE | 0.5345 | 0.2645 | **0.5502** |
| | | | LS | 0.4946 | 0.3696 | 0.6142 |

We can observe that in a close competitions, using the grid search method, LSE outperforms in 4 out of 9 experiments. With the adaptive method, LSE performs better in 7 out of 9 experiments, and when using the random method, LSE outshines in all 9 experiments.

### G.12. OPE on UCI datasets

We evaluate our method's performance in OPE by conducting experiments on 5 UCI classification datasets, as explained in Table 32,

We use the same supervised-to-bandit approach as in OPL experiments. Suggested by Sakhi et al. (2024), we consider a set of softmax policies as the target and logging policy. Consider an ideal policy as a softmax policy peaked on the true label of the sample. Moreover, a faulty policy is an ideal policy that has a set of its actions shifted by 1, hence, doing mostly wrong on the samples from the shifted labels. For the logging policy, we use faulty policies on the first $K/2$ actions with temperatures $\tau_0 = \{0.6, 0.7, 0.8\}$, and faulty policies on the last $K/2$ actions with $\tau = \{0.1, 0.3, 0.5\}$ as target policies, a total of 9 different experiments for each dataset. We create a bandit dataset using the logging policy $\pi_0$ and estimate the

Table 30: MSE of $\text{LSE}_{\lambda_n}$ and $\text{LS}_{\lambda_n}$ estimators with data-driven $\lambda_n = \frac{1}{\sqrt{n}}$ for $\beta = 1.0, 1.5, 2.0$ and $n = 1000$. The experiment was run 100000 times with different values of $\alpha$, $\alpha'$, and $\beta$.

| $\beta$ | $\alpha$ | $\alpha'$ | **Estimator** | Bias | Variance | MSE |
|---|---|---|---|---|---|---|
| 0.5 | 1.0 | 1.0 | $\text{LSE}_{\lambda_n}$ | 0.0816 | 0.0029 | **0.0096** |
| | | | $\text{LS}_{\lambda_n}$ | 0.1314 | 0.0028 | 0.0200 |
| | | 1.5 | $\text{LSE}_{\lambda_n}$ | 0.2756 | 0.0054 | **0.0814** |
| | | | $\text{LS}_{\lambda_n}$ | 0.2841 | 0.0073 | 0.0880 |
| | | 2.0 | $\text{LSE}_{\lambda_n}$ | 0.4651 | 0.0063 | 0.2226 |
| | | | $\text{LS}_{\lambda_n}$ | 0.4476 | 0.0099 | **0.2103** |
| 1.0 | 1.5 | 1.0 | $\text{LSE}_{\lambda_n}$ | 0.1596 | 0.0053 | **0.0308** |
| | | | $\text{LS}_{\lambda_n}$ | 0.2610 | 0.0052 | 0.0733 |
| | | 1.5 | $\text{LSE}_{\lambda_n}$ | 0.4857 | 0.0091 | **0.2449** |
| | | | $\text{LS}_{\lambda_n}$ | 0.5129 | 0.0123 | 0.2754 |
| | | 2.0 | $\text{LSE}_{\lambda_n}$ | 0.7817 | 0.0100 | 0.6211 |
| | | | $\text{LS}_{\lambda_n}$ | 0.7645 | 0.0159 | **0.6004** |
| 2.0 | 2.5 | 1.0 | $\text{LSE}_{\lambda_n}$ | 0.3652 | 0.0111 | **0.1445** |
| | | | $\text{LS}_{\lambda_n}$ | 0.6177 | 0.0108 | 0.3924 |
| | | 1.5 | $\text{LSE}_{\lambda_n}$ | 0.9792 | 0.0169 | **0.9757** |
| | | | $\text{LS}_{\lambda_n}$ | 1.0722 | 0.0227 | 1.1723 |
| | | 2.0 | $\text{LSE}_{\lambda_n}$ | 1.4919 | 0.0180 | **2.2437** |
| | | | $\text{LS}_{\lambda_n}$ | 1.4952 | 0.0282 | 2.2637 |

Table 31: MSE of $\text{LSE}_{\hat{\lambda}}$ and $\text{LS}_{\hat{\lambda}}$ estimators with random $\hat{\lambda}$ for $\beta = 1.0, 1.5, 2.0$. The experiment was run 100000 times with different values of $\alpha$, $\alpha'$, and $\beta$.

| $\beta$ | $\alpha$ | $\alpha'$ | **Estimator** | Bias | Variance | MSE |
|---|---|---|---|---|---|---|
| | | 1.0 | $\text{LSE}_{\hat{\lambda}}$ | 0.3418 | 0.0139 | **0.1308** |
| | | | $\text{LS}_{\hat{\lambda}}$ | 0.6779 | 0.0640 | 0.5236 |
| 0.5 | 1.0 | 1.5 | $\text{LSE}_{\hat{\lambda}}$ | 0.6516 | 0.0247 | **0.4493** |
| | | | $\text{LS}_{\hat{\lambda}}$ | 0.8335 | 0.0581 | 0.7528 |
| | | 2.0 | $\text{LSE}_{\hat{\lambda}}$ | 0.8583 | 0.0262 | **0.7629** |
| | | | $\text{LS}_{\hat{\lambda}}$ | 0.9635 | 0.0491 | 0.9775 |
| | | 1.0 | $\text{LSE}_{\hat{\lambda}}$ | 0.6019 | 0.0365 | **0.3987** |
| | | | $\text{LS}_{\hat{\lambda}}$ | 1.2196 | 0.1783 | 1.6656 |
| 1.0 | 1.5 | 1.5 | $\text{LSE}_{\hat{\lambda}}$ | 1.0803 | 0.0594 | **1.2264** |
| | | | $\text{LS}_{\hat{\lambda}}$ | 1.4290 | 0.1521 | 2.1941 |
| | | 2.0 | $\text{LSE}_{\hat{\lambda}}$ | 1.3890 | 0.0603 | **1.9898** |
| | | | $\text{LS}_{\hat{\lambda}}$ | 1.6017 | 0.1237 | 2.6890 |
| | | 1.0 | $\text{LSE}_{\hat{\lambda}}$ | 1.1942 | 0.1180 | **1.5442** |
| | | | $\text{LS}_{\hat{\lambda}}$ | 2.4803 | 0.6019 | 6.7537 |
| 2.0 | 2.5 | 1.5 | $\text{LSE}_{\hat{\lambda}}$ | 2.0218 | 0.1686 | **4.2565** |
| | | | $\text{LS}_{\hat{\lambda}}$ | 2.7676 | 0.4797 | 8.1390 |
| | | 2.0 | $\text{LSE}_{\hat{\lambda}}$ | 2.5258 | 0.1654 | **6.5451** |
| | | | $\text{LS}_{\hat{\lambda}}$ | 3.0074 | 0.3796 | 9.4239 |

Table 32: UCI datasets specifications. $N$ is the number of samples, $K$ is the number of actions, and $p$ is the number of features.

| Dataset | $N$ | $K$ | $p$ |
|---|---|---|---|
| Yeast | 1,484 | 10 | 8 |
| Page-blocks | 5,473 | 5 | 10 |
| Optdigits | 5,620 | 10 | 64 |
| Satimage | 6,430 | 6 | 36 |
| Kropt | 28,056 | 18 | 6 |

expected reward of the $\pi_\theta$ which is calculated as below,

$$V(\pi_\theta) = \frac{1}{n} \sum_{i=1}^{n} \pi_\theta (y_i|x_i),$$

where $y_i$ is the true label of the data sample $x_i$. We also add a random uniform noise $\epsilon \sim \text{Uniform}(0, 1)$ to the policy logits before softmax. We ran each experiment in each setting 10 times and calculated the average MSE of each estimator over all 90 experiments. For hyperparameter selection, for LS, OS, IPS-TR, PM, and IX, we use their own proposals. For LSE and ES, we use $0.2$ of the dataset as a validation set to find the hyperparameter with the lowest MSE by grid search and evaluate the method on the remaining $0.8$ of the dataset. Table 33 illustrates this on the 5 datasets for different estimators.

Table 33: MSE of LSE, PM, ES, IX, OS, LS, IPS-TR and SNIPS estimators on 5 UCI classification datasets on the OPE task.

| Dataset | PM | ES | IX | OS | LS | IPS-TR | SN-IPS | LSE |
|---------|------|------|------|------|------|--------|--------|------|
| Yeast | 0.237 | 0.0096 | 0.0573 | 0.0131 | 0.0146 | 0.0255 | 0.0088 | **0.0077** |
| Satimage | 0.0033 | 0.0066 | 0.0057 | 0.0035 | 0.0047 | 0.0043 | 0.0086 | **0.0028** |
| Kropt | 0.0160 | 0.0041 | 0.0056 | 0.0169 | 0.0208 | 0.0189 | 0.0256 | **0.0015** |
| Optdigits | 0.0079 | 0.0066 | 0.0150 | 0.0076 | 0.0083 | 0.0098 | 0.0110 | **0.0042** |
| Page-Blocks | 0.0440 | **0.0002** | 0.0236 | 0.0487 | 0.0513 | 0.0445 | 0.0639 | 0.0008 |

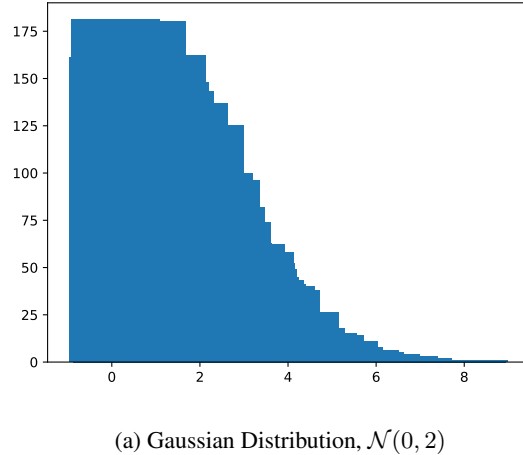

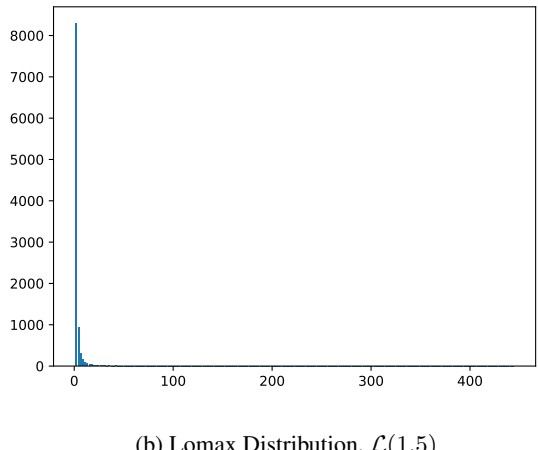

(a) Gaussian Distribution, $\mathcal{N}(0, 2)$            (b) Lomax Distribution, $\mathcal{L}(1.5)$

Figure 10: Histogram of 10K samples generated from Gaussian and Lomax distributions (we consider the absolute value of the Gaussian samples to focus on the tail of the distributions)

### G.13. Connection between heavy-tailed distributions and outlier modeling

We illustrate how heavy-tailed distributions can model outlier samples. Consider two sets of observations, the first one from a normal distribution $\mathcal{N}(0, 2)$ which has an exponential tail, and the second from a Lomax distribution $\mathcal{L}(1.5)$, which is heavy-tailed with $\epsilon = 0.5$. Figure 10 depicts the histogram of observed 10K samples from each distribution. We can observe that the Lomax distribution contains large, low-probability values (values around 400), but the total range for Gaussian observations is less than 10. The occurrence of sparse very low probability outlier values is possible by sampling from a heavy-tailed distribution like Lomax distribution. However, it does not hold for an exponential-tailed distribution like Gaussian. Hence, heavy-tailed distributions seem to be able to model scenarios with sparse large rewards or outliers, which is not possible using an exponential-tailed distribution. In the following, we discuss the heavy-tailed reward scenario in RL applications.

