# OpenReview forum: "Log-Sum-Exponential Estimator for Off-Policy Evaluation and Learning"
_ICML.cc/2025/Conference — ICML 2025 spotlightposter_

### Official Review · Reviewer_aZXo · 2025-02-24

**Overall Recommendation:** 3

**Summary:**

The paper introduces a Log-Sum-Exponential (LSE) estimator to address off-policy evaluation (OPE) and learning (OPL) in contextual bandit settings where rewards or propensity scores may be noisy or heavy-tailed. By applying a log-sum-exp transformation over importance-weighted rewards, this estimator improves robustness and variance control. The main theoretical contributions include a bound on the regret ($\mathcal{O}\left(n^{-\frac{t}{1+\varepsilon}}\right)$) under a ($1+\varepsilon$)-th moment assumption for the weighted reward and bias–variance analyses showing LSE can reduce variance compared to IPS. Experiments on synthetic data and supervised-to-bandit tasks (e.g., EMNIST) demonstrate lower MSE and higher accuracy than standard baselines (IPS variants, ES, PM, IX, etc.).

**Claims And Evidence:**

Major Claims
- Lower variance and bias–variance trade-off: By taking a log-sum-exp over the weighted rewards, the estimator becomes less sensitive to large outlier samples.


- Robustness under heavy-tailed reward distributions: They provide a theoretical analysis under a ($1 + \varepsilon$)-th moment assumption, which accommodates unbounded or heavy-tailed random variables.


- Favorable regret/convergence bounds in off-policy learning: They prove an $\mathcal{O}\left(n^{-\frac{\varepsilon}{1+\varepsilon}}\right)$ convergence rate, which interpolates between $\mathcal{O}(n^{-1/2})$ and $\mathcal{O}(n^0)$ depending on the bounding of higher moments.


- Empirical gains: The authors demonstrate in synthetic and real-ish tasks (EMNIST or KUAIREC-like data) that the LSE estimator can outperform baselines in terms of MSE (in OPE) or final policy accuracy (in OPL).

Evidence
- Theory: Theorem 5.3 (regret bounds) and Propositions 5.5/5.7 (bounds on bias and variance) offer analytical proof that LSE yields guaranteed convergence rates and improved variance control, provided certain moment assumptions.


- Experiments: On synthetic data, the LSE exhibits lower MSE and variance than standard IPS or other model-free estimators (PM, ES, IX, OS). On EMNIST-based bandit tasks, it achieves good accuracy even with noisy reward signals or estimated propensity scores.

These results collectively support the authors’ main claims. Some details (like data-driven choice of \lambda) are shown in appendices, along with further ablations and real-data results.

**Essential References Not Discussed:**

The paper covers the most relevant prior model-free approaches to mitigate variance in off-policy evaluation (IPS, truncated, ES, PM, etc.). It also references works on heavy-tailed or robust bandit algorithms. No obviously critical references are missing. I believe the submission includes enough prior work for the standard context.

**Experimental Designs Or Analyses:**

- Synthetic OPE: They design heavy-tailed reward distributions (Pareto-like or exponential) and random logging vs. target policies, measuring bias/variance/MSE across 10K replications.


- Supervised-to-bandit OPL: On EMNIST, the authors transform classification data into bandit feedback with partial logging. They vary logging policy quality (temperature \tau) and introduce both noisy propensity scores (modeled via inverse Gamma) and flipping reward noise.


- Baselines: They compare with up to 7–8 standard baseline methods.


- Metrics: MSE in OPE; classification accuracy for learned policies in OPL.


- Findings: LSE typically has lower MSE and variance than baselines in OPE, and yields better policy performance in OPL, especially under heavier reward tails or noisier propensity scores.

These experiment designs match the paper’s stated goals, although the authors might highlight more real-world scale tasks in future. The demonstration is nonetheless credible for a conference submission.

**Methods And Evaluation Criteria:**

Core method:
- The LSE estimator, $V_{\text{LSE}}^\lambda$​, applies $\frac{1}{\lambda} \log \left(\frac{1}{n} \sum_{i=1}^{n} e^{\lambda r_{i} w_{\theta}\left(a_{i}, x_{i}\right)}\right)$ with a tunable parameter $\lambda<0$.


Evaluation:
- In OPE, they empirically compare MSE, variance, and bias across multiple estimators.


- In OPL, a learned policy $\pi_{\theta}$ is optimized by maximizing the LSE-based objective. The main metric is regret or final classification accuracy after learning.


- Baselines: The paper carefully compares against standard IPS variants (e.g., truncated IPS), exponent smoothing (ES), power-mean (PM), self-normalized IPS (SNIPS), and so on.

All these benchmarks and metrics (variance, MSE, accuracy, regret) align with accepted practice in bandit OPE/OPL research.

**Other Comments Or Suggestions:**

See the above weaknesses section.

**Other Strengths And Weaknesses:**

Strengths
- This  log-sum-exp transformation is conceptually simple yet yields robust performance for unbounded or noisy data.


- Theoretical thoroughness: They provide formal bias-variance bounds and regret guarantees that unify heavy-tailed analysis with a single hyperparameter $\lambda$.


- Robustness: The authors show how the LSE estimator can remain stable even with substantial reward noise or propensity mis-specification.


- Empirical validations: They compare thoroughly to an array of baselines.

Weaknesses
- The paper discusses data-driven methods in the appendices, but real deployments might need more straightforward or adaptive selection heuristics.


- The approach is a single-pass aggregator, so computational complexity is no worse than other OPE methods, but potential memory or tuning overhead for extremely large datasets is not deeply addressed.


- They rely mostly on small to medium transformations (EMNIST or partial KUAIREC). The method’s performance in large, production-scale logs (with tens of millions of rows) is not tested.

**Questions For Authors:**

How does one best pick $\lambda$ in practical scenarios? Is there a recommended cross-validation strategy you find most reliable?

Could the LSE concept apply similarly in off-policy RL with trajectories? Are there any theoretical or practical obstacles?

Have you tried combining LSE with direct reward modeling or a doubly robust approach? If so, does the non-linear structure complicate it?

**Relation To Broader Scientific Literature:**

- The paper extends model-free OPE approaches (IPS, truncated IPS, ES, PM, IX, etc.) by introducing a non-linear transformation that can handle outliers.


- The approach is reminiscent of “robust mean estimation” under heavy tails (median-of-means, trimmed mean), but specialized to importance-weighted bandit feedback.


- The results connect to well-known bandit or offline RL settings under “pessimistic” or distribution-shift assumptions, though here it focuses specifically on the LBF dataset with (potentially) heavy-tailed or noisy rewards.

Overall, the authors do cite standard references (IPS, ES, PM, etc.) and situate their work among current OPE/OPL techniques. Additional connections to sub-Gaussian bounding and robust M-estimators are also discussed.

**Theoretical Claims:**

The authors present:

- Regret Analysis (Theorem 5.3 & Proposition 5.4): Shows the LSE-based OPL converges at $\mathcal{O}(n^{-\tfrac{\varepsilon}{1+\varepsilon}})$ under a heavy-tailed assumption on weighted rewards.


- Bias and Variance Bounds (Propositions 5.5 & 5.7): Provide an upper bound on LSE’s bias in terms of  $|\lambda|^{\varepsilon}$  and a straightforward variance bound that is no greater than that of IPS under second-moment assumptions.


- Robustness: Theorem 5.9 addresses noisy reward distributions, bounding the regret to show that total variation distance from the clean distribution plus the LSE’s own hyperparameter $\lambda$ determine the final bound.

These proofs are given at length in the appendices, citing standard concentration inequalities (Bernstein, etc.) and carefully handling non-linear transformations. The logic in the statements is consistent, and no major errors stand out in the derivations.

---

> ### Author Rebuttal · Authors · 2025-03-31
>
> We thank the reviewer for their careful reading and comments on the paper. We will address their concerns as detailed below.
>
> > single-pass aggregator
>
> **R1:**  Thanks for raising this point. It is true that theoretically, LSE is not separable and should be applied to the whole dataset. But, due to the nice gradient form of LSE, complete stochastic optimization of LSE is also possible for large-scale datasets.
> Suppose that $x_1, ..., x_{kl}$ is the data of $k$ batches of size $l$. LSE is a quasi-arithmetic mean function. So we have,
> $$\\mathrm{LSE}(x\_1, ..., x\_{kl}) = \\mathrm{LSE}\\left(\\mathrm{LSE}(x\_1, ..., x\_l), \\mathrm{LSE}(x\_{l \+ 1}, ..., x\_{2l}), ..., \\mathrm{LSE}(x\_{(k-1)l \+ 1}, ..., x\_{kl})\\right) \\\\
>     =\\mathrm{LSE}\\left(A\_1, A\_2, ..., A\_k\\right)
> $$
> Now we have,
> $$\\nabla\_{\\theta}\\mathrm{LSE}(x\_1, ..., x\_{kl}) \= \\sum\_{i=1}^{k}\\frac{d}{dA\_i}\\mathrm{LSE}(A\_1, ..., A\_k)\\nabla\_{\\theta}A\_i \= \<\\mathrm{softmax}(\\lambda \\mathbf{A}), \\nabla\_{\\theta}\\mathbf{A}\>
> $$
> So the gradient of the LSE on the entire data is a weighted average of the gradient of each batch, but these weights are not uniform as in the case of monte-carlo mean. We create the following procedure for SGD optimization.
>
> We store a coefficient $c\_i$ for each $A\_i$. Our final objective is to find $c_i$ such that $c_1, ..., c_k$ are proportional to $\\mathrm{softmax}(\\lambda A\_i)$. For this to happen, suppose we are at step $t \+ 1$ and we have, $c_1, ..., c_t$ as the coefficients of $A_1, ..., A_t$. We now want to find $c_{t + 1}$ and apply GD on $c\_{t \+ 1}A\_{t \+ 1}$. It is sufficient to have the following equality,
> $$ \\frac{e^{\\lambda A\_{t \+ 1}}}{e^{\\lambda A\_{t}}} \= \\frac{c\_{t+1}}{c\_t}  $$
> Hence, at step $t$, we apply gradient descent this way:
> $$ c\_{t \+ 1} = c\_t e^{\\lambda(A\_{t+1} \- A\_{t})}  \\theta^{(t \+ 1)} = \\theta^{t} \- \\eta c\_{t \+ 1}\\nabla\_{\\theta}A\_{t \+ 1} \\\\
> $$
> where $\\eta$ is the learning rate and $c\_1=1$. We will add this procedure for batch optimization as a discussion in the paper.
> > KUAIREC and real-world dataset
>
> **R2:** Thank you for the insightful comment. We agree that evaluating performance on production-scale datasets is crucial. KuaiRec’s 12M-interaction (big) dataset (7K users, 10K items) was used for training/validation, while a denser subset (small) was used for testing. The smaller subset was used solely for evaluation, given its density. We will clarify this in the final version to emphasize that our method has indeed been evaluated on a dataset of production-level scale. Furthermore, we conducted more experiments on more datasets, including the [**PubMed 200K RCT dataset**](https://anonymous.4open.science/r/icml2025_response-F5FC/response_rct.md).
>
> > Choosing $\lambda$ in practical scenarios
>
> **R3:** We have 3 types of $\lambda$ selection throughout the paper. One is based on validation data performance (gridsearch), one is data independent selection (App G.7), and the last one is data-driven selection (App G.8). The first method gives better performance and is recommended when it's computationally feasible, but for the large-scale datasets, the data-driven approach can work especially well, because it doesn't require any prior, or any heavy computations and can find a suitable $\lambda$ according to the data.
>
> >  LSE in off-policy RL with trajectories
>
> **R4:** Thank you for the insightful suggestion. In principle, the LSE operator could be applied in off-policy reinforcement learning, potentially serving as an alternative to traditional importance sampling. However, it brings theoretical (e.g., requiring i.i.d. assumptions) and practical challenges (e.g., computational overhead in algorithms like PPO [1]). We consider this a promising direction and plan to explore it in future work.
>
> > LSE with direct reward modeling or a doubly robust approach
>
> **R5:** Thank you for the suggestion. Estimators that incorporate reward modeling fall under model-based approaches, while those that do not are considered model-free. In this work, we primarily focus on model-free estimators. However, we did explore combining our estimator with the doubly robust (DR) approach, as discussed in Appendix G.3, and found that the resulting DR-LSE variant outperforms other baselines. Importantly, the non-linear structure of LSE does not introduce significant complications in this integration. For a fair comparison, we did not explore reward modeling based directly on the LSE framework, as our definition of LSE is grounded in weighted rewards rather than direct reward estimation. Nonetheless, we find this direction promising and plan to consider it in future work.
>
> ---
> **References:**
>
> [1]- Schulman, John, et al. "Proximal policy optimization algorithms."

---

### Official Review · Reviewer_6J3H · 2025-03-09

**Overall Recommendation:** 4

**Summary:**

The paper introduces a novel Log-Sum-Exponential (LSE) estimator for off-policy evaluation (OPE) and off-policy learning (OPL) in reinforcement learning, especially when dealing with logged bandit feedback datasets that may contain unbounded or heavy-tailed rewards. The paper analyzes the LSE estimator's regret bounds, bias, and variance, and it also explores its robustness to noisy rewards and propensity scores. The LSE estimator's performance is empirically compared against several baseline estimators, including truncated IPS, PM, ES, IX, Bandit-Net, LS-LIN, and OS, using both synthetic and real-world datasets. The document also offers theoretical insights into why the LSE estimator is well-suited for scenarios with heavy-tailed reward distributions and provides guidelines for selecting the estimator's parameter λ. The experimental code and data are also provided to support claims about its effectiveness.

**Claims And Evidence:**

The claims of this paper are generally supported by the argument in this paper.

**Essential References Not Discussed:**

Not aware of.

**Experimental Designs Or Analyses:**

The simulation experiments are comprehensive and convincing. That said, I think the experiments and validation could be stronger if data from RCTs can be used to evaluate the proposed LSE-based method.

**Methods And Evaluation Criteria:**

All evaluations are based on simulations. The results will be more convincing if data from randomized controlled trials can be used to validate the proposed method.

**Other Comments Or Suggestions:**

N.A.

**Other Strengths And Weaknesses:**

I think the paper can be strengthened with evaluations based on experimental data.

**Questions For Authors:**

Can the authors use RCT data to validate the proposed method?

**Relation To Broader Scientific Literature:**

The main contributions of this paper are three-fold. First, the authors develop a novel non-linear estimator based on the LSE operator, which substantially reduces the variance. Second, comprehensive theoretical performance guarantees of LSE-based OPE and OPL are provided. Third, simulated experiments show that the proposed estimator performs well compared with other SOTA algorithms.

**Theoretical Claims:**

The theoretical claims and proofs of the paper are rigorous and valid.

---

> ### Author Rebuttal · Authors · 2025-03-31
>
> We thank the reviewer for their comments, and generally positive assessment of the paper. We will address their concerns as detailed below.
>
> > The simulation experiments are comprehensive and convincing. That said, I think the experiments and validation could be stronger if data from RCTs can be used to evaluate the proposed LSE-based method. Can the authors use RCT data to validate the proposed method?
>
> **R1:** We appreciate the reviewer's suggestion for running experiments in RCT dataset. The result can be found in the [**following link**](https://anonymous.4open.science/r/icml2025_response-F5FC/response_rct.md).

---

### Official Review · Reviewer_Ve41 · 2025-03-21

**Overall Recommendation:** 4

**Summary:**

The paper proposes an estimator based on the log-sum-exponential (LSE) operator designed for off-policy evaluation (OPE) and off-policy learning (OPL) in contextual bandit settings. The LSE estimator addresses the issue of high variance in inverse propensity score (IPS) estimators by introducing robustness to noisy propensity scores and heavy-tailed reward distributions. The authors provide theoretical guarantees on the bias, variance, and regret for this estimator, with particular focus on its performance under heavy-tailed assumptions on weighted rewards. Empirical results from synthetic experiments and real-world datasets validate the practical effectiveness of the proposed method compared to existing estimators like IPS and others.

**Claims And Evidence:**

The paper makes different claims, which seem to be supported by theoretical results and empirical evidence.

First, it claims that the LSE estimator reduces variance and handles heavy-tailed reward distributions more effectively than the IPS estimator and its variants. This claim is supported by a detailed theoretical analysis, which includes bounds on bias, variance, and regret.

Empirically, the paper shows that the LSE estimator performs better in terms of mean squared error (MSE) and variance compared to competing methods in both synthetic and real-world experiments. This part, in my opinion, can be strengthened with an additional experiment.

**Essential References Not Discussed:**

I think that the authors did a good job in their related work sections and all the essential references are more or less present in the paper.

**Experimental Designs Or Analyses:**

The experimental design appears sound, but there are some limitations in the scope of the experiments. The experimental setup could benefit from more diverse datasets and a wider range of experimental conditions. For instance, experiments involving different types of reward distributions and more real-world applications

**Methods And Evaluation Criteria:**

The proposed LSE estimator is evaluated both theoretically and empirically. This is the usual way of evaluating OPE/OPL methods.

**Other Comments Or Suggestions:**

Very minor issue: there seem to be some inconsistencies in the References section

**Other Strengths And Weaknesses:**

In my opinion, the main weakness of the paper (apart from the experimental section which could be expanded, a point already discussed) is the apparent lack of novelty. Essentially, the paper applies the well-known log-sum-exponential (LSE) technique to OPE/OPL

However, despite this perceived lack of novelty, I believe the paper still meets the high standards of ICML due to its interesting theoretical analysis and the strong performance of the LSE estimator in the OPE/OPL domains. To the best of my knowledge, LSE has not previously been applied to OPE/OPL, and this work may represent the first contribution demonstrating that LSE can be a valuable addition to the OPE/OPL toolkit.

**Questions For Authors:**

- Could you clarify how the smoothing parameter $\lambda$ affects the performance of the LSE estimator?
- Have you considered experimenting with a broader range of real-world datasets to assess the robustness of the LSE estimator in different domains?

**Relation To Broader Scientific Literature:**

The paper is related to prior work in the area of off-policy evaluation and learning.
Also, it is related to heavy-tailed bandits, which is not typical in the OPE/OPL literature. I think that the authors did a good job in their related work sections, which position the paper with respect to specific related prior papers.

**Theoretical Claims:**

From what I could check, the theoretical claims in the paper seem well-supported.

---

> ### Author Rebuttal · Authors · 2025-03-31
>
> We thank the reviewer for their comments, and generally positive assessment of the paper. We will address their concerns as detailed below.
>
> > Novelty
>
> **R1:** We appreciate the reviewer’s feedback and the opportunity to clarify the novelty of our work. While it is true that we employ the log-sum-exponential (LSE) technique, one of our key contributions lies in the novel theoretical results we derive in the context of OPE/OPL. Specifically, our analysis establishes new insights that were not previously explored in the literature. These results go beyond a straightforward application of LSE, as we introduce a novel formulation and provide rigorous proofs that reveal deeper theoretical properties of the method in this setting under heavy-tailed assumptions.
>
> Additionally, while LSE is a well-known technique/operator, its application to OPE/OPL presents unique challenges, which we address through our theoretical analysis. We believe these contributions offer a meaningful advancement in understanding  LSE for heavy-tailed applications.
>
> We hope this clarification helps address the reviewer’s concern, and we would be happy to further elaborate on the specific theoretical novelties if needed.
>
> >  For instance, experiments involving different types of reward distributions and more real-world applications. Have you considered experimenting with a broader range of real-world datasets to assess the robustness of the LSE estimator in different domains?
>
> **R2:** We conducted more experiments on RCT dataset where the results can be found in the [**following link**](https://anonymous.4open.science/r/icml2025_response-F5FC/response_rct.md). In addition, additional experiments including multiple reward distributions are conducted. We fixed the distribution of the policies to be Gaussians with different locations. To test a variety of heavy-tailed distributions, we assume that we have a single state $s=s_0$ and when $a|s_0 \sim \pi_0$, the distribution of $r(s_0, a)$ is of a particular family. We considered Lomax, Generalized Extreme Value (GEV), T-student, and Fréchet distributions. We consider the absolute value of the samples to ensure the reward is positive. The table of the performance of our method compared to other methods is [**available here**](https://anonymous.4open.science/r/icml2025_response-FC68/response_Ve41_R2.md).
>
> >  Could you clarify how the smoothing parameter $\lambda$ affects the performance of the LSE estimator?
>
> **R3:** The effect of $\lambda$ for the supervised2bandit experiments where the reward is binary is investigated in App G.6, but for continuous heavy-tailed reward distributions, in the same setting mentioned in R2, we change $|\lambda| = \\{10^i | -3 \leq i \leq 2, i\in \mathbb{Z}\\}$ observe the change of MSE of the LSE estimator. The graphs are available at [**this link**](https://anonymous.4open.science/r/icml2025_response-F5FC/response_Ve41_R2.md).
>
> >  inconsistencies in the References section
>
>
> **R4:** Thanks for pointing out. It is fixed now.

---

> > ### Comment · Reviewer_Ve41 · 2025-04-05
> >
> > Thank you very much for the additional experiments and the clarifications!
> > They addressed many of my concerns. I raised my score accordingly.

---

> > > ### Author Response · Authors · 2025-04-06
> > >
> > > Dear Reviewer Ve41,
> > >
> > > We just wanted to sincerely thank you for taking the time to carefully read our rebuttal and for your thoughtful consideration of our responses. We truly appreciate the constructive feedback you provided throughout the review process, and we are grateful for your support and for the updated evaluation of our work.
> > >
> > > Your detailed comments and suggestions were very helpful to us in improving our paper, and we are glad that our clarifications could address your concerns.
> > >
> > > Thank you again for your time, effort, and support.
> > >
> > > Best regards,
> > >
> > > Authors

---

### Official Review · Reviewer_2WnN · 2025-03-24

**Overall Recommendation:** 4

**Summary:**

The paper proposes to use the log-sum-exponential operation as an off-policy estimator, proving bounds on the mean and variance of the estimates, as well as the performance gap between the optimal and learned policies in off-policy learning, and convergence rates. They follow this up with empirical evaluations.

The chosen estimator is chosen specifically to deal with the heavy-tailed distribution resulting from inverse propensity weights, so their analysis holds under heavy-tailed assumptions, which additionally holds for unbounded rewards, making the analysis applicable in a variety of domains.

**Claims And Evidence:**

There are two sets of claims to the paper.

The first is that the proposed estimator reduces variance, and has provable bounds on bias and variance with heavy tailed and noisy reward observations. Further, they provide regret bounds in the off-policy evaluation and learning setups, as well as the regret convergence under heavy-tailed settings which include unbounded reward. These claims are primarily and adequately supported by theoretical results.

The second set of claims is that the proposed estimator has competitive performance.

**Essential References Not Discussed:**

None that I am aware of

**Experimental Designs Or Analyses:**

Yes, the experimental designs for OPE and OPL were checked

**Methods And Evaluation Criteria:**

Yes, they do a standard setup (running various estimators with 10k trials of taking 1k samples, and calculating mean squared error and variance) for OPE on a synthetic data distribution and OPL on the EMNIST dataset.

**Other Comments Or Suggestions:**

none

**Other Strengths And Weaknesses:**

The work is original as far as I know.

The work is significant, being applicable to a common setting and addressing a key problem of heavy-tailed rewards.

The paper is consistently clearly written.

**Questions For Authors:**

It seems that these results could potentially be related to a Bayesian setting with exponential family distributions. Have the authors thought about this? I'm curious about the results, and it would increase my (already positive opinion of) the paper, even though I don't think it needs to be included here.

**Relation To Broader Scientific Literature:**

They compare against several similar off-policy estimators

**Theoretical Claims:**

Not carefully

---

> ### Author Rebuttal · Authors · 2025-03-31
>
> We thank the reviewer for their comments, and generally positive assessment of the paper. We will address their concerns as detailed below.
>
> > It seems that these results could potentially be related to a Bayesian setting with exponential family distributions. Have the authors thought about this? I'm curious about the results, and it would increase my (already positive opinion of) the paper, even though I don't think it needs to be included here.
>
> **R1:** We thank reviewer 2WnN for their insightful suggestion. An interesting direction for future work is to explore potential application of theoretical results in Bayesian inference frameworks, particularly through the lens of exponential family distributions and variational approximations. As the LSE can be interpreted as log-partition function in exponential family distribution, our methods can be applied in this field.

---

### Decision · Program_Chairs · 2025-05-01

**Decision:**

Accept (spotlight poster)

**Comment:**

This paper proposes to use log-sum-exp (LSE) operation as off-policy estimator for both OPE and OPL and demonstrate its effectiveness both theoretically and empirically. The paper received positive scores across the board, where all the reviewers praise the novelty of using LSE in the context of OPE/OPL, as well as the solid technical and theoretical contribution. There were some minor concerns/clarifications which were largely resolved during the author-reviewer discussion phase. Given the overall very positive feedback, I am voting for accept.